# Transient naive reprogramming corrects hiPS cells functionally and epigenetically

Sam Buckberry[1,2,3,4,23], Xiaodong Liu[5,6,7,8,9,10,11,23], Daniel Poppe[1,2,23], Jia Ping Tan[5,6,7,23], Guizhi Sun[5,6,7], Joseph Chen[5,6,7], Trung Viet Nguyen[1,2], Alex de Mendoza[1,2,12], Jahnvi Pflueger[1,2], Thomas Frazer[1,2], Dulce B. Vargas-Landín[1,2], Jacob M. Paynter[5,6,7], Nathan Smits[13], Ning Liu[14], John F. Ouyang[15], Fernando J. Rossello[5,6,7,20], Hun S. Chy[7,16], Owen J. L. Rackham[15,21], Andrew L. Laslett[7,16], James Breen[4,14], Geoffrey J. Faulkner[13,17], Christian M. Nefzger[5,6,22], Jose M. Polo[5,6,7,18,19 ✉] & Ryan Lister[1,2 ✉]

Cells undergo a major epigenome reconfiguration when reprogrammed to human induced pluripotent stem cells (hiPS cells). However, the epigenomes of hiPS cells and human embryonic stem (hES) cells differ significantly, which affects hiPS cell function[1–8]. These differences include epigenetic memory and aberrations that emerge during reprogramming, for which the mechanisms remain unknown. Here we characterized the persistence and emergence of these epigenetic differences by performing genome-wide DNA methylation profiling throughout primed and naive reprogramming of human somatic cells to hiPS cells. We found that reprogramming-induced epigenetic aberrations emerge midway through primed reprogramming, whereas DNA demethylation begins early in naive reprogramming. Using this knowledge, we developed a transient-naive-treatment (TNT) reprogramming strategy that emulates the embryonic epigenetic reset. We show that the epigenetic memory in hiPS cells is concentrated in cell of origin-dependent repressive chromatin marked by H3K9me3, lamin-B1 and aberrant CpH methylation. TNT reprogramming reconfigures these domains to a hES cell-like state and does not disrupt genomic imprinting. Using an isogenic system, we demonstrate that TNT reprogramming can correct the transposable element overexpression and differential gene expression seen in conventional hiPS cells, and that TNT-reprogrammed hiPS and hES cells show similar differentiation efficiencies. Moreover, TNT reprogramming enhances the differentiation of hiPS cells derived from multiple cell types. Thus, TNT reprogramming corrects epigenetic memory and aberrations, producing hiPS cells that are molecularly and functionally more similar to hES cells than conventional hiPS cells. We foresee TNT reprogramming becoming a new standard for biomedical and therapeutic applications and providing a novel system for studying epigenetic memory.

Somatic cell reprogramming requires substantial epigenome remodelling to establish states resembling hES cells. The generation of hiPS cells by the ectopic expression of the transcription factors OCT4, KLF4, SOX2 and MYC (hereafter referred to collectively as OKSM) is the most widely used method[9]. Despite the high similarity of induced pluripotent stem (iPS) cells and embryonic stem (ES) cells[10,11], substantial evidence indicates that iPS cells are epigenetically and functionally distinct from ES cells, including residual somatic cell epigenetic memory and de novo epigenetic aberrations[1–8]. Previous reports have shown that DNA methylation and histone modifications encode these epigenetic differences, which are transmissible through differentiation[1–4], limiting the potential use of hiPS cells in disease modelling, drug screening and cell therapies[12]. However, the mechanisms underpinning how aberrant epigenetic states emerge during reprogramming remain unknown.

The observation that cells reprogrammed by somatic cell nuclear transfer (SCNT) retain less epigenetic memory than OKSM-reprogrammed cells[13] indicates that epigenetic aberrations are not inherent to reprogramming and can be mitigated. Although the exact mechanisms are unknown, SCNT reprogramming appears to recapitulate the pre-implantation epigenome reset, mediated by the molecular environment within oocytes. Notably, although SCNT stem cells contain less epigenetic memory than hiPS cells[13], SCNT reprogramming requires donor oocytes, rendering the method inefficient, complex and unscalable.

Conventional OKSM reprogramming produces hiPS cells in a primed pluripotent state (primed-hiPS cells) resembling post-implantation epiblast cells[14,15]. Recent developments enable the reprogramming of somatic cells to a naive pluripotent state (naive-hiPS cells) resembling the pre-implantation epiblast, including low global DNA

methylation[16–18]. These two reprogramming paradigms provide tractable model systems to study how epigenome resetting is influenced by environments resembling distinct developmental states of pluripotency. Previous studies have focused on changes in DNA methylation when hES cells are switched between primed and naive culture conditions[19–21], but it is not known whether epigenetic memory and aberrations occur in naive-hiPS cell reprogramming. We therefore set out to study the origins, dynamics and mechanisms of epigenetic abnormalities in naive and primed reprogramming to comprehensively understand the reprogramming process.

## Divergent epigenome remodelling in hiPS cells

To investigate epigenome remodelling throughout naive and primed reprogramming, we reprogrammed human fibroblasts into both primed and naive pluripotent states using Sendai viral OKSM transcription factors[16], and isolated reprogramming intermediates throughout this process using intermediate cell surface markers[22] (Fig. 1a, Extended Data Fig. 1a,b and Supplementary Table 1). We then profiled DNA methylation using whole-genome bisulfite sequencing (WGBS) and analysed gene expression data previously generated by RNA sequencing (RNA-seq) from the same cells[22] (Fig. 1a). This enabled base-resolution quantification of the methylome throughout reprogramming. The largest changes in CG DNA methylation during primed reprogramming occur between days 13 and 21, with global levels reaching those similar to hES cells by passage 3 (Fig. 1b and Extended Data Fig. 1c). By contrast, most CG methylation changes in naive reprogramming occur before day 13 (Fig. 1b). As expected, naive conditions result in partial methylation at most CG dinucleotides (Extended Data Fig. 1c). Furthermore, intermediate levels of CG methylation in naive conditions is a result of sparse distribution of methylated CGs on individual DNA fragments, demonstrating that intermediate methylation is not caused by cell heterogeneity (Extended Data Fig. 1d).

CpH methylation (where H represents A, C or T) is a hallmark of pluripotent stem cells, and is mostly attributable to CA methylation (Extended Data Fig. 1e). We found that global CA methylation increases within the first 5 days of naive culture conditions, but after day 13 in primed reprogramming (Fig. 1c). Notably, we observed that CH methylation only accumulates upon changing cells to naive or primed culture conditions, concomitant with increased *DNMT3B* expression (Fig. 1c and Extended Data Fig. 1e,f).

Inspection of CG DNA methylation changes at regulatory elements revealed stepwise changes during primed reprogramming, but only one major change during naive reprogramming between days 7 and 13 (Fig. 1d). Fuzzy clustering identified five distinct classes of dynamic methylation at regulatory elements (Fig. 1e and Supplementary Table 2), with methylation changes generally occurring after, and being inversely correlated with, the expression change of linked genes (Fig. 1e and Extended Data Fig. 1g,h). This suggests that methylation changes at regulatory elements do not drive expression change during reprogramming but maintain repression, similar to reprogramming in mouse cells[23].

We then identified the transcription factor motifs associated with methylation changes at regulatory elements (Fig. 1f). Elements with increasing methylation during reprogramming (clusters 1–3) were enriched for the AP-1, JUN and FOS motifs, as was the transient cluster (cluster 5), which was also enriched for OCT4–SOX2 motifs (Fig. 1f). This is consistent with human and mouse studies suggesting that transcription factors at somatic enhancers are sequestered to transiently active elements bound by OKSM, which recruits transcription factors away from the loci maintaining somatic cell identity[22,23]. Demethylated regulatory elements featured OCT4–SOX2 motifs, and were associated with pluripotency genes, where expression increased after day 3 (cluster 4; Fig. 1e,f). Inspection of methylation changes driven by OKSM in fibroblast medium (up to day 7) revealed that 1,030 enhancers but only

39 promoters feature CG methylation loss of more than 20%, with these enhancers being enriched for AP-1 and pluripotency transcription factor motifs (Extended Data Fig. 1i). These time-course methylome profiles reveal that the first wave of epigenetic remodelling at regulatory elements is driven by OKSM, followed by distinct methylation states coincident with transitioning to primed and naive culture conditions.

## Emergence of aberrant DNA methylation

Several reports indicate that hiPS cells feature differentially methylated regions (DMRs) compared with hES cells that can be categorized as either somatic cell epigenetic memory or acquired aberrant methylation states that are unique to hiPS cells, which are not present in the cell of origin or hES cells[1–5,7,13,24]. Despite reports of DNA methylation differences between hiPS cells and hES cells, their temporal dynamics during reprogramming are not well characterized. We thus first identified CG-DMRs between multiple primed-hiPS cell and hES cell lines (Extended Data Fig. 1j). We identified 2,727 CG-DMRs (methylated CG (mCG)/CG difference >0.2; $P \le 0.05$), with 86.5% showing lower CG methylation levels in hiPS cells (Fig. 2a, Extended Data Fig. 1k and Supplementary Table 3). CG-DMRs could be classified as acquiring aberrant DNA methylation or retaining somatic cell epigenetic memory by comparing the DNA methylation levels between primed-hiPS cells and the fibroblasts that they originated from (Fig. 2b). This revealed that in primed-hiPS cells, 60.4% of the CG-DMRs were hypo-methylated relative to hES cells and showed less than 20% difference in methylation levels relative to fibroblasts, indicating somatic cell epigenetic memory, and an additional 24.2% of the CG-DMRs that were hypo-methylated relative to hES cells harboured higher methylation in primed-hiPS cells relative to fibroblasts, indicating partial epigenetic memory (Fig. 2b). Conversely, a majority of hyper-methylated CG-DMRs (54.2%) exhibited aberrant DNA methylation acquired during reprogramming, with methylation levels more than 20% higher than both fibroblasts and hES cells (Fig. 2b). Time-course analysis revealed that aberrant methylation begins to emerge between days 13 and 21 of primed reprogramming and continues to increase between day 21 and passages 3–10 (Fig. 2c). With memory CG-DMRs, minor transient demethylation (mCG/CG < 0.1) occurred in primed reprogramming (Fig. 2d), concordant with global CG methylation change (Fig. 1b). However, transitioning cells to naive medium triggered substantial demethylation in memory CG-DMRs by day 13 (Fig. 2d,e and Extended Data Fig. 1l,m). For hyper-methylated memory CG-DMRs, we observed demethylation to levels similar to those in hES cells by day 13 (Extended Data Fig. 1l). Overall, we found that aberrant CG methylation does not begin to accumulate upon OKSM induction during early reprogramming, and begins to emerge only after day 13 of primed reprogramming (Fig. 2c). Of note, aberrant CG hyper-methylation loci in primed-hiPS cells were not aberrant in naive reprogramming (Fig. 2c), indicating that aberrant hyper-methylation is a feature of primed and not naive reprogramming.

We next investigated DNA methylation at imprint control regions (ICRs), which are known to be abnormal in hiPS cells[25], with reports indicating that naive culture conditions triggers irreversible methylation loss at ICRs[16,20,21]. Analysis of CG methylation at known ICRs[21,26] revealed that imprints begin losing CG methylation between days 7 and 13, with the full loss of allele-specific methylation not occurring until after day 21 of naive reprogramming (Fig. 2e and Extended Data Fig. 1n). This indicates that demethylation at imprinted loci becomes more extensive the longer cells are cultured in naive conditions, and suggests that imprints may be maintained at day 13 of naive reprogramming.

## TNT reprogramming resets the epigenome

During early development, the pre-implantation embryo undergoes an epigenetic reset involving a wave of global demethylation, during which genomic imprints are protected from demethylation[27].

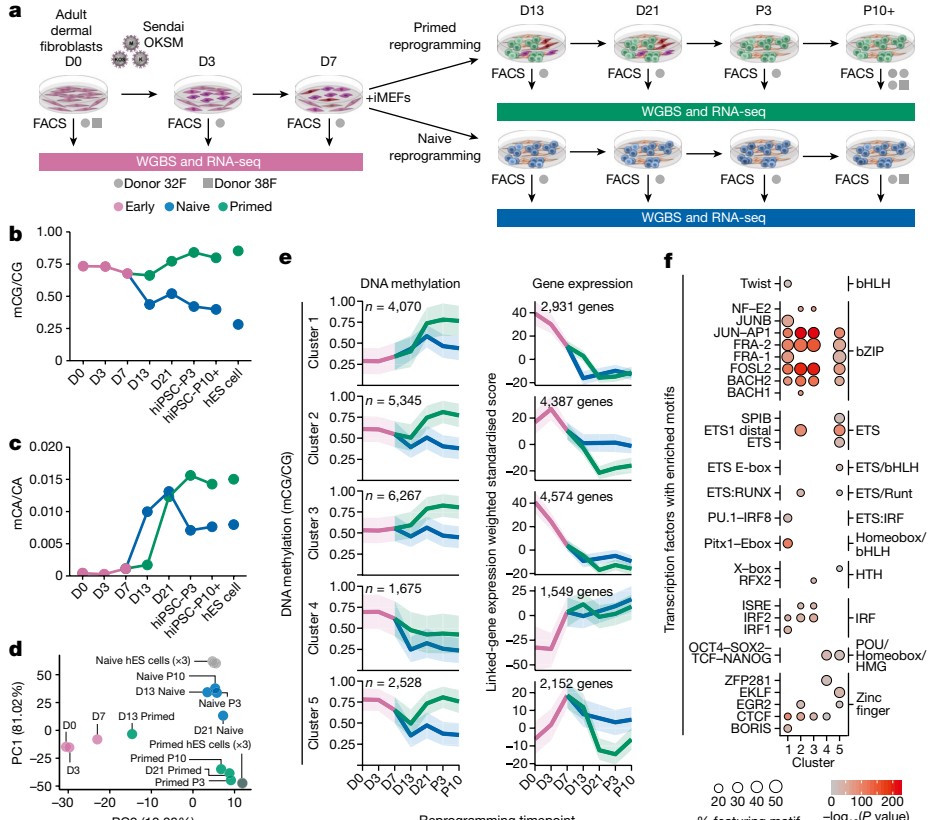

**Fig. 1 | Distinct trajectories of DNA methylation change during human naive and primed reprogramming. a**, Experimental design for time-course profiling of epigenomic changes that occur as cells are reprogrammed from fibroblasts to naive-hiPS and primed-hiPS cells. iMEFs, irradiated mouse embryonic fibroblasts; FACS, fluorescence-activated cell sorting. D indicates day of experiment and P indicates passage number. **b,c**, Dynamics of global CG methylation (**b**) and CA methylation (**c**) during naive and primed reprogramming compared with primed and naive hES cells. DNA methylation levels were calculated as a coverage-weighted mean (Methods). **d**, Principal component analysis of CG DNA methylation levels at GeneHancer regulatory elements throughout reprogramming. **e**, c-Means fuzzy cluster analysis of CG DNA methylation levels in regulatory elements throughout primed and naive reprogramming. Gene-expression plots of genes identified through GeneHancer's double-elite set of gene–enhancer validated pairs[47]. The line is the nonparametric boot strap mean and the ribbon shows the 99% confidence interval. **f**, Transcription factors (grouped by family) with significantly enriched motifs for DNA binding domains in regulatory elements for each cluster in **e**. Homer hypergeometric enrichment test; false discovery rate (FDR) < 0.01.

By combining our new understanding of epigenomic reconfiguration during reprogramming, we hypothesized that we could avoid somatic cell epigenetic memory and aberrant DNA methylation by reprogramming through a transient naive-like state, similar to the demethylation observed during embryonic development. Thus, we devised two experimental systems. In the first system, we reprogrammed fibroblasts with a transient naive culture treatment for 5 days after the initial 7 days of culturing in fibroblast medium, followed by culturing in primed medium for the remainder of the reprogramming (Fig. 3a), to give rise to transient-naive-treatment hiPS cells (TNT-hiPS cells). In the second system, we first established naive-hiPS cell colonies by extended naive culturing and then transitioned the cells to a primed pluripotent state to give rise to naive-to-primed hiPS cells (NTP-hiPS cells) (Fig. 3a).

We first confirmed that TNT-hiPS cells and NTP-hiPS cells were morphologically and molecularly similar to hES cells (Extended Data Fig. 2a). Testing for genetic aberrations in the hiPS cell lines revealed that two NTP-hiPS cell lines had megabase-scale deletions, and one primed-hiPS cell line had a deletion of about 600 kb, whereas we detected no aberrations in the TNT-hiPS cell lines (Extended Data Fig. 2b). When assessing CG-DMRs detected between primed-hiPS cell and hES cell lines, we observed that a majority of CG-DMRs show epigenetic correction to a state that is highly similar to hES cells for both TNT-hiPS (71.3%) and NTP-hiPS (77.8%) cells (Fig. 3b–d and Extended Data Fig. 2c–f). CG-DMR correction was highly concordant between the TNT and NTP systems (Extended Data Fig. 2c–f). Re-analysis of WGBS data from hiPS cells corrected by SCNT reprogramming[13] revealed that TNT-hiPS and NTP-hiPS cells have more CG-DMRs corrected compared to the 59.9% that are corrected in SCNT reprogramming (Extended Data Fig. 2g–i), indicating that TNT reprogramming is more effective at epigenetic correction.

We performed permutation testing to identify the genomic features that show a statistical over- or under-representation of CG-DMRs, revealing that corrected CG-DMRs are highly enriched in regions featuring the repressive histone modification H3K9me3 in fibroblast cells (z-score = 38.9; FDR < 0.01) but depleted in regions of hES cell-specific H3K9me3 (z-score = −4.5; FDR < 0.01; Fig. 3e and Extended Data Fig. 3a). Consistently, corrected CG-DMRs were over-represented in partially methylated domains (PMDs) in fibroblasts (z-score = 25.8; FDR < 0.01; Fig. 3e) and lamina associated domains (LADs) (z-score = 10.6; FDR < 0.01), which are known to co-occur with H3K9me3 in large domains of heterochromatin that are gene-poor, repressive and relate to higher order genome architecture[28]. We further analysed the relationship between CG-DMRs and repressive chromatin domains by performing H3K9me3 chromatin immunoprecipitation–sequencing (ChIP–seq). Regions enriched for H3K9me3 in fibroblasts that intersect with corrected CG-DMRs showed higher H3K9me3 in primed-hiPS cells compared with TNT-hiPS and NTP-hiPS cells, which were both more similar to hES cells (Fig. 3f), suggesting that repressive chromatin domains featuring epigenetic memory are reset by TNT reprogramming. Another epigenome feature that differs between hiPS cells and hES cells is

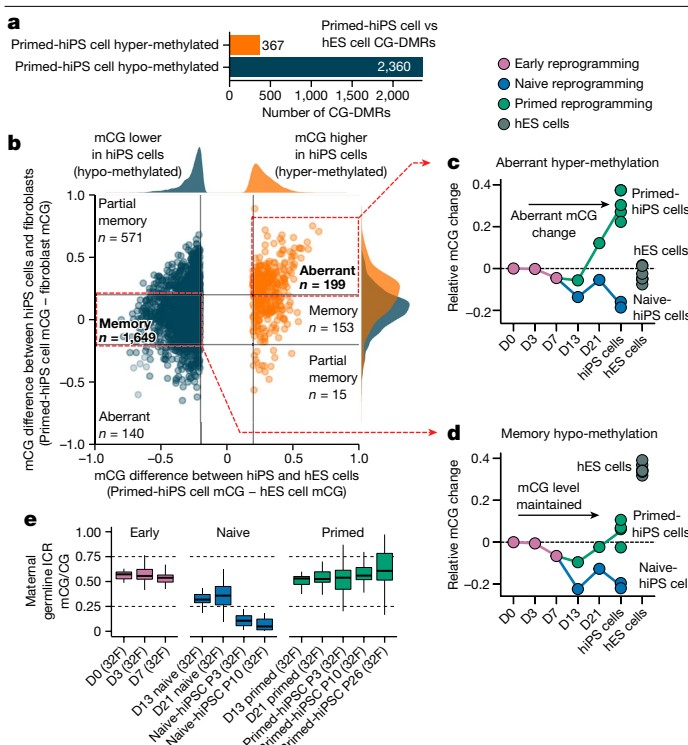

**Fig. 2 | Aberrant CG DNA methylation is acquired after day 13 of primed reprogramming and is absent in naive-hiPS cells. a**, Number of CG-DMRs detected in primed-hiPS versus hES cells. Hypo-methylated CG-DMRs are those that are less methylated in primed-hiPS cells than in hES cells, and hyper-methylated CG-DMRs are those that are more methylated in primed-hiPS cells than in hES cells. **b**, Relative CG DNA methylation difference at CG-DMRs in primed-hiPS cells versus hES cells (*x* axis) and fibroblasts (*y* axis). Each point on the graph represents an individual CG-DMR; blue points represent hypo-methylated DMRs and orange points represent hyper-methylated DMRs. The plot is divided into segments using a cut-off of 0.2 difference in mCG/CG between cell types for classification purposes. Kernel density estimate plots (top and right of the main graph) show the distribution of CG-DMR methylation differences for hypo- and hyper-methylated DMRs. **c,d**, Time-course of mean CG methylation change across aberrant hyper-methylated CG-DMRs (**c**) and hypo-methylated memory CG-DMRs (**d**) relative to the progenitor fibroblast state (day 0). Each point represents mean CG DNA methylation change compared to day 0 for individual samples. The hiPS cell time point includes all passages. **e**, Methylation at maternal germline ICRs throughout naive and primed reprogramming. In box plots, the horizontal line is the median, the box represents the interquartile range (IQR) and whiskers show either 1.5 × IQR or the data range. *n* = 1 independent experiment per box plot. ICRs are defined in ref. 21.

megabase-scale CH-DMRs, which collectively span 122.3 Mb (4.4%) of the WGBS-mappable genome, and co-occur with cell-of-origin H3K9me3[4,13]. When profiling CH-DMRs (defined in refs. 4,13), we found that CG-DMRs were highly enriched within them (Extended Data Fig. 3a). Moreover, 94.1% of CG-DMRs within CH-DMRs were corrected to an hES cell-like state, compared with 69.0% of CG-DMRs that do not overlap CH-DMRs (Extended Data Fig. 3b). TNT-hiPS and NTP-hiPS cells also showed a greater magnitude of CG methylation correction in CG-DMRs that overlap CH-DMRs (Extended Data Fig. 3c). Inspection of CA methylation in hypo-methylated CH-DMRs (*n* = 28) revealed that TNT-hiPS and NTP-hiPS cells have a CA methylation profile that is highly similar to hES cells, which is distinct from the low CA methylation levels observed in primed-hiPS cells (Fig. 3g and Extended Data Fig. 3d), in contrast to hyper-methylated CH-DMRs (*n* = 15; Extended Data Fig. 3e). We observed strong H3K9me3 enrichment in hypo-methylated CH-DMRs for primed-hiPS cells, at levels similar to those in fibroblasts, but TNT-hiPS and NTP-hiPS cells were more similar to hES cells, with markedly less H3K9me3 (Fig. 3h).

As existing hiPS cell lines may feature epigenetic anomalies, we tested whether culturing primed-hiPS cells in naive medium could correct aberrant DNA methylation. We generated primed-to-naive hiPS cells (PTN-hiPS cells) by culturing an established primed-hiPS cell line in naive medium for an extended period, and then transitioned these PTN-hiPS cells back into primed medium to produce primed–naive–primed-hiPS cells (PNP-hiPS cells). Attempts at TNT-like culturing of primed-hiPS cells (5 days in naive medium) caused extensive cell death and spontaneous differentiation when transitioning back to primed medium. PNP-hiPS cells exhibit remethylation and correction of a subset of the CG-DMRs detected between primed-hiPS cells and hES cells (Extended Data Fig. 3f), and show correction of many of the CH-DMRs (Extended Data Fig. 3g). Therefore, PNP reprogramming appears to correct aberrant DNA methylation patterns in primed-hiPS cells, although we observed increased variation in CG methylation at ICRs (Extended Data Fig. 3h). We emphasize that extended culturing of cells in some naive conditions may cause an increase in the frequency of genetic abnormalities[16,21]; therefore, although epigenetic correction is possible with PNP reprogramming, performing TNT reprogramming is optimal for minimizing genetic abnormalities and disruption of imprinting.

We then tested whether the improved qualities of TNT-hiPS cells result from clonal selection by randomly inserting a known DNA sequence into fibroblasts by lentiviral transduction and then reprogramming them by primed and TNT methods. Cas9-mediated enrichment and nanopore sequencing indicated that TNT-hiPS cells do not result from the selection of rare cell subpopulations (Extended Data Fig. 3i and Supplementary Table 4).

Our results indicate that large repressive chromatin domains associated with the nuclear lamina harbour epigenetic memory in primed-hiPS cells. For example, we detected a 1.7-Mb CH-DMR on chromosome 10 that was enriched for lamin-B1 in fibroblasts but not in hES cells, that also spans a cluster of 175 smaller CG-DMRs, intersects a larger fibroblast PMD and shows more than fivefold enrichment of H3K9me3 in fibroblasts and primed-hiPS cells, but not in TNT-hiPS and NTP-hiPS cells (Fig. 3i). Notably, aberrant epigenomic states in this large domain as well as other domains have been previously observed in primed-hiPS cells using a variety of progenitor cells and reprogramming methods[4,6,13]. The correction of CG and CH methylation and H3K9me3 in TNT-hiPS and NTP-hiPS cells demonstrates that the majority of epigenetic memory in hiPS cells can be corrected, and suggests that TNT reprogramming reorganizes chromatin architecture beyond what is achieved in conventional reprogramming. This reorganization may affect OKSM-mediated epigenome remodelling, as repressive chromatin domains are refractory to OKSM binding[29].

We then assessed the reproducibility of DMRs between studies, observing that even when processed with identical methods, the locations and number of CG-DMRs varies between studies (Extended Data Fig. 4a,b). However, the enrichment of CG-DMRs in repressive chromatin and CH-DMRs was similar across studies (Extended Data Fig. 4c,d,f). When assessing CA methylation using an identical set of CH-DMRs, we observe consistent reproducibility (Extended Data Fig. 4e,f). Principal component analysis revealed that principal component 1 (PC1) and PC2 captured study-dependent differences, whereas PC3 separated primed-hiPS cells and hES cells for all studies, and showed that TNT-hiPS cells were more similar to hES cells by this measure (Extended Data Fig. 4g–i).

Previous studies indicate that naive culturing triggers the loss of genomic imprinting, which is not recovered upon re-priming[16,20,21]. By contrast, we observed that TNT-hiPS cells have CG methylation patterns that are indicative of imprinting (Fig. 3j and Extended Data Fig. 5a). Analysis of WGBS reads—representative of single DNA molecules—showed equivalent proportions of unmethylated and methylated molecules at ICRs for TNT-hiPS cells, similar to fibroblasts (Fig. 3k and Extended Data Fig. 5b). This is in contrast to NTP-hiPS cells, in which we observed increased variance in the methylation levels at imprinted

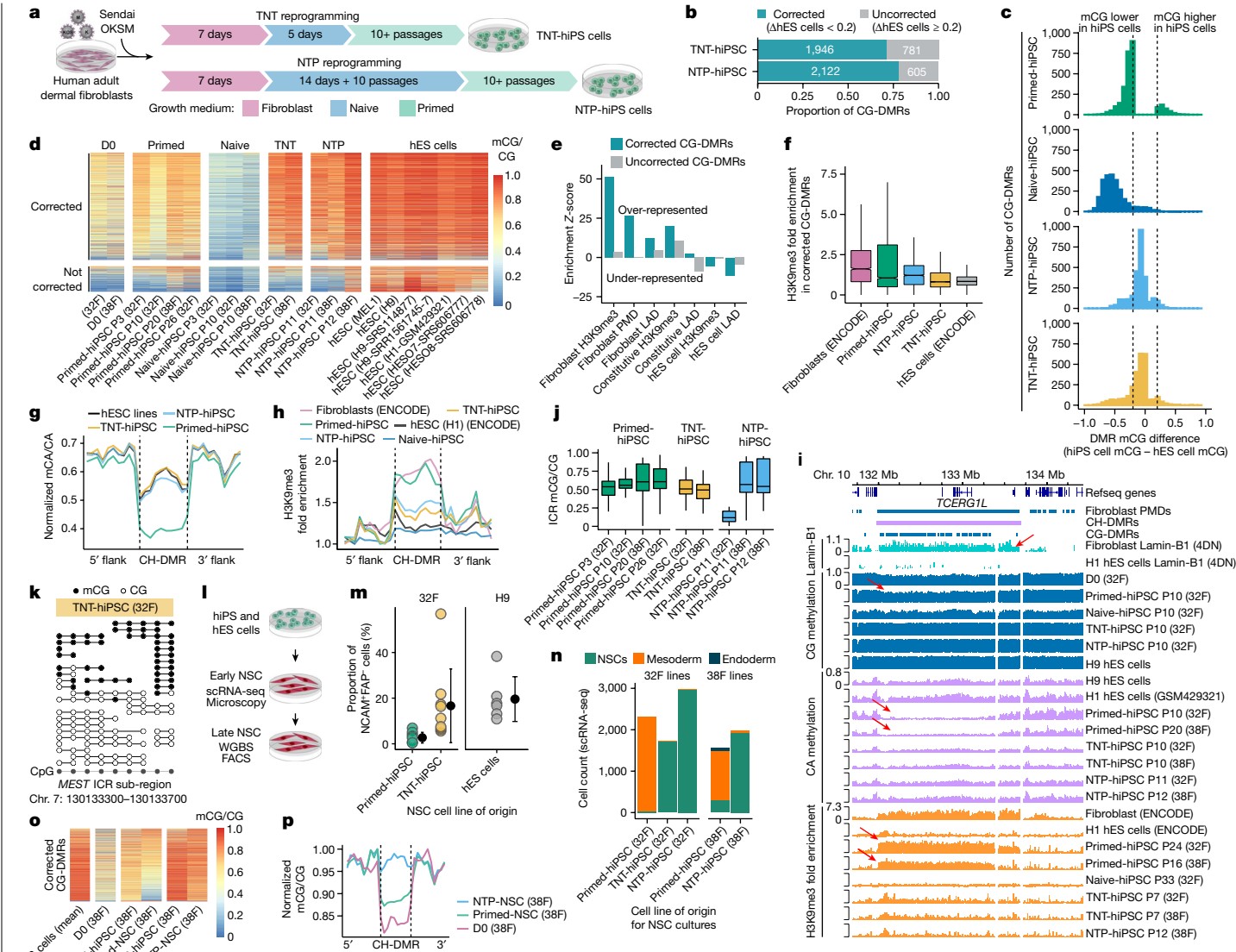

**Fig. 3 | Reprogramming through the naive state erases somatic cell memory and produces hiPS cells that closely resemble hES cells. a**, New reprogramming strategies for TNT-hiPS and NTP-hiPS cells. **b**, The proportion of CG-DMRs for primed-hiPS cells and hES cells corrected by TNT and NTP reprogramming to a difference of less than 0.2 mCG/CG. **c**, Differences in DNA methylation between hiPS and hES cells at CG-DMRs. Dashed lines indicate the threshold of 0.2 difference in CG-DMR methylation level. **d**, Methylation levels in corrected (top) and uncorrected (bottom) CG-DMRs. **e**, Enrichment permutation testing of corrected and uncorrected CG-DMRs in repressive chromatin. **f**, H3K9me3 enrichment in corrected CG-DMRs. Primed-hiPS: *n* = 2; TNT-hiPS: *n* = 2; NTP-hPSC, *n* = 3 independent experiments. In box plots, the horizontal line is the median, the box represents the interquartile range (IQR) and whiskers show either 1.5 × IQR or the data range. **g**, Aggregate profile of CA methylation in hypo-methylated CH-DMRs. Lines represent flank-normalized means. **h**, H3K9me3 enrichment in hypo-methylated CH-DMRs. Lines

represent flank-normalized means. **i**, Genome track of a CH-DMR intersecting a PMD, fibroblast LAD and a CG-DMR cluster. Arrows indicate partial CG methylation, CA methylation depletion and H3K9me3 enrichment in a fibroblast LAD. **j**, CG methylation in ICRs. The horizontal line is the median, the box represents the IQR and whiskers show either 1.5 × IQR or the data range. *n* = 1 independent experiment per box plot. **k**, WGBS reads at the *MEST* ICR. **l**, Schematic of NSC differentiation and profiling. **m**, The proportion of NCAM⁺FAP⁻ cells during differentiation into NSCs. Primed-hiPS: *n* = 9; TNT-hiPS: *n* = 9; H9-hES: *n* = 6 independent differentiation experiments. Data are mean ± s.d. **n**, Proportions of different cell types detected in early NSC cultures by single-cell RNA-seq (scRNA-seq). **o**, Methylation levels in CG-DMRs corrected by NTP reprogramming (as in Fig. 3d) in hiPS cells and derived NSC cultures. **p**, CG methylation (flank-normalized mCG/CG) in hypo-methylated CH-DMRs in NSCs and progenitor fibroblasts.

loci (Fig. 3j and Extended Data Fig. 5a). These data demonstrate that epigenetic memory erasure in TNT reprogramming can co-occur with maintenance of genomic imprinting. We then examined X chromosome inactivation in hiPS cell lines. CG methylation clustering of hiPS cell lines on the basis of 5-kb windows and promoters showed that none of the primed-hiPS, NTP-hiPS or TNT-hiPS cell lines clustered by hiPS cell type and were distributed among the hES cell lines (Extended Data Fig. 5c,d), indicating that TNT-hiPS and NTP-hiPS cells feature appropriate X chromosome inactivation.

## Correction persists through differentiation

Previous studies indicate that epigenetic memory and aberrations in primed-hiPS cells can persist through differentiation[1–4], which could functionally affect the resulting cells. We tested whether CG-DMR correction was maintained by differentiating primed-hiPS, TNT-hiPS and NTP-hiPS cells into neural stem cells (NSC) (Fig. 3l). We observed that NSC cultures derived from primed-hiPS cells produce many fibroblast-like cells in the early NSC cultures, similar to endoderm

differentiation[30]. Notably, these fibroblast-like cells did not emerge when differentiating TNT-hiPS and NTP-hiPS cells (Extended Data Fig. 5e). FACS quantification of NCAM⁺FAP⁻ cells in the differentiating culture revealed that TNT-hiPS cells differentiate more efficiently into NSCs, at a rate similar to hES cells (Fig. 3m). We characterized these cultures by scRNA-seq, revealing that early NSC cultures from fibroblast-derived primed-hiPS cells (which are of mesoderm origin) consist of 75.9–98.7% mesoderm-like cells (defined by the markers *BMP4*, *HAND1* and *TGFB1*), which were absent from NSC cultures generated from fibroblast-derived TNT-hiPS cells (0.35%) and NTP-hiPS cells (0.06–0.27%) (Fig. 3n and Extended Data Fig. 5f). After clearing the NSC cultures of fibroblast-like cells (by passaging at least 6 times), we performed WGBS profiling of the remaining NSCs to assess maintenance of corrected epigenetic states through differentiation. Whereas the hypo-methylation persisted at CG-DMRs in primed-hiPS cell derived NSCs, epigenetic correction was maintained for NSCs derived from NTP-hiPS cells (Fig. 3o). We then assessed CH-DMRs to inspect partial CG methylation, reflective of a PMD state, as this would suggest transmission of repressive chromatin of fibroblast origin. NSCs derived from primed-hiPS cells indeed maintained partial CG methylation, in contrast to NTP-hiPS cells, which showed high CG methylation levels suggestive of remodelling of repressive chromatin (Fig. 3p). These results indicate that epigenetic memory in primed-hiPS cells impairs differentiation efficiency and persists through differentiation.

## Isogenic evaluation of hiPS and hES cells

Up to this point, we have shown that TNT reprogramming epigenetically resets hiPS cells to a molecular state that is more similar to hES cells. However, previous reports suggest that genetic background variation may confound comparisons of pluripotent cell lines[31,32], including comparisons of hiPS cells and hES cells[11]. Therefore, we designed a series of isogenic reprogramming experiments to unambiguously compare hiPS cells and hES cells. We first differentiated hES cells into secondary fibroblast-like cells[11] and confirmed that they were CD90⁺TRA160⁻ and clustered with primary fibroblast lines based on CG methylation (Extended Data Fig. 6a–c). We then reprogrammed these secondary fibroblasts using the primed-hiPS, TNT-hiPS and NTP-hiPS cell protocols and performed WGBS, RNA-seq, assay for transposase-accessible chromatin with sequencing (ATAC–seq) and H3K9me3 ChIP–seq (Fig. 4a).

To visualize the differences between the isogenic hiPS cells and hES cells, we calculated principal components for global measures of CG and CA methylation, chromatin accessibility, gene and transposable element expression and H3K9me3 enrichment (Fig. 4b). This confirmed that even when controlling for genetic differences, TNT-hiPS cells are consistently highly similar to hES cells, whereas primed-hiPS cells are molecularly distinct. Next, we performed differential testing for CG-DMRs, gene and transposable element expression and ATAC–seq peaks for hES cells versus primed-hiPS, TNT-hiPS and NTP-hiPS cells (Fig. 4c,d). We detected 2,709 CG-DMRs for primed-hiPS cells (mCG difference >0.2; FDR <0.05), and only 358 for TNT-hiPS and 1,200 for NTP-hiPS cells (Fig. 4d, Extended Data Fig. 6d–h and Supplementary Table 5). Moreover, TNT-hiPS and NTP-hiPS cells also showed CA methylation levels in CH-DMRs similar to their origin hES cells, contrary to primed-hiPS cells (Fig. 4e and Extended Data Fig. 6i).

We identified 994 genes that were differentially expressed between isogenic primed-hiPS cells and hES cells (log₂-transformed fold change (FC) > 1, FDR <0.05), however these differences were largely ameliorated in TNT-hiPS and NTP-hiPS cells, with only 95 and 165 genes being differentially expressed, respectively (Fig. 4c,d, Extended Data Fig. 7a and Supplementary Table 6). When assessing the relationship between differential gene expression and promoter CG-DMRs, we observed that differential methylation is associated with gene-expression change (Extended Data Fig. 7b and Supplementary Table 7). For primed-hiPS

cells, 172 out of 547 (31.4%) of promoter CG-DMRs showed associated differential expression, whereas only 49 out of 215 (22.7%) of promoter CG-DMRs in TNT-hiPS cells had linked gene-expression differences. Gene ontology analyses revealed that genes that were differentially expressed in primed-hiPS cells are enriched for mesoderm development, among other terms (Supplementary Table 6). We then profiled the expression of genes with mesoderm-related ontologies, revealing that TNT-hiPS cells cluster more closely with hES cells than primed-hiPS cells (Extended Data Fig. 7c). Early mesoderm differentiation markers for WNT signalling (*WNT5A*, *WNT3* and *WNT11*) and mesoderm progenitor markers (*BMP4*, *MESP1* and *FOXC1*) showed increased expression in primed-hiPS cells compared with hES cells, which is largely corrected in TNT-hiPS cells (Extended Data Fig. 7d). Inspection of fibroblast-specific genes that retain their expression in primed-hiPS cells showed that primed-hiPS cells feature a gene-expression signature with elements of the fibroblast state that are not observed in TNT-hiPS or NTP-hiPS cells (Extended Data Fig. 7e), further demonstrating that the molecular memory of the cell of origin in primed-hiPS cells is corrected by TNT reprogramming.

When testing for differences in chromatin accessibility, we observed 411 differential ATAC–seq peaks between hES cells and primed-hiPS cells, whereas only 3 peaks were different between hES cells and TNT-hiPS cells, making them practically indistinguishable (log₂FC > 2, FDR <0.05; Fig. 4c,d and Extended Data Fig. 8a,b). NTP-hiPS cells exhibited 483 differential peaks, but not the same direction as primed-hiPS cells (Fig. 4d and Extended Data Fig. 8a). Motif analysis showed that primed-hiPS cells lack accessibility at loci enriched for OKSM binding motifs, and regions with uniquely accessible chromatin in primed-hiPS cells are enriched for transcription factors associated with differentiation (Extended Data Fig. 8c).

For genomic imprinting, TNT-hiPS cells did not show extensive demethylation at ICRs, in contrast to NTP-hiPS cells, which more closely resembled naive-hiPS cells (Extended Data Fig. 8d), consistent with previous reports of naive cultured hES cells showing imprinting loss when re-primed[20]. Clustering analysis based on imprinted gene expression also showed that TNT-hiPS cells were more similar to hES cells than NTP-hiPS cells (Extended Data Fig. 8e), and differential expression testing indicated imprinting loss in NTP-hiPS cells, but not in TNT-hiPS cells, for genes including *PEG3*, *MEG3* and *KCNQ1* (Supplementary Table 6). Moreover, when examining the relationship between CG methylation at ICRs with the change in expression of the linked imprinted gene, NTP-hiPS cells showed the greatest loss of imprinting at the expression level, with TNT-hiPS cells being the most similar to hES cells (Extended Data Fig. 8e,f). This further demonstrates that loss of imprinting is caused by extended naive culturing and can be avoided with TNT reprogramming.

As transposable element expression signatures are characteristic of different pluripotent cell states[20,33–35], we next tested for differential abundance of transposable elements between hES cells and hiPS cells. We identified 246 up-regulated and 13 down-regulated transposable elements in primed-hiPS cells (log₂FC >1, FDR <0.05; Fig. 4c,d). Notably, these differences were almost completely abolished by TNT reprogramming, with only 8 up- and 2 down-regulated transposable elements, whereas NTP-hiPS cells still showed 65 differentially expressed transposable elements (Fig. 4c,d, Extended Data Fig. 8g and Supplementary Table 8). We further found that genes within 50 kb of up-regulated transposable elements frequently showed upregulation in primed-hiPS cells, but not in TNT-hiPS or NTP-hiPS cells (Extended Data Fig. 8h). We also observed enrichment of primed-hiPS cell ATAC–seq peaks at long terminal repeat (LTR) transposable elements, co-occurring with reduced CG methylation (Fig. 4f). Closer inspection revealed that the up-regulated transposable elements in primed-hiPS cells are predominantly human endogenous retrovirus subfamily H (HERV-H) elements (80%, 197 out of 246) and their flanking LTR7 sequences, and that primed-hiPS cells express distinct copies of these elements compared with those expressed in naive-hiPS cells

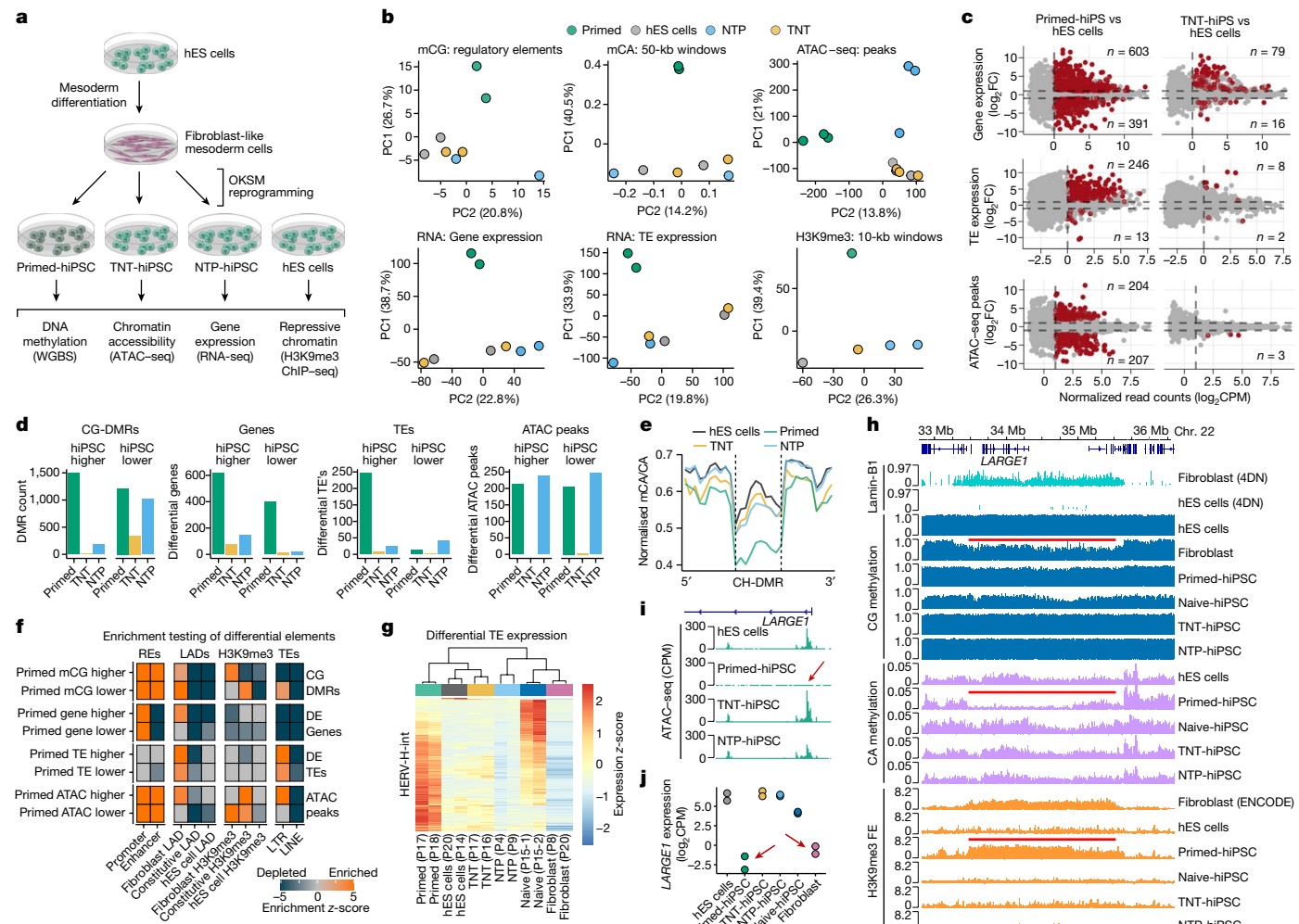

**Fig. 4 | The isogenic differentiation and reprogramming system confirms that TNT reprogramming enhances epigenome resetting. a**, Experimental design for differentiating hES cells to fibroblast-like cells and then reprogramming them to hiPS cells using the primed, TNT and NTP methods. **b**, Principal component analysis of CG methylation at GeneHancer elements, mCA/CA of 50-kb genome windows, normalized ATAC–seq read counts in peaks, normalized global gene expression, normalized global transposable element (TE) expression and normalized H3K9me3 ChIP–seq read counts. Data were quantile-normalized counts per million (CPM). **c**, Differential-testing MA plots for gene expression (determined by RNA-seq), TE expression (RNA-seq), and chromatin accessibility (ATAC–seq) for hiPS cells versus hES cells. Red points indicate FDR <0.05. Numbers on plots enumerate the 'up' or 'down' significant-features counts for each comparison. **d**, Differential testing of hES cells versus hiPS cell types for CG-DMRs, gene expression, TE expression and ATAC–seq peaks. 'hiPS cell higher' indicates that the value is higher in hiPS cells

than in hES cells, and 'hiPS cell lower' indicates that the value is lower in hiPS cells than in hES cells. **e**, Aggregate profile plot of CA methylation levels in hypo-methylated CH-DMRs. **f**, Permutation testing enrichment (z-scores) of differential elements. z-scores larger than 5 were reduced to 5 for visualization. REs, regulatory elements. **g**, Relative expression heatmap of HERV-H-int elements that are differentially expressed between hES cells and primed-hiPS cells (n = 167). **h**, Genome track of a CH-DMR region detected in hES cells versus primed-hiPS cells and associated epigenomic features. Red lines show fibroblast LAD, fibroblast PMD in the primed-hiPS cells and fold enrichment (FE) of H3K9me3 in primary fibroblasts, as indicated. **i**, Normalized ATAC–seq signal at the *LARGE1* promoter. The red arrow highlights the absence of an ATAC–seq peak in primed-hiPS cells. **j**, Gene expression of *LARGE1* in isogenic hES cells, hiPS cells and progenitor fibroblasts. Red arrows indicate repression in primed-hiPS cells and fibroblasts.

(Fig. 4g, Extended Data Fig. 8i and Supplementary Table 8). This is exemplified by the up-regulated HERV-H-int_dup2429 copy in primed-hiPS cells, featuring reduced DNA methylation and a 5' ATAC–seq peak, neither of which are present in the hES or TNT-hiPS cells (Extended Data Fig. 8j). We further validated our observations that transposable element expression is also different between hiPS cells and hES cells by performing the same transposable element differential expression analyses on two published RNA-seq datasets[11,13] (Extended Data Fig. 8k,l). We observed that transposable element expression in primed-hiPS cells can be partially corrected by SCNT reprogramming (Extended Data Fig. 8l), further demonstrating that dysregulation of transposable elements can be avoided by enhanced epigenome-resetting approaches[13]. The correction of abnormal transposable element expression is important, as it may

contribute to the phenotypic heterogeneity of hiPS cells and could lead to mutagenesis[36], and increased HERV-H expression can inhibit hiPS cell differentiation efficiency[37].

When analysing the relationship between differential DNA methylation, gene expression and chromatin states, we observed that fibroblast-associated repressive chromatin domains were highly enriched for the elements that we identify as significantly different in primed-hiPS cells (Fig. 4f). When inspecting an approximately 2-Mb fibroblast LAD on chromosome 22, we observed that primed-hiPS cells had a PMD with concomitant H3K9me3 enrichment similar to the fibroblast cells, but distinct from isogenic TNT-hiPS cells, NTP-hiPS cells and hES cells (Fig. 4h). Moreover, within this fibroblast LAD, the *LARGE1* promoter showed no chromatin accessibility in primed-hiPS cells, coupled

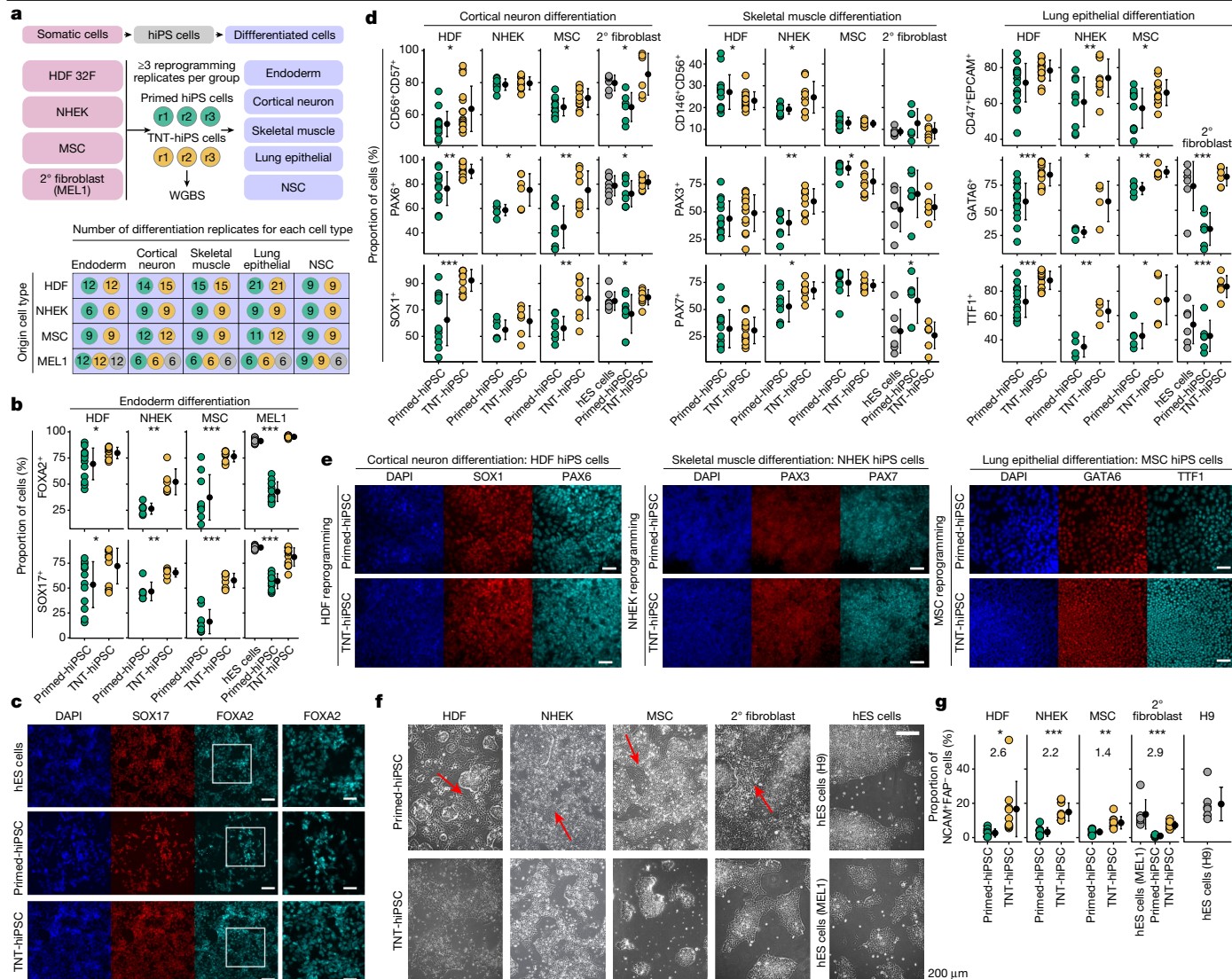

**Fig. 5 | Multi-lineage reprogramming and differentiation confirms that TNT reprogramming enhances differentiation. a**, Experimental design for multi-lineage primed and TNT reprogramming and differentiation into five cell types. Top, the four somatic cell lines reprogrammed into primed-hiPS cells and TNT-hiPS cells with three independent reprogrammings (r1–r3) performed per group, and with each subsequently differentiated into five different cell types, with independent replication. Bottom, the number of independent differentiation replicates performed for origin cell types (rows) and differentiated cell types (columns). Coloured circles represent primed-hiPS cell (green), TNT-hiPS cell (yellow) and hES cell (grey). 2° fibroblasts, secondary fibroblasts. **b**, Endoderm differentiation quantification for hiPS cells derived from secondary fibroblasts, showing the proportion of cells positive for FOXA2 and SOX17 by immunofluorescence analysis. **c**, Representative images from immunofluorescence analysis of FOXA2 and SOX17 in endoderm differentiation of hiPS cells derived from secondary fibroblasts. The outlined region is enlarged on the right. Scale bars, 100 μm (main image), 50 μm (enlarged

region). **d**, Quantification of multi-lineage cell differentiation in hiPS cell lines by FACS and immunofluorescence analyses using CD56, CD57 (FACS), PAX6 and SOX1 (immunofluorescence) for cortical neuron differentiation, CD146, CD56 (FACS), PAX3 and PAX7 (immunofluorescence) for skeletal muscle differentiation, and CD47, EPCAM (FACS), GATA6 and TTF1 (immunofluorescence) for lung epithelial differentiation. **e**, Representative images from immunofluorescence analysis of cell differentiation using SOX1 and PAX6 for cortical neurons, PAX3 and PAX7 for skeletal muscle, and GATA6 and TTF1 for lung epithelial cells. Scale bars, 50 μm. **f**, Phase-contrast images taken four days after passaging plated embryoid bodies during differentiation into NSCs. Large stretched-out fibroblast-like cells are evident during differentiation from primed-hiPS cells (red arrows). **g**, The percentage of NCAM+FAP− cells (from FACS analysis) after plating of embryoid bodies during NSC differentiation. log2FC values are shown on the graph. **d**,**g**, Data are mean ± s.d; two-sided *t*-test for primed versus TNT; \*\*\**P* < 0.0001, \*\**P* < 0.001, \**P* < 0.05. Details of replication are presented in Methods, 'Statistics and reproducibility'.

with strong transcriptional repression (Fig. 4i,j), also exemplified by the *MYH14–KCNC3* locus (Extended Data Fig. 9a). These examples highlight that lamina-associated megabase-scale regions of repressive chromatin that are present in differentiated cells are retained in primed-hiPS cells, but can be reset by reprogramming through the naive state. To further validate the ability of TNT reprogramming to produce hiPS cells that more closely resemble hES cells than those produced by conventional reprogramming, we evaluated published criteria[6,38,39] for using DNA

methylation and gene-expression signatures for selecting good hiPS cell clones, which indicated that TNT-hiPS cells would produce better hiPS cells for differentiation (Extended Data Fig. 9b–e).

## Improved differentiation of TNT-hiPS cells

Substantial evidence indicates that epigenetic memory in iPS cells affects differentiation; however, the functional differences between iPS

cells and ES cells remain topics of debate[1-3,11]. Therefore, we generated additional independent hiPS cell lines that were reprogrammed from primary human dermal fibroblasts (HDFs), keratinocytes (NHEK cells), mesenchymal stem cells (MSCs) and our hES cell-derived isogenic secondary fibroblasts to comprehensively test for differences in primed and TNT-hiPS cell differentiation capacity (Fig. 5a). We reprogrammed each origin somatic cell type in triplicate to produce both TNT-hiPS and primed-hiPS cells and then differentiated each hiPS cell line into definitive endoderm, cortical neurons, skeletal muscle cells, lung epithelial cells and neural stem cells.

We first performed WGBS and tested for CG-DMRs between primed and TNT-hiPS cells for each origin cell type to identify epigenetic differences that are not confounded by genetic differences. Clustering of samples on the basis of CG methylation in DMRs revealed that, irrespective of origin cell type, TNT-hiPS cells consistently cluster with hES cells, whereas primed-hiPS cells cluster more closely with their origin cells (Extended Data Fig. 9f). We again observed that CA methylation in TNT-hiPS cells was more similar to hES cells at CH-DMRs that are hypo-methylated in primed-hiPS cells, but note that the magnitude of difference for CA methylation between primed and TNT-hiPS cells from NHEK cells and MSCs was less than that observed for those from HDFs (Extended Data Fig. 9g). Testing for differences in CG methylation at ICRs revealed no differences between primed-hiPS and TNT-hiPS cells for reprogrammed HDFs, whereas TNT-hiPS cells from MSCs showed increased CG methylation at two ICRs, and at 15 out of 67 for hiPS cells reprogrammed from keratinocytes, although 8 of these were in a single cluster of secondary ICRs (Extended Data Fig. 9h). Despite the cell-of-origin-dependent differences, which may be due to different initial epigenomes and reprogramming kinetics, the DNA methylation differences between these additional primed-hiPS and TNT-hiPS cells were broadly consistent with the previously analysed lines (Figs. 3 and 4).

We then extensively tested the differentiation capacity of all these hiPS cell lines by FACS and immunofluorescence quantification (Fig. 5a, Extended Data Fig. 10, Supplementary Tables 9 and 10 and Supplementary Data 1). When assessing definitive endoderm differentiation, we observed that TNT-hiPS cells were consistently more efficient in differentiating into definitive endoderm compared with primed-hiPS cells, irrespective of the origin cell type (Fig. 5b,c and Extended Data Fig. 10b–d). Moreover, TNT-hiPS cells generated from secondary fibroblasts derived from hES cells, primary HDFs and MSCs differentiated more efficiently than primed-hiPS cells into both cortical neurons and lung epithelial cells, which both showed a greater proportion of cells expressing key markers of these cell types (Fig. 5d,e and Extended Data Fig. 10a–d; Methods). For skeletal muscle cell differentiation, both TNT-hiPS and primed-hiPS cells generated from MSCs, HDFs and secondary fibroblasts differentiated at similar efficiencies (Fig. 5d,e and Extended Data Fig. 10a–d; Methods). In the case of NHEK-derived hiPS cells, both primed-hiPS and TNT-hiPS cells differentiated at a similar efficiency into cortical neurons, but TNT-hiPS cells were more efficient at differentiating into lung epithelial cells and skeletal muscle cells than primed-hiPS cells (Fig. 5d,e and Extended Data Fig. 10c,d). Finally, during early differentiation into NSCs, when NSC colonies were forming, we again observed the spontaneous appearance of elongated fibroblast-like cells when the cells were derived from primed-hiPS cells, but not when they were derived from TNT-hiPS cells (Fig. 5f). Quantification of NSC differentiation efficiency showed that the proportion of NSCs (NCAM⁺FAP⁻) was consistently higher in cultures derived from TNT-hiPS cells than those derived from primed-hiPS cells and closer to the differentiation efficiency observed for hES cell lines (Fig. 5g). These reprogramming and differentiation experiments provide strong evidence that the epigenetic differences in primed-hiPS cells are associated with reduced differentiation capacity that can be attenuated by TNT reprogramming.

## Discussion

Our characterization of naive and primed reprogramming dynamics enabled new insights into the nature of epigenetic remodelling in iPS cells, guiding the development of the TNT reprogramming strategy. Our study extends previous work[1-4,13] by showing that epigenetic memory is concentrated in repressive chromatin domains from the cell of origin marked by H3K9me3, that are associated with the nuclear lamina in the origin cell type. We found that TNT reprogramming effectively erases epigenetic memory, particularly in regions of chromatin–lamina interactions, and improves differentiation. If a cell's response to differentiation cues depends on how chromatin is spatially organized to make loci available for transcription factor binding[40], the differentiation bias in primed-hiPS cells may be due to heterochromatic memory influencing transcription factor binding dynamics.

The more complete epigenome reset achieved through TNT reprogramming suggests that this strategy may mimic aspects of the epigenetic reset that occurs during human pre-implantation development. First, TNT reprogramming remodels H3K9me3 heterochromatin, which also occurs during early embryonic development before lineage-specific H3K9me3 is established post-implantation[41]. Second, TNT reprogramming facilitates transient genome-wide demethylation, similar to pre-implantation development[42]. Third, genomic imprints are protected from erasure during pre-implantation epigenome resetting, and our data indicate that the transient nature of TNT reprogramming can minimize loss of imprinting, as imprinting loss appears to be symptomatic of extended culturing in naive medium.

Our observation that HERV-H transposable elements show higher expression in primed-hiPS cells compared with hES cells—but not in TNT-hiPS cells—is particularly important, as aberrant HERV-H transcription has been reported to increase the chance of L1 transposable element mRNA expression initiated from HERV-H promoters, leading to mutagenesis in hiPS cells[43]. Previous studies suggest that transcriptional and epigenetic signatures present in hiPS cells can be donor-dependent, even in isogenic systems[11,44,45]. Here we independently verified that isogenic primed-hiPS cells and hES cells exhibit significant differences in gene expression, but further demonstrated that these differences can be abolished through TNT reprogramming. This indicates that the epigenome has an important role in driving the differences between hES cells and hiPS cells. Moreover, our differentiation experiments demonstrate that genetically matched TNT-hiPS cells have an enhanced and more homogeneous differentiation potential than primed-hiPS cells.

By leveraging the TNT reprogramming system, we have revealed the functional benefit of more completely resetting the epigenome. Prior to this work, SCNT reprogramming was the only method shown to improve DNA methylation anomalies[13]. However, SCNT-reprogrammed cells can still feature persistent cell-of-origin H3K9me3 heterochromatin[46], and the technique is difficult and unfeasible to scale. Our work shows that TNT reprogramming is a practical and scalable approach to overcome these intrinsic characteristics of hiPS cells, which is important for the clinical delivery of this technology. As TNT reprogramming enables high-fidelity resetting of the epigenome and transcriptome along with improved differentiation, we view this as a powerful model system for studying epigenetic memory and the mechanisms maintaining cell-of-origin heterochromatin.

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

[1]Harry Perkins Institute of Medical Research, QEII Medical Centre and Centre for Medical Research, The University of Western Australia, Perth, Western Australia, Australia. [2]ARC Centre of Excellence in Plant Energy Biology, School of Molecular Sciences, The University of Western Australia, Perth, Western Australia, Australia. [3]Telethon Kids Institute, Perth, Western Australia, Australia. [4]John Curtin School of Medical Research, College of Health and Medicine, Australian National University, Canberra, Australian Capital Territory, Australia. [5]Department of Anatomy and Developmental Biology, Monash University, Melbourne, Victoria, Australia. [6]Development and Stem Cells Program, Monash Biomedicine Discovery Institute, Melbourne, Victoria, Australia. [7]Australian Regenerative Medicine Institute, Monash University, Melbourne, Victoria, Australia. [8]School of Life Sciences, Westlake University, Hangzhou, China. [9]Research Center for Industries of the Future, Westlake University, Hangzhou, China. [10]Westlake Laboratory of Life Sciences and Biomedicine, Hangzhou, China. [11]Westlake Institute for Advanced Study, Hangzhou, China. [12]School of Biological and Behavioural Sciences, Queen Mary University of London, London, UK. [13]Mater Research Institute, University of Queensland, Brisbane, Queensland, Australia. [14]South Australian Health and Medical Research Institute, Adelaide, South Australia, Australia. [15]Program in Cardiovascular and Metabolic Disorders, Duke–National University of Singapore Medical School, Singapore, Singapore. [16]Biomedical Manufacturing, Commonwealth Scientific and Industrial Research Organisation, Melbourne, Victoria, Australia. [17]Queensland Brain Institute, University of Queensland, Brisbane, Queensland, Australia. [18]Adelaide Centre for Epigenetics, School of Biomedicine, Faculty of Health and Medical Sciences, The University of Adelaide, Adelaide, South Australia, Australia. [19]The South Australian Immunogenomics Cancer Institute, Faculty of Health and Medical Sciences, The University of Adelaide, Adelaide, South Australia, Australia. [20]Present address: Murdoch Children's Research Institute, Melbourne, Victoria, Australia. [21]Present address: School of Biological Sciences, University of Southampton, Southampton, UK. [22]Present address: Institute for Molecular Bioscience, University of Queensland, Brisbane, Queensland, Australia. [23]These authors contributed equally: Sam Buckberry, Xiaodong Liu, Daniel Poppe, Jia Ping Tan. ✉e-mail: jose.polo@monash.edu; ryan.lister@uwa.edu.au

# Methods

## Cell culture

All cell lines used and derived by different approaches in this study are listed in Supplementary Table 1. Detailed information about the experimental design, materials and reagents is presented in the Reporting Summary. Primary human adult dermal fibroblasts (HDFa) from three different female donors were obtained from Gibco (C-013-5C, lot no. 1029000 for 38F and lot no. 1569390 for 32F) and cultured following the manufacturer's recommendations. In brief, cells were thawed and plated into flasks in Medium 106 (Gibco) supplemented with low serum growth supplement (LSGS) (Gibco) for expansion. Cells were cultured in a 37 °C, 5% $O_2$ and 5% $CO_2$ incubator, and the medium was changed every other day. The use of human embryonic stem cells (H9 and MEL1) was carried out in accordance with approvals from Monash University and the Commonwealth Scientific and Industrial Research Organisation (CSIRO) Human Research Ethics Offices. Conventional primed-hiPS cells and H9 hES cells (WiCell Research Institute; http://www.wicell.org) were maintained as described in the below section. The cell lines used in this study were regularly tested and were mycoplasma negative. Human dermal fibroblasts and NHEKs were authenticated by ThermoFisher and Lonza, respectively, as per description in the CoA. hES cells were authenticated in the Laslett lab. MSCs were authenticated in the Heng lab. These cell lines were also routinely authenticated in-house via morphological assessment, immunofluorescence for identity markers, or RNA-seq.

## Cell culture media

Fibroblast medium: DMEM (ThermoFisher), 10% fetal bovine serum (FBS, Hyclone), 1% non-essential amino acids (ThermoFisher), 1 mM GlutaMAX (ThermoFisher), 1% penicillin-streptomycin (ThermoFisher), 55 μM β-mercaptoethanol (ThermoFisher) and 1 mM sodium pyruvate (ThermoFisher). Naive medium (t2iLGoY)[19]: 50:50 mixture of DMEM/F12 (ThermoFisher) and neurobasal medium (ThermoFisher), supplemented with 2 mM L-glutamine (ThermoFisher), 0.1 mM β-mercaptoethanol (ThermoFisher), 0.5% N2 supplement (ThermoFisher), 1% B27 supplement (ThermoFisher), 1% penicillin-streptomycin (ThermoFisher), 10 ng ml⁻¹ human leukaemia inhibitory factor (made in-house), 250 μM L-ascorbic acid (Sigma), 10 μg ml⁻¹ recombinant human insulin (Sigma), 1 μM PD0325901 (Miltenyi Biotec), 1 μM CHIR99021 (Miltenyi Biotec), 2.5 μM Gö6983 (Tocris), 10 μM Y-27632 (Abcam). Primed hiPS cell medium (KSR/FGF2): DMEM/F12 (ThermoFisher), 20% knockout serum replacement (KSR) (ThermoFisher), 1 mM GlutaMAX (ThermoFisher), 0.1 mM β-mercaptoethanol (ThermoFisher), 1% non-essential amino acids (ThermoFisher), 50 ng ml⁻¹ recombinant human FGF2 (Miltenyi Biotec), 1% penicillin-streptomycin (ThermoFisher). Primed hiPS cell medium (Essential 8 (E8)): 10 ml of E8 supplement (Gibco) to 500 ml medium basal (Gibco), supplemented with 1% penicillin-streptomycin (Gibco).

## Derivation of TNT-hiPS cells and NTP-hiPS cells

Human somatic cell reprogramming was performed as previously described[16,22,48]. In brief, early passages (<P6) fibroblast cells were seeded into 6-well plates at 50,000–70,000 cells per well before transduction in fibroblast medium. Cells in one well were trypsinized for counting to determine the volume of virus required for transduction (multiplicity of infection), and transduction was performed using the CytoTune 2.0 iPSC Sendai Reprogramming Kit (Invitrogen) consisting of four transcription factors (OCT4, SOX2, MYC and KLF4). Twenty-four hours later, the medium was removed, with subsequent medium changes performed every other day. For the derivation of primed-hiPS cells, cells were reseeded onto a layer of iMEFs on day 7 of reprogramming and transitioned to primed medium (KSR/FGF2 or E8 on vitronectin; Supplementary Table 1) on the next day. The cells were cultured to confluency (around day 18–21 of reprogramming) and

further passaged with Collagenase IV (ThermoFisher) for cell line establishment. For derivation of TNT-hiPS cells, the day 7 reprogramming intermediates were transitioned to naive medium (t2iLGoY) instead. When dome-shaped colonies were evident 5 days later, intermediate cells were collected using Accutase (Stem Cell Technologies) and reseeded onto a layer of iMEFs in naive conditions. The medium was switched to primed medium (KSR/FGF2 or E8; Supplementary Table 1) the following day. When the culture became confluent, cells were collected using collagenase IV and maintained in primed medium (KSR/FGF2 or E8; Supplementary Table 1) on iMEFs. Cells were cultured in a 37 °C, 5% $O_2$ and 5% $CO_2$ incubator with daily medium change. Cells are usually passaged every 4–5 days. For derivation of NTP-hiPS cells: after 16–18 days post-transduction (8–10 days in naive condition), naive-hiPS cells were collected using Accutase (Stem Cell Technologies) and passaged more than 10 times. The established naive-hiPS cells were confirmed by flow cytometry and immunostaining for naive pluripotency-associated markers. Naive-hiPS cells were then collected using Accutase (Stem Cell Technologies) and reseeded in naive condition, the medium was then switched to Primed hiPSC medium (E8) the following day. When the culture became confluent, cells were collected using Collagenase IV (ThermoFisher) and maintained in Primed hiPSC medium (E8). Cells were cultured in 37 °C, 5% $O_2$ and 5% $CO_2$. All cell lines were tested by CGH array and reported normal.

## Estimations of cell diversity by Cas9 enrichment for lentivirus insertion mapping

To prepare enriched Oxford Nanopore Technologies (ONT) sequencing libraries, we used PoreChop to design 2 guide RNAs (gRNAs) (5′-AGATCC GTTCACTAATCGAATGG-3′ and 5′-GGAACAGTACGAACGCGCCGAGG-3′) for Cas9-mediated cleavage approximately 1 kb within each end of the integrated lentiviral sequences. These gRNAs were designed to not match elsewhere in the hg38 human reference genome. We confirmed their on-target efficiency by Cas9 (IDT: Alt-R S.p. Cas9 Nuclease V3; catalogue no. 1081058) cleavage of the lentiviral DNA, visualized on gel, in a separate experiment. DNA dephosphorylation (NEB: Quick CIP; M0525S), single guide (IDT: Alt-R CRISPR–Cas9 CRISPR RNA (crRNA) and Alt-R CRISPR–Cas9 trans-activating crRNA (tracrRNA); catalogue no. 1072532) and RNP formation, Cas9 cleavage and subsequent library preparation (ONT: SQK-CS9109) were largely performed according to the ONT Cas9 enrichment guidelines. We increased the starting amount of DNA to 5 μg, and the dephosphorylation and cleavage incubation times to 2 h and 24 h, respectively. For two replicates of each reprogramming method, we then loaded 350 ng of the enriched DNA library onto a MinION R9.4 flow cell, as per the manufacturer's recommendations, and sequenced for 48 h. Additionally, for the 32F fibroblast sample, 3 μg of unenriched DNA was sequenced on a PromethION R9.4 flow cell (library prep kit SQK-LSK110) by the Kinghorn Centre for Clinical Genomics (KCCG). For data analysis, reads with a Phred score ≥10 were basecalled with Guppy (version 5.0.11). These reads were mapped with minimap2 (version 2.17) to both the human reference genome (hg38), and the sequence of the expected lentiviral insert[49]. Alignment maps were filtered with samtools (version 1.13) to only keep primary alignments with a length ≥800 bp, and a mapping quality[50] of 60. Reads that mapped to both hg38 and the lentivirus sequence were retained and then subjected to another round of filtering. Here, reads were discarded when the base pair interval between the alignments to the lentiviral sequence and hg38 on the read was ≥51 bp. Reads that originated from the unenriched library and comprised a complete (≥4,500 bp) putative lentiviral insert, spanned by a genomic alignment, as identified by TLDR (version 1.2.2) were kept[51]. Exact insert sites per read were identified based on the coordinates of both alignment maps (hg38 and lentiviral) to the original read. Exact insert sites were clustered together with bedtools (version 2.30.0) cluster within a 50-bp interval[52]. For each cluster, the coverage was calculated and the smallest start and largest end coordinates were selected as the exact insert site.

The diversity of cell populations was estimated by a Poisson bootstrap[53]. Here, we model a Poisson distribution of total insertion landscape based on the sequencing coverage of unique lentiviral insert sites. This model infers the amount of non-sequenced insertion sites, which in return is used to adapt the model until convergence, and results in an estimate for the lentiviral insertion diversity.

## Secondary fibroblast reprogramming system

hES cells were cultured in fibroblast medium without FGF2 containing DMEM, 10% FBS, 1 mM L-glutamine, 100 µM MEM non-essential amino acids, and 0.1 mM β-mercaptoethanol, for a week. Cells were passaged three times using 0.25% trypsin and then sorted for THY1+TRA160− populations.

## Neural stem cell differentiations

hiPS cells were cultivated in E8 medium (Life Technologies) on Cultrex (R&D Systems) coated TC dishes and split 1:10 every 5 days. Colonies were mechanically disaggregated with 0.5 mM EDTA in PBS (Sigma). After splitting, pieces of colonies were collected by sedimentation and resuspended in E8 medium with 10 µM ROCK inhibitor (Selleckchem) and cultured in petri dishes to form embryoid bodies in suspension. After 24 h, the medium was changed to Knockout DMEM (Life Technologies) with 20% Knockout Serum Replacement (Life Technologies), 1 mM β-mercaptoethanol (Sigma), 1% non-essential amino acids (NEAA, Life Technologies), 1% penicillin/streptomycin (Life Technologies) and 1% Glutamax (Life Technologies) supplemented with 10 µM SB-431542 (Selleckchem), 1 µM dorsomorphin (Selleckchem) for neural induction, as well as 3 µM CHIR99021 (Cayman Chemical) and 0.5 µM PMA (Sigma). Medium was replaced on day 3 by N2B27 medium (50% DMEM-F12 (Life Technologies), 50% Neurobasal (Life Technologies) with 1:200 N2 supplement (R&D Systems), 1:100 B27 supplement lacking vitamin A (Miltenyi Biotec) with 1% penicillin-streptomycin (Life Technologies) and 1% Glutamax (Life Technologies)) supplemented with the same small molecule supplements. On day 4, SB-431542 and dorsomorphin were withdrawn and 150 µM ascorbic acid (Sigma) was added to the medium. On day 6, the embryoid bodies were triturated with a 1,000 µl pipette into smaller pieces and plated on Cultrex-coated 12-well plates at a density of about 10–15 per well in NSC expansion medium (N2B27 with CHIR, PMA, and ascorbic acid). After another 5 days, cells were split at a ratio of 1:5 using Trypsin-EDTA (Life Technologies) and Trypsin inhibitor (Sigma) onto a new Cultrex-coated well. After another 5 days, cells were collected by 10 min trypsinization at 37 °C to generate a single-cell suspension for scRNA-seq workflow.

## Endoderm progenitor differentiation

The endoderm differentiation was adapted and performed as previously described[54,55]. In brief, hiPS cells were collected and replated onto plates coated with Matrigel and cultured in primed hiPS cell medium (KSR/FGF2) with medium change for an additional day before differentiation. To differentiate into endodermal progenitor cells, the cells were cultured in chemically defined medium containing 100 ng ml[−1] activin A, 20 ng ml[−1] FGF2, 10 ng ml[−1] bone morphogenetic factor 4 (BMP4), and 10 µM LY294002 for 3–4 days and assessed for differentiation efficiency.

## Cortical neuron differentiation

hiPS cells were seeded onto flasks coated with Matrigel at a density of 0.5–1 × 10[4] cells per cm[2] in primed hiPS cell medium (KSR/FGF2). After 48 h, the medium was changed to neural induction medium containing DMEM/F12, B27 without vitamin A supplement (Gibco, ThermoFisher Scientific), N2 supplement (Gibco, ThermoFisher Scientific), 0.1% β-mercaptoethanol (Gibco, ThermoFisher Scientific), 0.66% bovine serum albumin (Sigma-Aldrich), 1% sodium pyruvate (Gibco, ThermoFisher Scientific), 1% non-essential amino acids (Gibco, ThermoFisher Scientific), 1% penicillin and streptomycin, 100 ng ml[−1] LDN193189 (Tocris Bioscience, Bio-Techne) for 14 days.

## Skeletal muscle cell differentiation

hiPS cells were seeded onto flasks coated with Matrigel at a density of 0.5–1 × 10[4] cells per cm[2] in primed hiPS cell medium (KSR/FGF2). After 24 h, medium was changed to DMEM/F12-based medium supplemented with ITS (insulin + transferrin + selenium; Sigma-Aldrich) with 1% penicillin and streptomycin (Gibco, ThermoFisher Scientific), 3 µM CHIR99021 (Miltenyi Biotec), 0.5 µM LDN193189 (Tocris Bioscience, Bio-Techne) for 3 days. On days 4–6, the medium was changed to DMEM/F12-based medium supplemented with ITS and 3 µM CHIR99021, 20 ng ml[−1] FGF2 (Miltenyi Biotec), 0.5 µM LDN193189. On days 7–8, the medium was changed to DMEM/F12-based medium supplemented with 20 ng ml[−1] FGF2, 0.5 µM LDN193189, 2 ng ml[−1] IGF1 (Peprotech). On days 9–30, the medium was changed to DMEM/F12-based medium supplemented with 15% knockout serum replacement (Gibco, ThermoFisher Scientific), 1% penicillin and streptomycin, 0.05 mg ml[−1] BSA (Sigma-Aldrich), 2 ng ml[−1] IGF1.

## Lung alveolar type 2 cell differentiation

Induced pluripotent stem cells were seeded onto flasks coated with Matrigel at a density of 0.5–1 × 10[4] cells per cm[2] in primed hiPS cell medium (KSR/FGF2). After 48 h, the medium was changed daily with RPMI-based medium with B27 supplement (Gibco, ThermoFisher Scientific), 100 ng ml[−1] activin A (Peprotech), 1 µM CHIR99021, 1% penicillin and streptomycin for 3 days. On days 4–8, the medium was changed daily with DMEM/F12-based medium with N2 (Gibco, ThermoFisher Scientific) and B27 supplements, 0.05 mg ml[−1] ascorbic acid (Sigma-Aldrich), 0.4 mM monothioglycerol (Sigma-Aldrich), 2 µM dorsomorphin (Peprotech), 10 µM SB-431542 (Miltenyi Biotec), 1% penicillin and streptomycin. On days 9–12, the medium was changed daily with DMEM/F12-based medium with B27 supplement, 0.05 mg ml[−1] ascorbic acid, 0.4 mM monothioglycerol, 20 ng ml[−1] BMP4 (Peprotech), 0.5 µM all-*trans* retinoic acid (Sigma-Aldrich), 3 µM CHIR99021, 1% penicillin and streptomycin. On days 12–20, the medium was changed every other day with DMEM/F12-based medium with B27 supplement, 0.05 mg ml[−1] ascorbic acid, 0.4 mM monothioglycerol, 10 ng ml[−1] FGF10 (Stemcell Technologies), 10 ng ml[−1] FGF7 (Peprotech), 3 µM CHIR99021, 50 nM dexamethasone (Sigma-Aldrich), 0.1 mM 8-bromoadenosine 3′,5′-cyclic monophosphate (Sigma-Aldrich), 0.1 mM 3-isobutyl-1-methylxanthine (Sigma-Aldrich), 1% penicillin and streptomycin.

## Flow cytometry

To obtain a single-cell suspension for flow cytometric analysis or sorting experiments, cells were collected using TrypLE express (Life Technologies) and resuspended in labelling mix (PBS, 2% FBS, 10 µM ROCK inhibitor Y-27632). Reprogramming intermediates and mature hiPS cells were labelled in a stepwise manner for cell surface markers. Step 1: F11R (mouse IgG antibody; 1:150), SSEA3-PE (rat IgM antibody; 1:10, BD Biosciences); step 2: Alexa Fluor 647 goat anti-mouse IgG (1:2,000, ThermoFisher), PE anti-rat IgM (1:200 eBioscience); step 3: CD13-PE-Cy7 (1:400, BD Biosciences), BV421-EpCAM (1:100, BD), TRA-1-60-BUV395 (1:100, BD Biosciences). Cells were incubated for 10 min on ice and then washed with PBS and resuspended in FACS buffer (PBS, 2% FBS, 10 µM Y-27632 and PI (1 in 500)). Prior to sorting, cells were passed through a 35-µm nylon filter. Sorted cells were collected for replating or downstream analyses. For differentiation experiments, cultures were dissociated using Accutase (Stemcell Technologies) and pelleted at 400g for 5 min. For neural differentiation experiments, cells were then resuspended in APC CD57 antibody (322314; Biolegend) and BUV395 CD56 antibody (563554; BD Biosciences); for muscle differentiation experiments, cells were resuspended in PE-Cy7 CD146 antibody (562135; BD Biosciences), BUV395 CD56 antibody (563554; BD Biosciences); for lung differentiation experiments,

cells were resuspended in BV421 CD47 antibody (323116; Biolegend) and Brilliant Violet 421 CD326 antibody (324220; Biolegend); for NSC differentiation experiments, cells were labelled with BUV395 CD56 (NCAM) antibody and Alexa647 FAP antibody (FAB3715R; R&D Systems). Cells were resuspended in 2% fetal bovine serum (FBS; Gibco, ThermoFisher Scientific) and PBS (Gibco, ThermoFisher Scientific) and incubated for 15 min at 4 °C. The cell suspension was washed with PBS and pelleted at 400*g* for 5 min for analysis. Viability of cells was determined using propidium iodide solution (P4864; Sigma-Aldrich). Samples were analysed using an LSR IIb analyser (BD Biosciences) or a FACSAria II cell sorter (BD Biosciences) using BD FACSDiva software (BD Biosciences).

## Immunostaining
Cells were fixed in 4% PFA (Sigma), permeabilized with 0.5% Triton X-100 (Sigma) in DPBS (ThermoFisher), and blocked with 5% goat serum (ThermoFisher). All antibodies used in this study are detailed in Supplementary Table 9 (for example, primary antibodies used were rabbit anti-NANOG polyclonal (1:100, Abcam) and mouse anti-TRA-1-60 IgM (1:300, BD Biosciences)). Primary antibody incubation was conducted overnight at 4 °C on shakers followed by incubation with secondary antibodies (1:400) for 1 h. After labelling, cells were stained with 4′,6-diamidino-2-phenylindole, dihydrochloride (DAPI) (1:1,000, ThermoFisher) for 30 min. Images were taken using an IX71 inverted fluorescent microscope (Olympus). The following markers were assessed for respective differentiation assays: SOX17 and FOXA2 for endoderm progenitor differentiation experiments; SOX1 and PAX6 for neural differentiation experiments; PAX3 and PAX7 for skeletal muscle differentiation experiments; GATA6 and TTF1 for lung differentiation experiments.

## Quantitative PCR with reverse transcription
RNA was extracted from cells using RNeasy micro kit (Qiagen) or RNeasy mini kit (Qiagen) and QIAcube (Qiagen) according to the manufacturer's instructions. Reverse transcription was then performed using Quanti-Tect reverse transcription kit (Qiagen). Real-time PCR reactions were set up in duplicate using QuantiFast SYBR Green PCR Kit (Qiagen) and then carried out on the 7500 Real-Time PCR system (ThermoFisher) using LightCycler 480 software. The *GAPDH* gene was used to calculate the relative expression of each assessed gene. Information regarding the PCR primers used in this study is available in Supplementary Table 9.

## WGBS library preparation
Genomic DNA was isolated with the Qiagen Blood and Tissue Kit according to the manufacturer's instructions. 0.5% (w/w) of unmethylated lambda phage DNA (Promega) was added to the sample genomic DNA for the purpose of an unmethylated control to measure the bisulfite non-conversion frequency in each sample. Genomic DNA was fragmented with either either a Covaris S2 sonicator or a Covaris M220 sonicator to a mean length of 200 bp, then end-repaired, A-tailed, ligated to methylated Nextflex Bisulfite-Seq barcodes (Perkin Elmer) using the NxSeq AmpFREE low DNA library kit (Gene Target Solutions) and subjected to PCR amplification with KAPA HiFi Uracil+ DNA polymerase (KAPA Biosystems)[56]. Sequencing was performed single-end on a HiSeq 1500, NextSeq 500, or paired-end on a NovaSeq 6000 (Illumina).

## polyA RNA-seq
RNA was extracted using the Agencourt RNAdvance Cell v2 (Beckman Coulter) system following the manufacturer's instruction with one additional DNAse (NEB) treatment step. RNA amounts and RINe scores were assessed on a TapeStation using RNA Screen Tape (Agilent), and 500 ng of total RNA were used per sample to generate RNA-seq libraries. ERCC ExFold RNA Spike-In mixes (Thermo Scientific) were added as internal control. Libraries were prepared using the TruSeq Stranded mRNA library prep kit (Illumina), using TruSeq RNA unique dual index adapters (Illumina).

Libraries were quantified by qPCR on a CFX96/C1000 cycler (Bio-Rad) and sequenced on a NovaSeq 6000 (Illumina) in 2× 53-bp paired-end format.

## ATAC–seq
Approximately $10^6$ freshly collected cells were pelleted and washed in PBS, then resuspended in 1 ml of RSB buffer (10 mM Tris-HCl, 10 mM NaCl, 3 mM MgCl$_2$, 0.1% NP-40, 0.1% Tween-20, 0.01% Digitonin). After 10 min incubation on ice, samples were spun at 500*g* for 5 min and resuspended in 500 µl RSB without NP-40 or digitonin, then strained through a 30-µm filter and pelleted again. Resulting nuclei were counted using trypan blue and 50,000 nuclei were resuspended in 25 µl of 2× TD buffer (20 mM Tris-HCl, 10 mM MgCl$_2$, 20% dimethyl formamide). Tagmentation mix was completed by adding 100 U of loaded Tn5, 16.5 µl PBS, 0.5 µl of 1% digitonin and 0.5 µl of Tween-20 to a final volume of 50 µl, followed by incubation for 30 min at 37 °C with 1,000 rpm mixing on a thermo block. After tagmentation, samples were cleaned up using the Qiagen MinElute PCR purification kit. Eluate was amplified using NEBNext 2× MasterMix and Nextera-based adapters as primers. After 10 PCR cycles, a double-sided bead purification was performed using 0.5× and 1.8× Ampure XP beads. Libraries were quantified by qPCR on a CFX96/C1000 cycler (Bio-Rad) and sequenced on a NovaSeq 6000 (Illumina) in 2× 61-bp paired-end format.

## H3K9me3 ChIP–seq
Cells were crosslinked for 10 min in 1% formaldehyde and quenched in 125 mM glycine. Prior to ChIP, antibodies were bound to beads by mixing 3 µg H3K9me3 antibody (Abcam, ab8898) with 50 µl washed Dynabead M-280 Sheep Anti-Rabbit IgG (ThermoFisher) in 500 µl RIPA-150 buffer (50 mM Tris-HCl pH 8.0, 0.15 M NaCl, 1 mM EDTA, 0.1% SDS, 1% Triton X-100 and 0.1% sodium deoxycholate) and incubated at 4 °C for 6 h on a rotator. Crosslinked cells were lysed on ice for 10 min in 15 ml ChIP lysis buffer (50 mM HEPES pH 7.9, 140 mM NaCl, 1 mM EDTA, 10% glycerol, 0.5% NP-40, 0.25% Triton X-100) supplemented with 1x EDTA-free Protease Inhibitor Cocktail (Roche). Lysed cells were centrifuged at 3,200*g* for 5 min, supernatant removed and followed by two washes with 10ml ChIP wash buffer (10 mM Tris-Cl pH 8.0, 200 mM NaCl and 1 mM EDTA pH 8.0). Lysed cells were resuspended in 130 µl nuclei lysis buffer (50 mM Tris-HCl pH 8.0, 10 mM EDTA and 1% SDS) supplemented with 1× EDTA-free Protease Inhibitor Cocktail (Roche), transferred to Covaris tubes (microTUBE AFA Fiber 6 × 16 mm) and sheared with the Covaris (S220) for 5 min (5% duty cycle, 200 cycles per burst and 140 watts peak output at 4 °C). Sheared chromatin was transferred to 1.5 ml eppendorf tubes, centrifuged at 10,000*g* for 10 min. The supernatant was transferred to 2 ml low-bind tubes containing 1.2 ml ChIP dilution Buffer (50 mM Tris-HCl pH 8.0, 0.167 M NaCl, 1.1% Triton X-100 and 0.11% sodium deoxycholate) and 0.65 ml RIPA-150 buffer, and incubated with the previously prepared H3K9me3 antibody bound Dynabeads at 4 °C overnight on a rotator. Chromatin bound beads were subsequently washed one time with 1 ml RIPA-150 buffer, two times with 1 ml RIPA-500 buffer (50 mM Tris-HCl pH 8.0, 0.5 M NaCl, 1 mM EDTA, 0.1% SDS, 1% Triton X-100 and 0.1% sodium deoxycholate), two times with 1ml RIPA-LiCl buffer (50 mM Tris-HCl pH 8.0, 1 mM EDTA, 1% NP-40, 0.7% sodium deoxycholate and 0.5 M LiCl$_2$) and two times with TE buffer (10 mM Tris-HCl, pH 8.0, 0.1 mM EDTA). After wash steps, DNA was eluted, crosslinks were reversed, and immunoprecipitated DNA was purified by Agencourt AMPure XP beads (Beckman Coulter, A63880). Libraries were prepared from ChIP eluate containing 10 ng DNA using the SMARTer ThruPLEX DNA-Seq Kit (Takara) with SMARTer DNA unique dual index (Takara). After limited PCR amplification, libraries were purified using Agencourt AMPure XP beads (Beckman Coulter), and eluted in a final volume of 20 µl. Libraries were sequenced on a NovaSeq 6000 (Illumina).

## scRNA-seq
Single-cell suspensions were counted using a haemocytometer and 200,000 cells per sample used for incubation with hashtag antibodies.

Cells were filtered through a 40 μm cell strainer, centrifuged at 800*g* for 5 min and resuspended in a total volume of 46 μl cell staining buffer (2% BSA (Sigma), 0.01% Tween (Sigma) in 1× DPBS (Life Technologies)) with 4 μl of Fc blocking reagent (Biolegend) and incubated for 10 min on ice. Then, each sample received 0.2 μg of a different TotalSeq-A anti-human Hashtag antibody (Biolegend) and was incubated for 30 min on ice for antibody binding. After the incubation, 1 ml of cell staining buffer was added, and sample centrifuged at 300*g* for 3 min. Supernatant was removed and cells washed again for a total of three washes to remove all unbound antibodies. Cells were counted, and equal cell numbers for each sample combined to get a cell concentration suitable for loading on the 10x Chromium controller aiming to get 10,000 cells represented. The mixed cell suspension was filtered one more time using a 40-μm cell strainer and processed for scRNA-seq using the 10x Genomics 3′ v3 chemistry following the manufacturer's instructions. Libraries for scRNA-seq were made following the standard workflow, while HTO libraries for hashtag information were generated as follows: during the cDNA amplification step, HTO primers were added to allow amplification of the HTO barcodes, and supernatant from the first step of clean-up after cDNA amplification PCR was not discarded but used to prepare the HTO library. HTO products were purified using 2x SPRI beads and amplified for 8 PCR cycles with 10× SI-PCR oligo and TruSeq Small RNA RPIx primers to generate a library of ~180 bp fragment size. Sequencing was performed on a NovaSeq 6000 to generate ~420 million reads for the scRNA-seq library and ~40 million reads for the HTO library.

## WGBS methylation analysis

Sequencing adapters were trimmed with BBduk with the options mink = 3, qtrim = r, trimq = 10 minlength = 20 before alignment to hg19 with Bowtie and BSseeker2 with the option -n 1[57,58]. PCR duplicates were removed using Sambamba[59] and DNA methylation levels at base resolution calculated using CGmap tools[60]. The non-conversion rate was calculated using the DNA methylation levels for the spiked-in lambda phage genome. When DNA methylation levels were calculated for regions such as promoters, enhancers, DMRs or ICRs, DNA methylation levels were calculated as a coverage-weighted mean by summing the number of methylated C calls (mC) and dividing that by the total number of reads with either a C or T call (C), for the CG or CA dinucleotide contexts separately (defined as mCG/CG and mCA/CA, respectively). To calculate methylation in CH contexts (where H is A, T or C), the level of methylation was calculated as above (mCH/CH) with the non-conversion rate subtracted from this value. When CH methylation was calculated for individual contexts, for example CA methylation, the non-conversion rate for that context was subtracted from the calculated methylation levels. For CA methylation browser tracks, mCA/CA was calculated for 5 kb sliding windows (1-kb slide), with the CA methylation non-conversion rate for that library subtracted from each window. To calculate per-read methylation, reads classified as methylated had methylation calls at every CG position in the read; unmethylated reads had zero methylation calls at CG positions; partially methylated reads had at least one CG methylation call and one non-methylated CG call.

## DMR analyses

To test for differentially methylated regions between hiPS cells and hES cells, we first collapsed the stranded mCG values to obtain one value for the symmetrical CG dinucleotides and then performed DMR testing using DMRseq with the options bpSpan = 500, maxGap = 500, maxPerms = 10 and subsequently filtered for DMRs[61] with mCG/CG difference >0.2 and *P* value < 0.05. For CH-DMR analyses, we used the CH-DMRs as previously defined[13]. We took each CH-DMR and equivalent upstream and downstream genomic regions and divided them into 30 equal-length bins and calculated mCA/CA for each bin and then flank-normalized the binned mCA/CA values by dividing them by their maximum value.

## Quantification of gene and transposable element expression

PolyA RNA-seq (Fig. 4 and Extended Data Fig. 10): adapters were trimmed using fastp with default parameters[62], and mapped to hg19 using HISAT2 with the options --no-mixed --dta --rna-strandness RF -k 2[63]. Alignments were then filtered to keep only unique mapping read pairs using Samtools view -F "[NH]==1"[50]. Gene and transposable element read counts were calculated using TEtranscripts and the TElocal script and the curated TE GTF files for hg19 that accompany this software[64]. Differential expression testing was performed using the glmLRT function within edgeR and genes were determined as significant if $\log_2$FC was <1, FDR <0.05 and average log counts per million for the gene was >1. When testing for differential expression of individual transposable elements, we obtained a matrix that contained counts for all genes and individual transposable elements, then filtered this for low or not expressed elements using the filterByExpr function and then calculated the normalization factors for the count matrix. We then performed differential expression testing on this matrix using the glmLRT function to obtain fold-change and significance values. As we were not testing for differential expression of genes, but wanted to retain their counts for library normalization, we then filtered the fold-change and significance table to only include the transposable elements, and then recalculated the FDR for transposable elements only. Significant transposable elements were then classed as differentially expressed if $\log_2$FC was <1, FDR <0.05 and average $\log_2$ counts per million for the transposable element was >0.

## ATAC–seq analysis

Sequencing adapters were trimmed with BBduk with the options mink = 3, ktrim = r, before alignment to hg19 with Bowtie2 with the option -X 2000. Reads were filtered for proper pairs, and PCR duplicates and mitochondrial reads removed using SAMtools. Bigwig browser tracks were normalized for library size using the counts per million method at single base resolution. ATAC–seq peaks were called with MACS2 with the options --nomodel --keep-dup all --gsize hs. Reads counts in peaks for each library were calculated using the summarizeOverlaps function in the GenomicAlignments R package. Differential peak analyses were performed using EdgeR with the glmQLFit glmQLFTest functions. ATAC–seq peaks were considered differentially expressed if the FDR was <0.05, the average log counts per million was >1, and the absolute $\log_2$FC was >2. Although we observed differences in ATAC–seq peak counts for NTP-hiPS cells that were not consistent with DNA methylation or gene expression for two outlier samples (Fig. 4b–d), we believe this is due to an additional freeze-thaw cycle for the ATAC–seq samples, and the extended recovery of these two replicates which required two additional passages.

## H3K9me3 ChIP–seq analysis

Adapters were trimmed using fastp with default parameters[62], and mapped to hg19 using bowtie2 with the option -X 2000. H3K9me3 fold enrichment was calculated for each ChIP and associated input library using the MACS2 bdgcmp function with the option -FE. H3K9me3 fold-enrichment values and peaks for primary fibroblasts and hES cells were downloaded from the ENCODE database for the following accessions: ENCFF735TXC (fibroblast H3K9me3 fold enrichment bigwig file); ENCFF963GBQ (fibroblast H3K9me3 peaks); ENCFF108MOZ (hES cell H3K9me3 fold enrichment bigwig); ENCFF001SUW (hES cell H3K9me3 peaks).

## Regulatory element principal component analysis, c-means clustering and motif enrichment analysis

DNA methylation levels were calculated for GeneHancer promoter and enhancer elements using the 'ClusteredInteractionsDoubleElite' elements[47] in the UCSC hg19 table browser. These regulatory elements include a linked gene and a confidence score for gene linkage.

For principal component analysis (PCA) and c-means clustering (Fig. 1d), we calculated the coverage-weighted mean methylation level (mCG/CG) for all the regulatory elements. Principal components were calculated using the R function pr. For Fig. 1e, c-means clustering was performed on regulatory elements that featured ≥20% mCG change at any time through primed reprogramming. Clusters were then identified for both the primed and naive reprogramming time courses with the functions included with the R package Mfuzz[65], highly overlapping clusters between the two time courses merged. To plot the expression of genes for each cluster, we first calculated the transcripts per million (TPM) for all genes and then quantile-normalized the gene-expression matrix. Each gene-expression measure was then weighted by enhancer interaction score (TPM × interaction score) to down-weight the expression of linked genes with low interaction scores as many elements were linked to more than one gene. The gene-expression plots in Fig. 1e shows the mean weighted and normalized gene-expression value and the 99% confidence interval. Gene ontology was performed on cluster genes using g:Profiler[66]. Enriched motifs for each cluster were identified using HOMER with findMotifsGenome.pl and the options hg19 -size given[67].

### Genomic feature enrichment analysis

To perform association analysis of genomic regions we performed permutation tests calculate enrichment of genomic elements with elements obtained from the GeneHancer database[47]; ultra-conserved elements as defined previously[68]; repeat elements as defined by UCSC repeat masker for hg19; fibroblast partially methylated domains calculated for day_0 fibroblasts with MethylSeeker[69]; promoters defined as 2 kb upstream and 500 bases downstream of TSS as defined in UCSC genes; Exons and introns as defined in UCSC genes; LADs for fibroblasts (4DNFIUIDLJJI) and H1 ES cells (4DNFIP6N54B3) as defined by 4D nucleome project for hg38 and lifted over to hg19 coordinates[70,71]. H3K9me3 peaks were retrieved from the ENCODE database for fibroblasts (ENCFF963GBQ) and hES cells (ENCFF001SUW)[72]. Constitutive regions for LADs or H3K9me3 were defined as those regions where peaks intersected for both fibroblasts and hES cells. In these enrichment analyses, the permutation tests calculate how many overlaps the features of interest (that is, CG-DMRs) have, for example, with fibroblast-specific H3K9me3 regions compared to randomly selected regions, and permuted 200 times. This approach addresses the problem of simply comparing the percentage of overlaps, as one does not know how many of those occur by chance. The z-scores from the permutation testing are a measure of the strength of the association, and is defined as the distance between the expected value and the observed one, measured in standard deviations. For example, a z-score of +25 would indicate that the number of overlaps is 25 standard deviations higher than one would expect by chance.

### Gene ontology

All gene ontology analyses were performed using g:Profiler using default options and the background set as all detectable genes in the dataset being tested[66].

### scRNA-seq analysis

RNA-seq fastq files were processed using CellRanger count 3.1.0, while HTO fastq files were processed using CITE-seq-Count 1.4.3 using parameters -cbf 1 -cbl 16 -umif 17 -umil 26 -cells 10000 and feeding sequences of oligonucleotide barcodes. RNA and HTO data were loaded into Seurat 3.1.1 and combined by intersecting cell barcodes found in both datasets. RNA data was log normalized, variable features detected by mean variance while HTO data was normalized by centred log-ratio transformation with margin = 1. Mitochondria were removed based on low UMI counts and enrichment for mitochondrial transcripts. HTODemux was used with positive.quantile = 0.99 to assign single cells back to their sample origins and to exclude doublets and negatives from further analysis. Top 1000 most variable features were used for

scaling and PCA of RNA data, using 10 dimensions with a resolution of 0.6 for clustering and UMAP. Cluster identities were defined based on the expression of markers for mesoderm (*BMP1*, *BMP4*, *HAND1*, *SNAI1*, *TGFB1* and *TGFB2*), endoderm (*AFP*, *ALB*, *CLDN6*, *FABP1*, *FOXA1* and *HNF4A*) and neural stem cells (*NCAM1*, *NES*, *NR2F1*, *PAX3*, *SOX1* and *SOX2*). No clusters expressing markers of pluripotency (*FUT4*, *KLF4*, *MYC*, *NANOG*, *POU5F1* and *ZFP42*) could be detected. By using the HTO identity for each singlet cell, the proportion of cell identities within each of the samples used could be defined.

### Statistics and reproducibility

The experiments on characterizing the cell lines derived in this study were not randomized. The investigators were not blinded to allocation during experiments and outcome assessment. All the experiments have been performed as at least two independent experiments as indicated in Methods or figure legends. The derivation of respective primed and TNT-iPS cells has been performed in four biological replicates (four cell types: primary HDFs, NHEK cells, MSCs and our hES cell-derived secondary fibroblast isogenic reprogramming system (secondary fibroblasts) as described in this Article) and was repeated in three independent reprogramming experiments. For the differentiation assays performed in Fig. 5 and Extended Data Fig. 10, a summary of the sample size can be found in Supplementary Table 10.

### Reporting summary

Further information on research design is available in the Nature Portfolio Reporting Summary linked to this article.

## Data availability

Raw and processed high-throughput sequencing datasets have been deposited at the NCBI Gene Expression Omnibus (GEO) repository under the SuperSeries accession number GSE159297; the dataset comprises WGBS, bulk RNA-seq, scRNA-seq, H3K9me3 ChIP–seq, ATAC–seq and nanopore sequencing data. Bulk RNA-seq data for human naive and primed reprogramming intermediates are available under GSE149694. Other publicly available data used in this study are available under GEO accessions GSE60945, GSE16256, GSE57179, GSE73211, GSE53096, GSM1003585, GSM1003553 and Sequence Read Archive (SRA) accession SRP003529. Lamin-B1 data are from the 4D nucleome project (https://www.4dnucleome.org/), with accessions 4DNFIUIDLJJI and 4DNFIP6N54B3. H3K9me3 peaks were retrieved from the ENCODE database for fibroblasts (ENCFF963GBQ) and hES cells (ENCFF001SUW). Genome browser for genomic data is available at http://tnt.listerlab.org. Source data are provided with this paper.

## Code availability

All data were analysed with commonly used open-source software programs and packages as detailed in Methods. The code is openly accessible at https://github.com/ListerLab/tnt.

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

**Acknowledgements** This work was supported by the following sources. R.L.: National Health and Medical Research Council (NHMRC) project grant 1069830, NHMRC investigator grant 1178460, Silvia and Charles Viertel Senior Medical Research Fellowship, Howard Hughes Medical Institute International Research Scholarship, Western Australia Department of Health Research Excellence Award and Australian Research Council (ARC) LE170100225. J. M. Polo: Silvia and Charles Viertel Senior Medical Research Fellowship, ARC Future Fellowship FT180100674, and NHMRC project grants 1069830 and 1104560. S.B.: NHMRC/ARC Dementia Research Development Fellowship 1111206 and Western Australia Department of Health Merit Award 1174766. X.L,: Monash International Postgraduate Research Scholarship, Monash Graduate Scholarship and the Carmela and Carmelo Ridolfo Prize in Stem Cell Research, Westlake Education Foundation. A.L.L.: NHMRC project grant 1104560. C.M.N.: Monash University strategic grant. O.J.L.R. and J.F.O.: Singapore National Research Foundation Competitive Research Programme NRF-CRP20-2017-0002. G.J.F.: NHMRC investigator grant 1173711 and the Mater Foundation. T.V.N. and D.B.V.-L.: Forrest Research Foundation PhD scholarships. The Australian Regenerative Medicine Institute is supported by grants from the State Government of Victoria and the Australian Government. The South Australian immunoGENomics Cancer Institute (SAiGENCI) received grant funding from the Australian Government. Genomics data was generated at the ACRF Centre for Advanced Cancer Genomics and Genomics WA. We thank staff at Monash Flowcore Facility and UWA Centre for Microscopy, Characterisation and Analysis for providing high-quality cell sorting services and technical input; S. Wang and T. Wilson for assistance with library preparation and Illumina sequencing; G. Neely and O. Bogdanovic for valuable feedback on this work; and T. Heng for providing MSC cell lines.

**Author contributions** R.L. and J. M. Polo conceived the study. S.B., X.L., D.P., J.P.T., J. M. Polo and R.L designed experiments and analyses. S.B. performed bioinformatics and data analyses with support from A.d.M., G.J.F., D.B.V-L, D.P., N.L. and J.B. X.L. and J.P.T. performed reprogramming experiments, collection and isolation of cells and intermediates, and functional validation experiments with support from G.S., C.M.N., J.C. and J. M. Paynter. D.P. performed neural stem cell differentiations, scRNA-seq, ATAC–seq, WGBS and polyA RNA-seq with assistance from T.F. T.V.N. and D.P performed ChIP–seq experiments. J.P. made WGBS libraries. N.S. and G.J.F. performed lentiviral quantification of cell diversity. D.P. and J.P. performed sequencing. J.F.O., F.J.R. and O.J.L.R. helped with single primer isothermal amplification RNA-seq analysis. H.S.C. and A.L.L. provided reagents and technical assistance. Independent replication for any given experiment was performed by the same individual leading the specific aspect of the project. S.B., X.L., D.P., J.P.T., J. M. Polo, and R.L. wrote the manuscript with input from A.d.M., J.F.O., O.J.L.R., J.P., C.M.N., A.L. and G.J.F. All authors approved of the final version of the manuscript.

**Competing interests** S.B., X.L., J. M. Polo and R.L. are co-inventors on a pending patent (PCT/AU2019/051296) filed by the University of Western Australia and Monash University related to this work. R.L. is a co-inventor on a patent (WO/2012/058634) concerning methods of characterizing the epigenetic signature of human induced pluripotent stem cells. Although unrelated to this manuscript, O.J.L.R. and J. M. Polo are co-inventors on a patent (WO/2017/106932) and are co-founders and shareholders of Mogrify, a cell therapy company. X.L. is a co-founder of iCamuno Biotherapeutics. The other authors declare no competing interests.

**Additional information**
**Correspondence and requests for materials** should be addressed to Jose M. Polo or Ryan Lister.

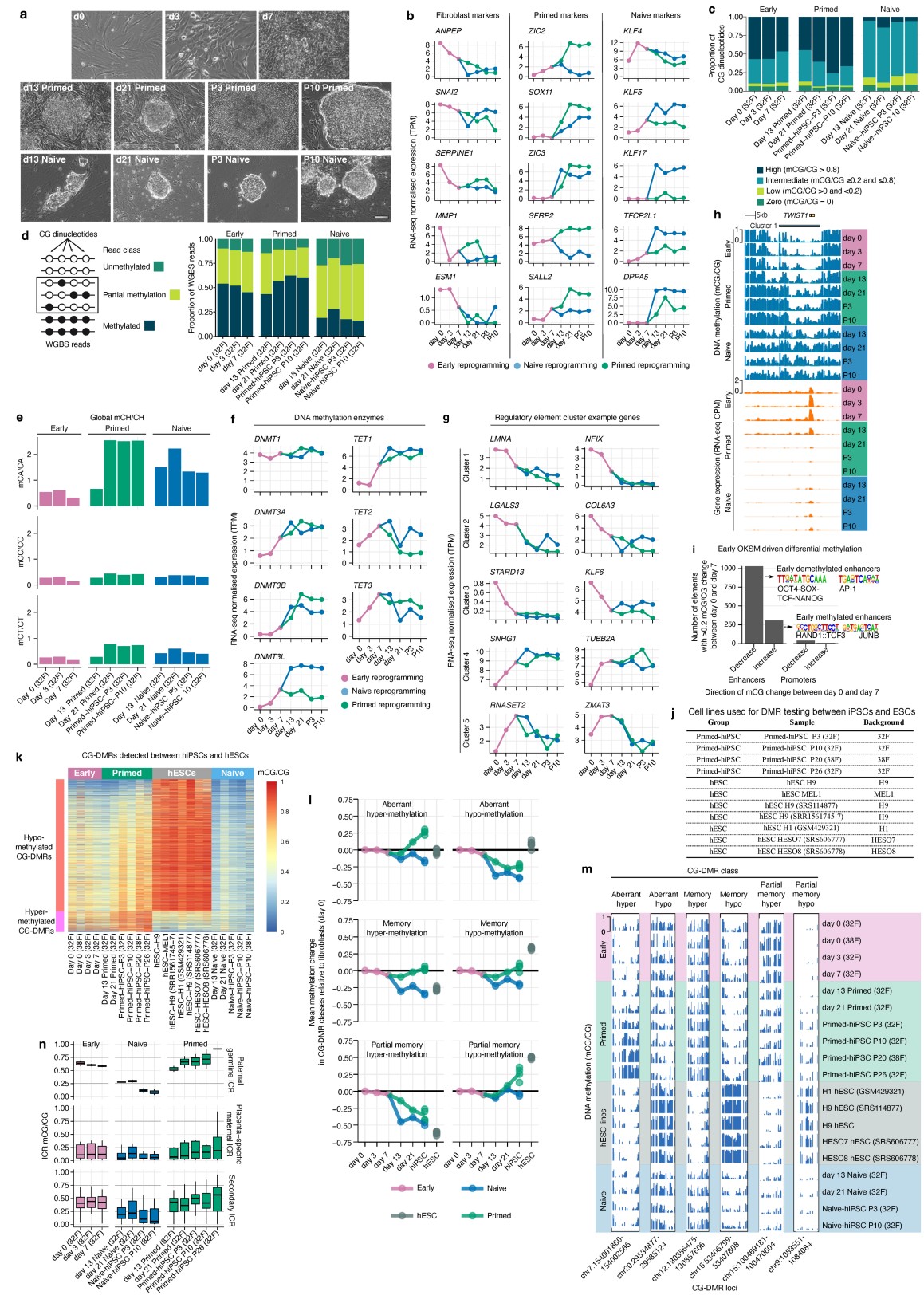

**Extended Data Fig. 1 | See next page for caption.**

**Extended Data Fig. 1 | Distinct trajectories of DNA methylation change throughout human naive and primed reprogramming. a)** Phase contrast images of reprogramming intermediates and hiPSCs throughout primed and naive reprogramming, n = 2 independent experiments. Scale bar: 100 μm. **b)** Gene expression profiling of marker genes for fibroblasts, Primed-hiPSCs, and Naive-hiPSCs throughout the time course of human reprogramming into both pluripotent states. **c)** Genome-wide proportion of CG dinucleotides in four categories of methylation levels: high, intermediate, low, and zero. **d)** Proportion of unmethylated, partially methylated, and fully methylated reads from WGBS libraries. **e)** Genome-wide levels of CH context DNA methylation (mCH/CH) for all dinucleotide contexts. **f)** Expression levels of genes encoding key enzymes in the cytosine DNA methylation (DNMTs) and demethylation (TETs) pathways. **g)** Regulatory element cluster gene examples from Fig. 1e where C-means fuzzy clustering of CG DNA methylation levels in GeneHancer regulatory elements was performed throughout primed and naive reprogramming. **h)** Genome track of CG DNA methylation levels and gene expression of a cluster 1 element (horizontal bar) encompassing the *TWIST1* gene. **i)** Number of enhancers and promoters that change DNA methylation level > 0.2 between day 0 and day 7 of reprogramming, before cells are cultured in primed or naive media. Motif enrichment analysis shows enhancers that undergo CG demethylation before day 7 are enriched for OKSM factors and AP1 motifs. Enhancers with increased CG methylation between day 0 and day 7 are enriched for HAND1/JUNB motifs. **j)** Cell lines used to test for CG context differentially methylated regions (DMRs) between Primed-hiPSCs and hESCs. Background column indicates genetic background identifier for the cell line. **k)** Heatmap representation of CG methylation levels in the CG-DMRs. **l)** Mean CG DNA methylation changes across hypo-methylated memory CG-DMRs and aberrant hyper-methylated CG DMRs relative to the progenitor fibroblast state (day 0). Each datapoint represents mean CG DNA methylation change compared to d0 for individual samples. **m)** Genome track showing CG methylation levels for examples of each of the six CG-DMR classes indicated in Fig 1b. **n)** CG methylation at imprint control regions (ICRs) for paternal germline ICRs and secondary ICRs. Boxplots: median and IQR, whiskers = 1.5 × IQR. n = 1 independent experiment per boxplot. ICRs defined in[21].

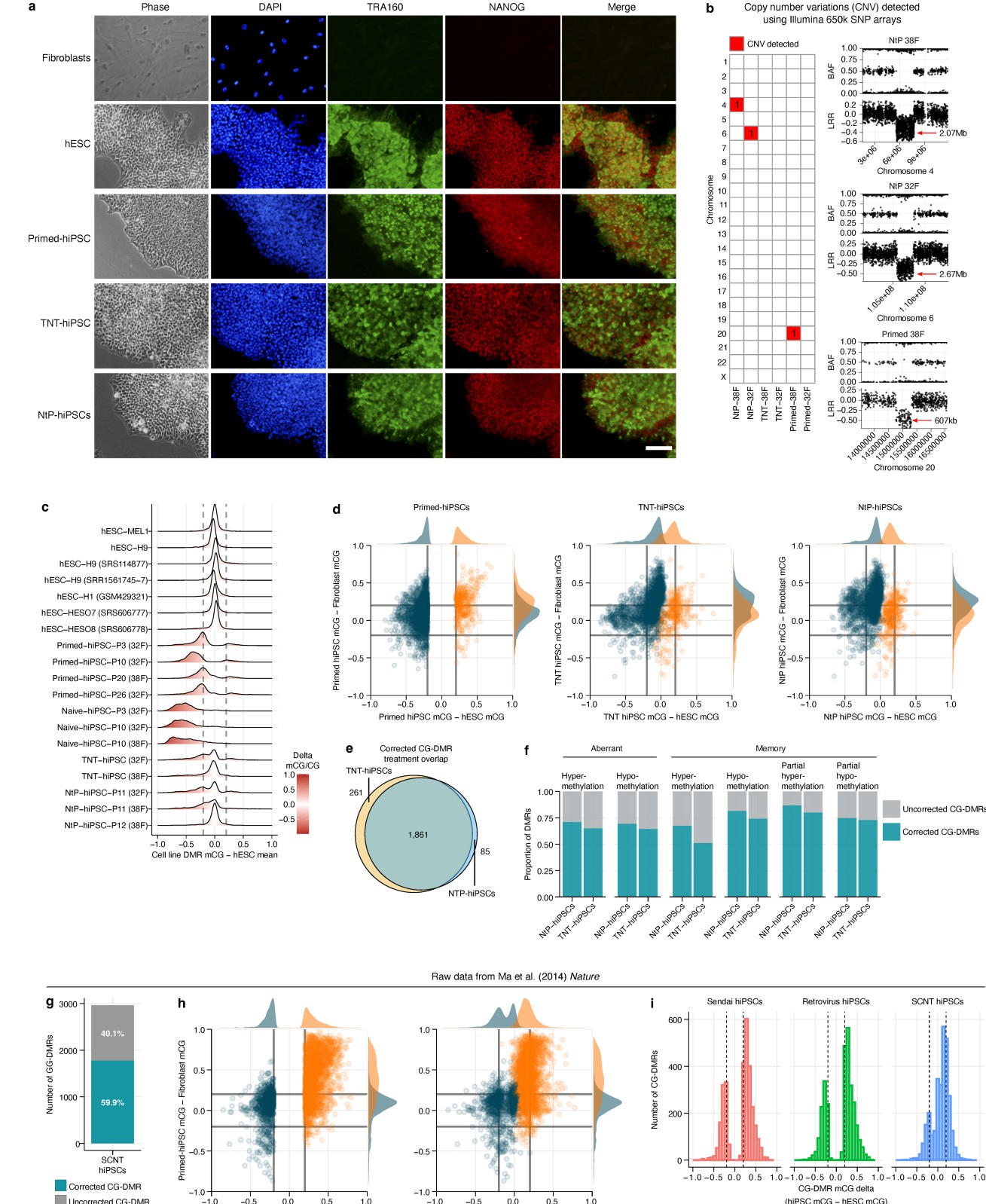

**Extended Data Fig. 2 |** See next page for caption.

**Extended Data Fig. 2 | Reprogramming through the naive state erases somatic cell memory and produces hiPSCs that closely resemble hESCs.**
**a)** Immunostaining of pluripotency markers NANOG and TRA160 for fibroblasts, hESC and different hiPSC lines, n = 2 independent experiments. Scale bar: 100 μm. **b)** Summary plot of copy number variation (CNV) analysis performed using Illumina 650k arrays. Left grid plot indicates the samples and chromosomes where CNVs were detected. Right plots show B allele frequency (BAF) and log R ratio (LRR) for samples where a CNV was detected, with each datapoint representing variant sites. **c)** Kernel density plots of DNA methylation difference in CG-DMRs for individual cell lines and replicates relative to the mean methylation of all hESC lines. **d)** Scatter plot of relative CG DNA methylation difference in CG-DMRs for Primed-hiPSCs, TNT-hiPSCs, and NtP-hiPSCs compared to primed hESC lines (x-axis) and progenitor fibroblasts (y-axis). Each CG-DMRs is represented by an individual point with the methylation values representing the average of all samples in that group. Blue points: hypo-methylated CG-DMRs. Orange points: hyper-methylated CG-DMRs. Dashed lines represent the 0.2 (i.e. 20%) methylation level difference used as a minimum threshold for differential DNA methylation. Kernel density estimate plots (top and right) show the distribution of CG-DMR methylation difference for hypo- and hyper-methylated CG-DMRs. **e)** Overlap of corrected CG-DMRs for TNT-hiPSCs and NtP-hiPSCs. **f)** Proportion of CG-DMRs that are corrected by NtP and TNT reprogramming for each category specified in Fig. 2b. **g)** Number of CG-DMRs corrected by SCNT reprogramming. Raw data are from Ma *et al*. (2014)[13] for (g-i). **h)** Scatter plot of relative CG DNA methylation difference in CG-DMRs for Primed-hiPSCs (left) and SCNT-iPSCs (right) compared to primed hESCs (x-axis) and fibroblasts (y-axis) as in (d). **i)** Histograms showing the difference in DNA methylation level at CG-DMRs for Primed-hiPSCs and SCNT-iPSCs. Vertical dashed lines indicate the 0.2 (i.e. 20%) methylation level difference used as the minimum threshold for differential DNA methylation.

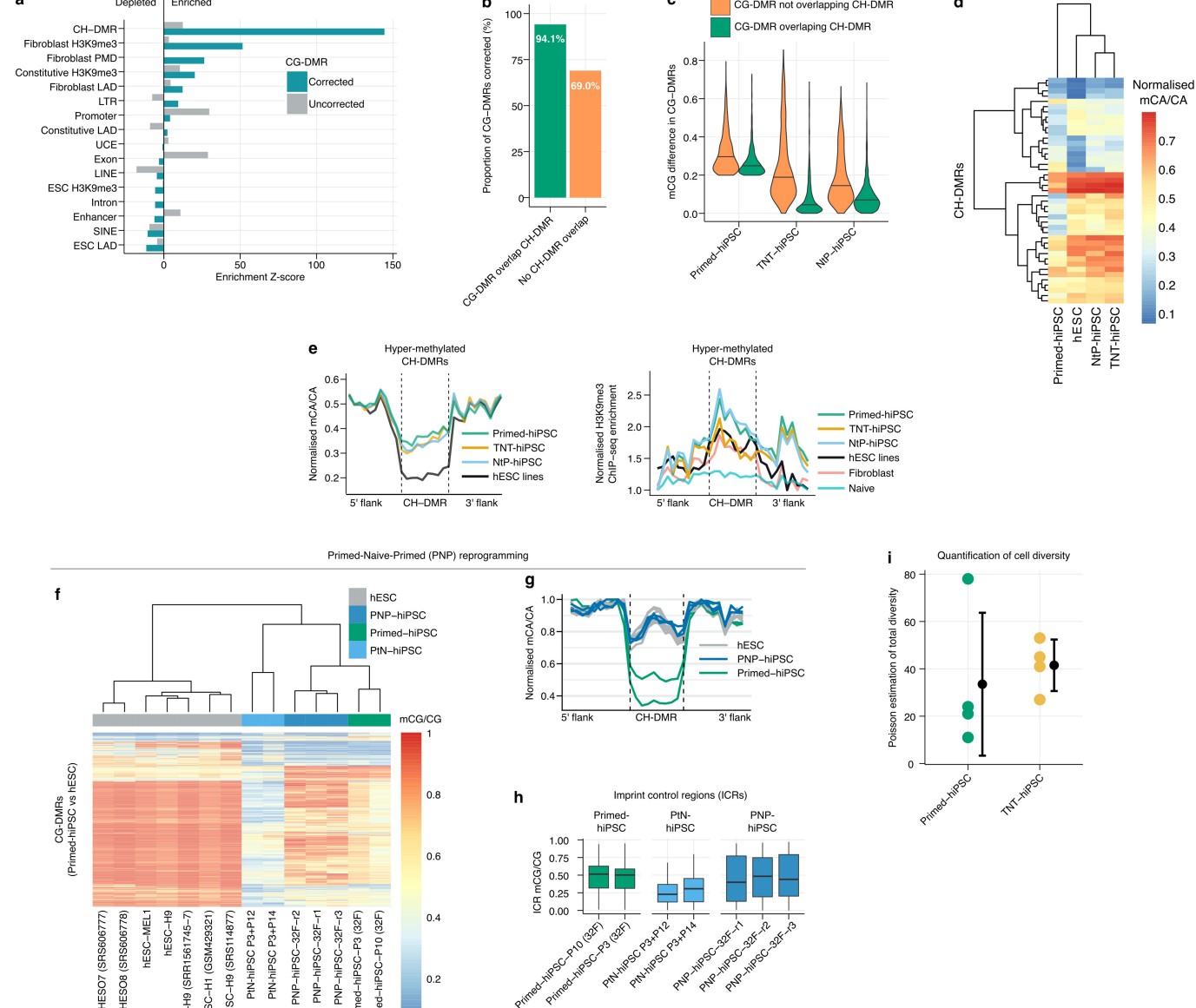

**Extended Data Fig. 3 | Reprogramming through the naive state erases somatic cell memory and produces hiPSCs that closely resemble hESCs.**
**a)** Enrichment z-scores determined from permutation testing of enrichment of genomic regions for corrected and uncorrected CG-DMRs. This is an expanded set of regions to those shown in Fig. 3e. **b)** Proportion of CG-DMRs corrected with respect to whether their genomic location overlaps with the larger CH-DMRs or not. **c)** Distribution of the difference in CG methylation between hESCs and hiPSCs at CG-DMRs that do or do not intersect CH-DMRs. **d)** Heatmap of normalised CA methylation levels in CH-DMRs. **e)** Left panel: aggregate profile plot of CA methylation levels in hyper-methylated CH-DMRs. Right panel: H3K9me3 enrichment in the same CH-DMRs. **f)** Heatmap representation of CG methylation levels in the CG-DMRs showing Primed-Naive-

Primed cells (PNP-hiPSCs) in the context of Primed-hiPSCs, Primed-to-Naive cells (PtN-hiPSCs), and hESCs. **g)** Aggregate profile plot of CA methylation levels in hypo-methylated CH-DMRs. **h)** CG methylation levels at maternal germline imprint control regions (ICRs). Boxplots: median and IQR, whiskers = 1.5 × IQR. n = 1 independent experiment per boxplot. **i)** Estimation of cell diversity after reprogramming fibroblasts by conventional Primed and TNT methods using lentivirus-mediated transduction of a sequence randomly integrated into the genome of primary adult fibroblasts, followed by reprogramming using either the Primed or TNT approach. Genomic DNA was subsequently isolated from the Primed- or TNT-hiPSCs and the locations of the lentivirus insertions in the genome mapped by nanopore sequencing. n = 4 independent reprogramming experiments per group, error bars show mean ±SD.

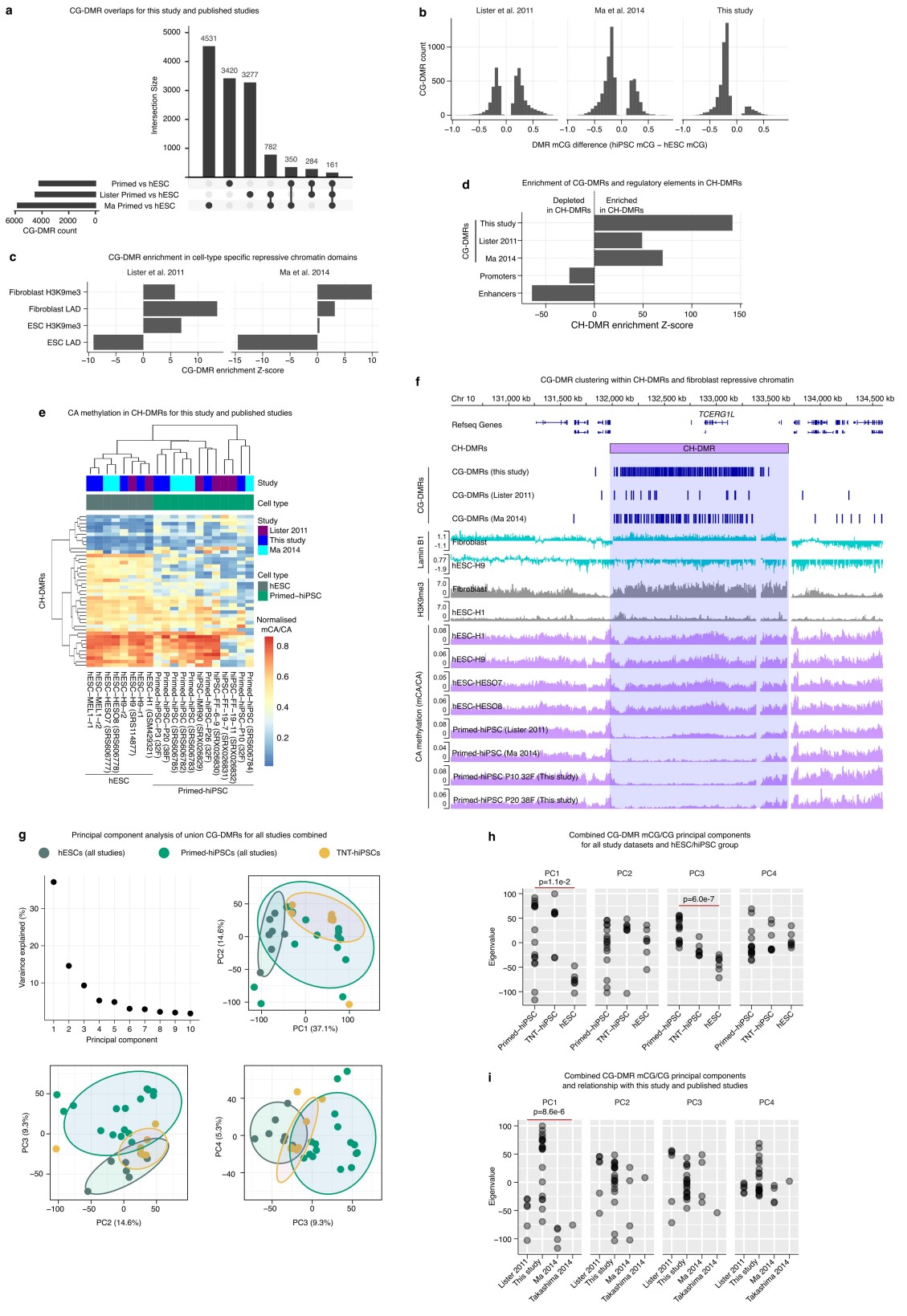

**Extended Data Fig. 4** | See next page for caption.

**Extended Data Fig. 4 | Comparison of CG and CH DMRs across studies.**
**a)** Upset plot shows number of CG-DMRs detected for this study and how they overlap with CG-DMRs detected from previously published data processed using identical methods. **b)** Difference in DNA methylation level between hiPSCs and hESCs at CG-DMRs identified between Primed-hiPSCs and hESCs. Vertical dashed lines indicate the threshold of 20% minimum difference in CG DNA methylation level at CG-DMRs. **c)** Enrichment z-score determined from permutation testing of enrichment of CG-DMRs in repressive chromatin domains and of **d)** CH-DMRs in published studies. **e)** Heatmap of CA methylation levels in CH-DMRs in this study and previously published studies showing Primed-hiPSCs from all studies clustering separately to hESCs. **f)** Genome track of a CH-DMR region that intersects a PMD, fibroblast lamina associated domain (LAD), and clusters of CG-DMRs in each study. **g)** Principal component analysis of CG methylation levels in CG-DMRs for all studies combined. Top left plot shows the proportion of variance explained by each principal component. Scatter plots with coloured points show principal component separation of hESCs, Primed-hiPSCs, and TNT-hiPSCs. Ellipses around points indicate 95% confidence interval for a multivariate t-distribution. These data indicate that principal component 3 (PC3) in the bottom left plot clearly separates Primed-hiPSCs and hESCs for all studies, and shows that TNT-hiPSCs are more similar to hESCs by this measure. **h)** Plots of eigenvalues for each principal component for Primed-hiPSCs, TNT-hiPSCs, and hESCs, and **i)** data split by study/lab. Red bars indicate $P < 0.05$ for one-way ANOVA, with FDR reported above red bars.

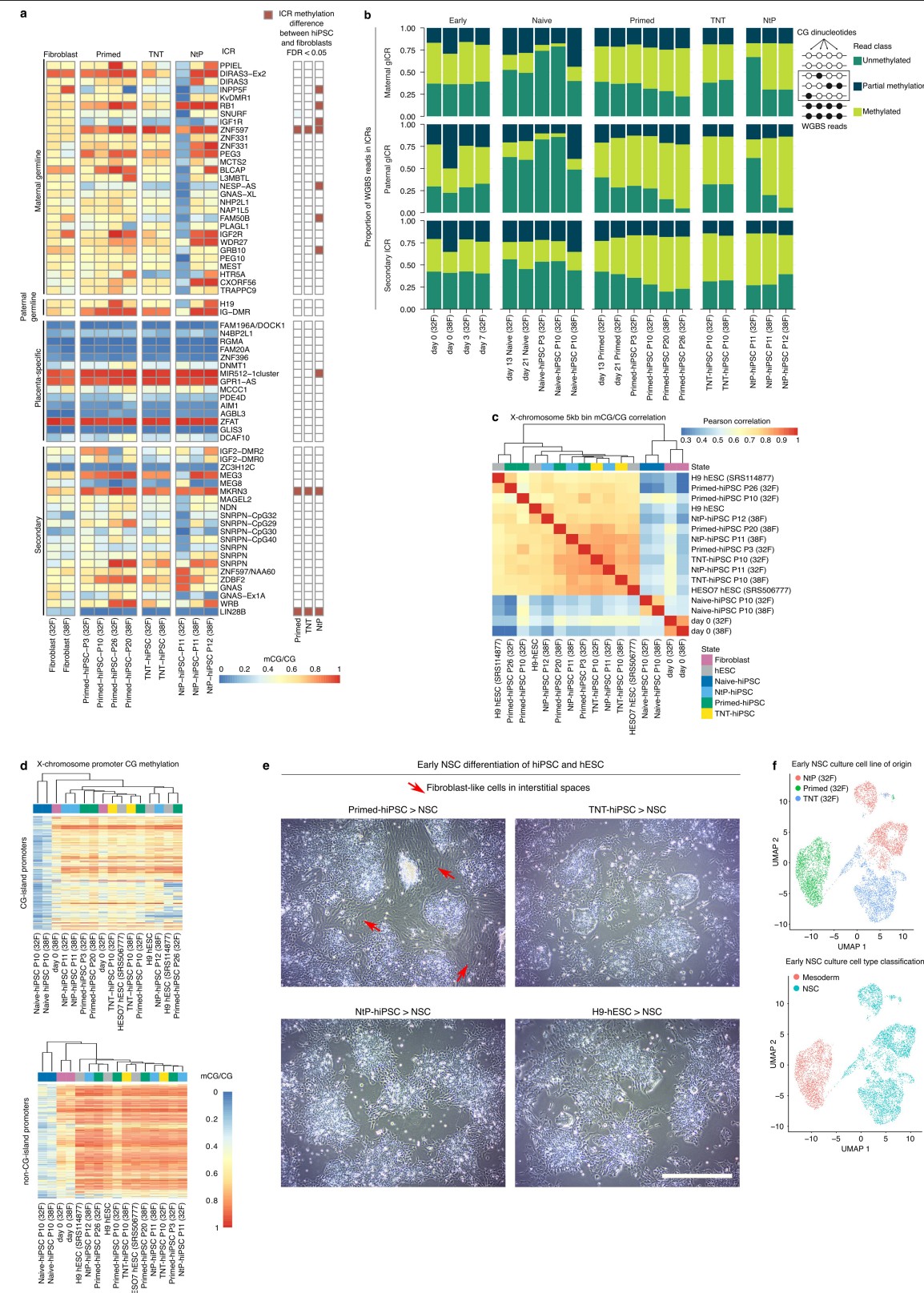

**Extended Data Fig. 5** | See next page for caption.

**Extended Data Fig. 5 | Genomic imprinting, X chromosome DNA methylation and neural stem cell differentiation of hiPSCs reprogrammed through the naive state. a)** CG methylation in imprint control regions (ICRs) for fibroblasts and hiPSCs reprogrammed from these fibroblasts. Right grid shows which hiPSC groups had significantly different (t-test FDR < 0.05) CG methylation levels compared to fibroblasts. The data indicate that TNT-hiPSCs do not show an increase in loss of imprinting over Primed-hiPSCs, in contrast to NtP-hiPSCs. ICRs as defined previously[21]. **b)** Proportion of methylated, unmethylated, and partially methylated WGBS reads in different classes of ICRs. **c)** Correlation matrix heatmap showing Pearson correlation levels of samples, calculated from CG-DNA methylation levels in 5 kb bins of the X-chromosome. **d)** Heatmaps of promoter DNA methylation levels split by CG island intersecting promoters (upper) and those promoters not intersecting CG islands (lower). **e)** Bright field microscopy images of early NSC cultures (3-7 days after plating embryoid bodies) generated from the different hiPSC lines. Large stretched-out fibroblast-like cells are evident during differentiation from Primed–hiPSCs, exemplified by the red arrow. Scale bar: 200 μm. **f)** UMAP plots from scRNA-seq analysis of early NSC cultures coloured by treatment group (reprogramming method, upper) and cell type classification (lower). Accompanies Fig. 3n.

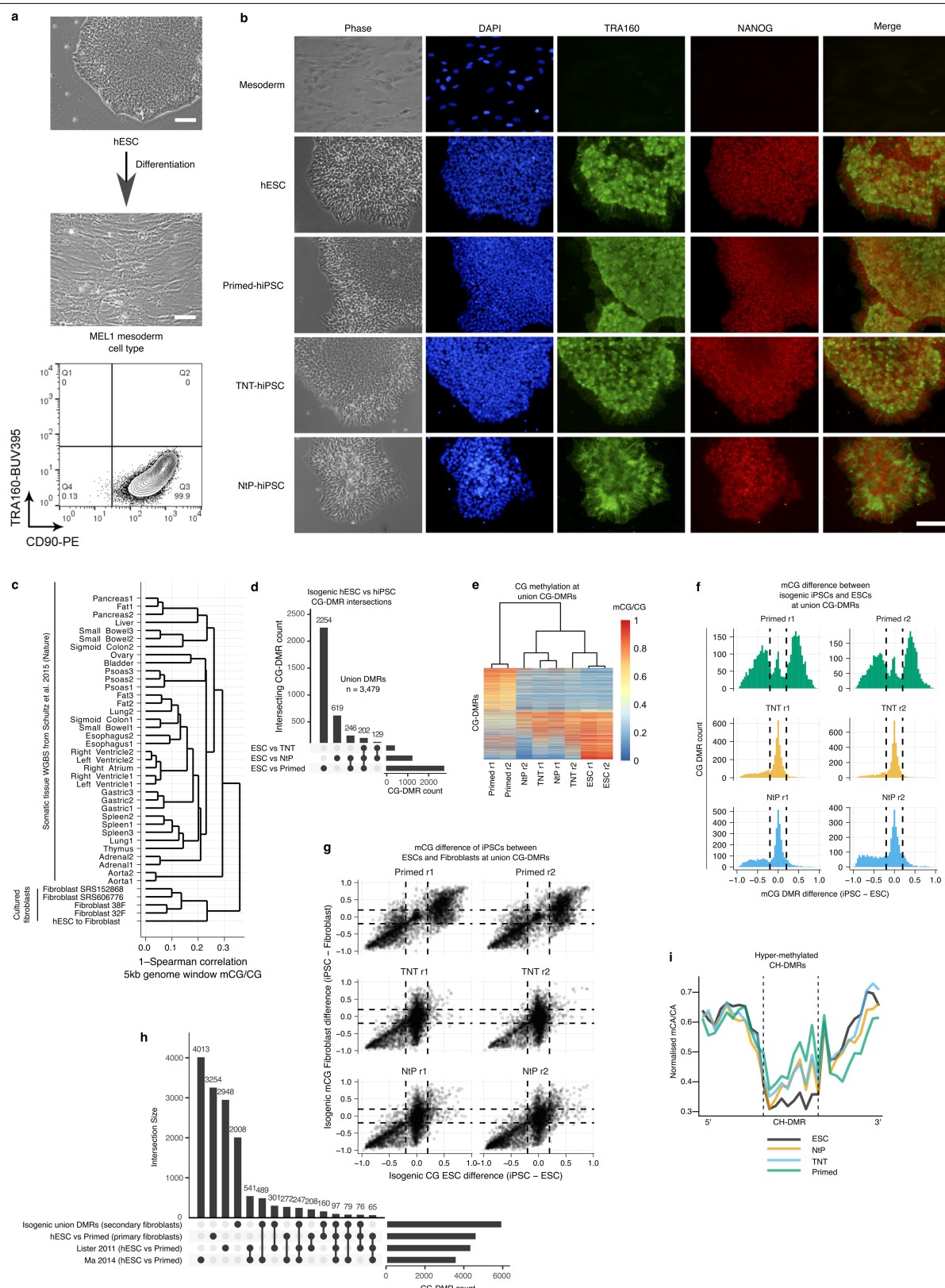

**Extended Data Fig. 6 |** See next page for caption.

**Extended Data Fig. 6 | Isogenic differentiation and reprogramming system confirms transient-naive-treatment reprogramming enhances epigenome resetting. a)** Phase contrast images showing the generation of fibroblast cells from MEL1 hESCs, where these cells were TRA160 negative and CD90 (Thy1) positive as shown by FACS analysis. Scale bar: 100 μm. **b)** Immunostaining of pluripotency markers NANOG and TRA160 for the MEL1 hESCs and the different Primed-hiPSC, TNT-hiPSC, and NtP-hiPSC lines derived from the MEL1-derived fibroblast-like cells, n = 2 independent experiments. Scale bar: 100 μm. **c)** Hierarchical clustering of 5 kb genome bin mCG/CG values for human tissues, cultured fibroblasts, and fibroblasts differentiated from hESCs. Somatic tissue WGBS data from Schultz et al. (2015)[73]. **d)** Upset plot showing the number of intersecting CG-DMRs detected between the hESC and hiPSC lines. **e)** Heatmap of CG DNA methylation levels in all lines in CG-DMRs detected between isogenic hESCs and Primed-hiPSCs, where r represents the replicate number. **f)** Histograms of the difference in CG DNA methylation level at CG-DMRs for Primed-hiPSCs, TNT-hiPSCs, and NtP-hiPSCs. Vertical dashed lines indicate the threshold of 0.2 (i.e. 20%) difference in CG DNA methylation level at CG-DMRs. **g)** Scatter plot of the relative CG DNA methylation difference in CG-DMRs for hiPSCs compared to hESCs (x-axis) and hiPSCs compared to fibroblasts (y-axis). Individual CG-DMRs are represented by individual points. **h)** Upset plot showing intersecting CG-DMRs detected for isogenic secondary fibroblast Primed-hiPSCs compared with CG-DMRs for primary fibroblast Primed-hiPSCs from this study and samples from previously published studies. **i)** Aggregate profile plot of CA methylation levels in hyper-methylated CH-DMRs.

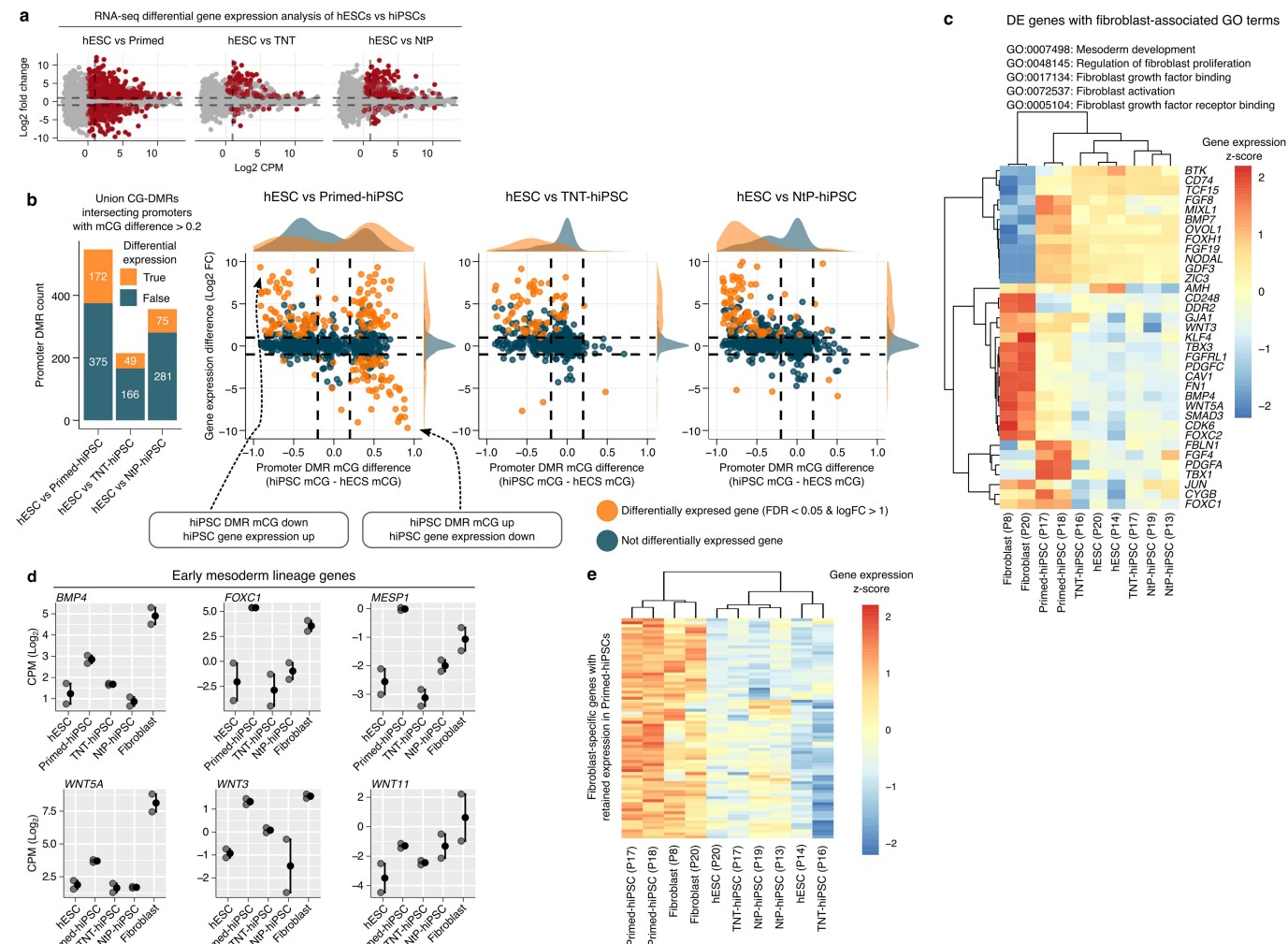

**Extended Data Fig. 7 | Isogenic differentiation and reprogramming system confirms transient-naive-treatment reprogramming corrects transcriptional profiles of hiPSCs. a)** MA plots showing differentially expressed genes between hESCs and each class of hiPSC (Primed, TNT, NtP). Red points represent significantly differentially expressed genes ($\log_2$FC > 1, FDR < 0.05, $\log_2$CPM > 1). Plots indicate that TNT-hiPSCs and NtP-hiPSCs are more transcriptionally similar to hESCs than Primed-hiPSCs. **b)** Barplots (left) show the number of CG-DMRs that intersect promoters, for CG-DMRs detected in hiPSCs compared to hESCs. Colours indicate the proportion of genes linked to promoters that show significant differential expression (FDR < 0.05, $\log_2$FC > 1). Scatter plots show the relationship between promoter DNA methylation differences between hiPSCs and hESCs (x-axis) and gene expression differences (y-axis). Individual points indicate DMR-gene pairs, with point colours indicating if the gene was differentially expressed. **c)** Heatmap showing clustered standardised gene expression values for differentially expressed genes with fibroblast-associated gene ontology terms. **d)** Gene expression levels for early mesoderm lineage genes. Grey points represent individual samples, n = 2 independent experiments per group, error bars show mean and range. **e)** Gene expression heatmap of fibroblast-specific genes with retained expression in Primed-hiPSCs.

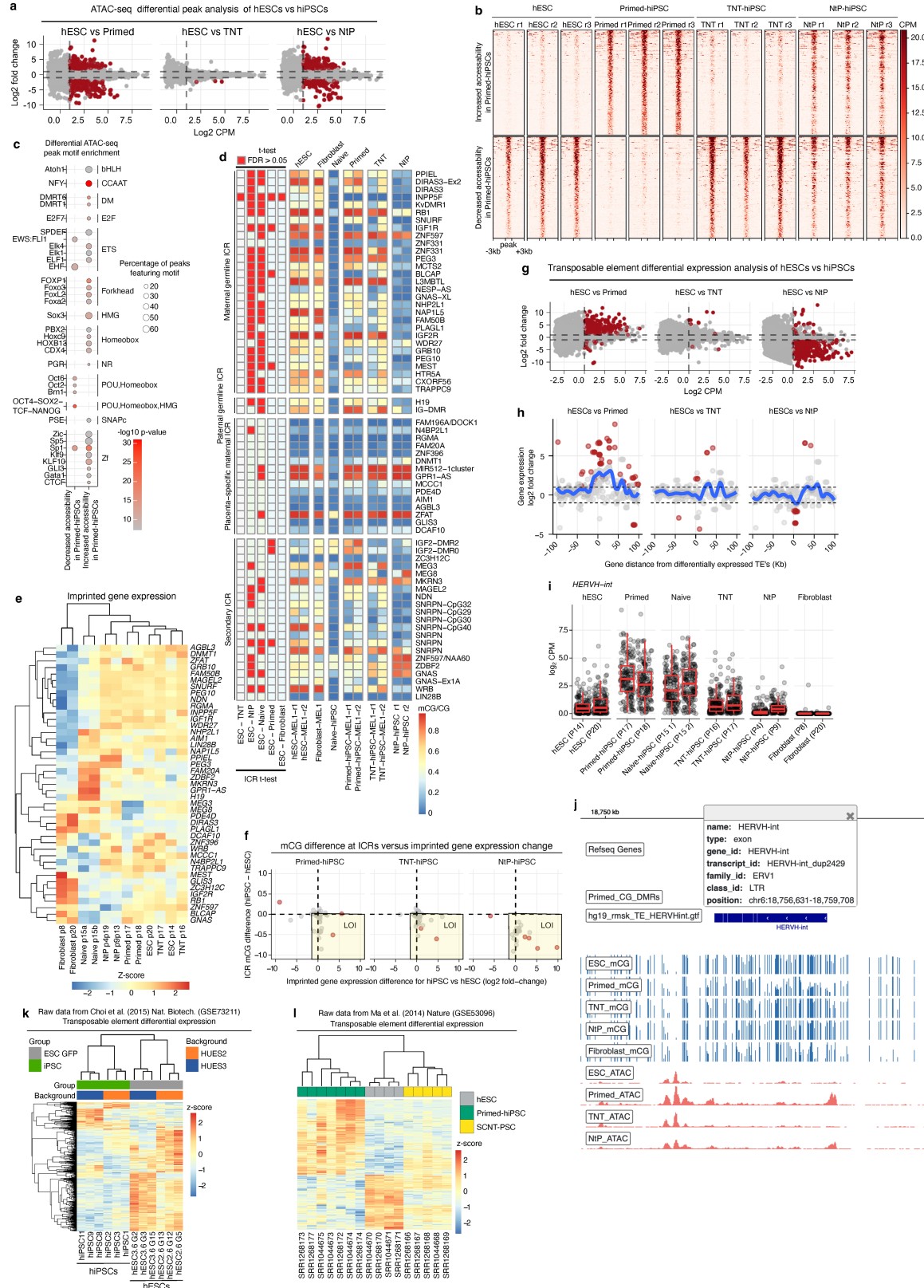

**Extended Data Fig. 8 | See next page for caption.**

**Extended Data Fig. 8 | Isogenic differentiation and reprogramming system confirms TNT-hiPSCs maintain imprinting and feature corrected transposable element expression. a)** MA plots and **b)** heatmap representation of differential ATAC-seq peaks between hESCs and hiPSCs, for each class of hiPSC (Primed, TNT, NtP). **c)** Transcription factors (TFs) with significantly enriched motifs in differential ATAC-seq peaks. **d)** CG-DNA methylation levels in ICRs for isogenic hESCs and all derived and reprogrammed lines. Grid with red squares on the left indicates if differential methylation between hESC and hiPSC was detected using the two-sample t-test with $p < 0.05$. ICRs defined in[21]. **e)** Gene expression heatmap and clustering of imprinted genes for isogenic hESCs, hiPSCs, and fibroblasts. Gene expression values are log2 CPM normalised and z-score scaled. **f)** Scatter plots of the relationship between DNA methylation change at imprint control regions (ICRs, y-axis) and imprinted gene expression difference for hiPSCs compared to hESCs. Each point represents an ICR and the linked imprinted gene. Yellow box highlights the data points potentially indicative of loss of imprinting (LOI), represented by loss of CG methylation and transcriptional gain. Red points indicate genes that are differentially expressed ($log_2FC > 1$, FDR < 0.05, $log_2CPM > 1$). **g)** MA plots showing differentially expressed transposable elements (TEs) between hESCs and each class of hiPSC (Primed, TNT, NtP). Red points represent significantly differentially expressed TEs ($log_2FC > 1$, FDR < 0.05, $log_2CPM > 1$), indicating that TNT-hiPSCs are more transcriptionally similar to hESCs than Primed-hiPSCs for TEs. **h)** Gene expression fold change (y-axis) relative to the distance (x-axis) from a differentially expressed TE. Individual points represent genes, with red points indicating significant differential expression as defined above. Blue line is a loess smoothed curve of fold change values over distance. **i)** Boxplots with data points show expression level of HERVH-int elements differentially expressed between hESCs and Primed-hiPSCs. boxplots: median and IQR, whiskers = $1.5 \times$ IQR. n = 1 independent experiment per boxplot. n = 1 independent experiment per boxplot. **j)** Browser screenshot of the HERVH-int_dup2429 locus with CG methylation and normalised ATAC-seq read counts for hESCs and hiPSCs. **k)** Differential expression heatmap of relative TE expression in HUES2 and HUES3 hESCs and Primed-hiPSCs derived from secondary fibroblasts in matched isogenic systems. Raw data are from[11] and were re-analysed using the same methods as in this study. **l)** Differential expression heatmap of relative TE expression in hESCs, Primed-hiPSCs, and SCNT-PSCs. Raw data are from Ma et al. (2014)[13] and were re-analysed using the same methods as in this study.

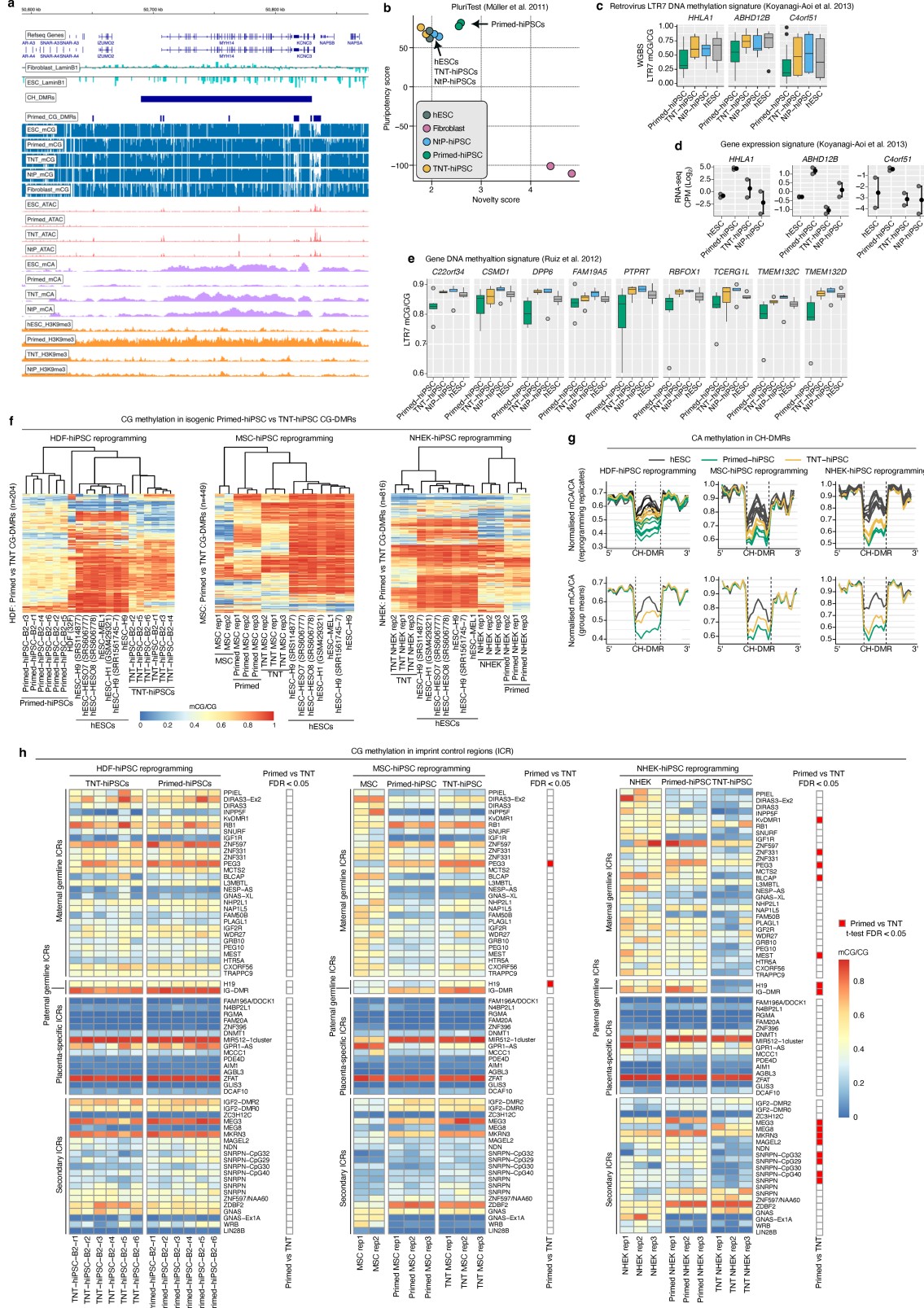

**Extended Data Fig. 9 |** See next page for caption.

**Extended Data Fig. 9 | Evaluation of TNT-hiPSCs using previously published criteria for hiPSC assessment, and TNT and Primed reprogramming of adult dermal fibroblasts, mesenchymal stem cells, and keratinocytes with DNA methylation profiling by WGBS. a)** Genome track of the *MYH14/KCNC3* CH-DMR. **b)** Results from PluriTest showing pluripotency and novelty scores for the isogenic fibroblasts, hiPSCs, and hESCs[38]. **c)** Boxplots showing CG methylation levels in LTR7 regions. Boxplots: median and IQR, whiskers = $1.5 \times$ IQR. n = 2 independent experiments per boxplot. **d)** Expression of genes previously defined for classifying hiPSC differentiation capacity[39], n = 2 independent experiments per group, error bars show mean and range. **e)** Boxplots of CG methylation in gene regions previously described as being able to segregate hESC and hiPSC lines regardless of the somatic cell source or differentiation state[6]. Boxplots: median and IQR, whiskers = $1.5 \times$ IQR. n = 2 independent experiments per boxplot. **f)** Heatmap of CG methylation levels in CG-DMRs detected for each origin cell type (HDF: primary human dermal fibroblasts; MSC: mesenchymal stem cells; NHEK: keratinocytes), with hierarchical clustering. **g)** Profile plots showing CA methylation levels in CH-DMRs where there was a significant difference detected between hiPSCs and hESCs. Upper row shows line plots for each reprogramming replicate, lower row shows replicate mean. **h)** CG-DNA methylation levels in ICRs. Grid with red squares on the right indicates if differential methylation between Primed and TNT-hiPSCs was detected using the two-sample t-test with p < 0.05. ICRs defined previously[21].

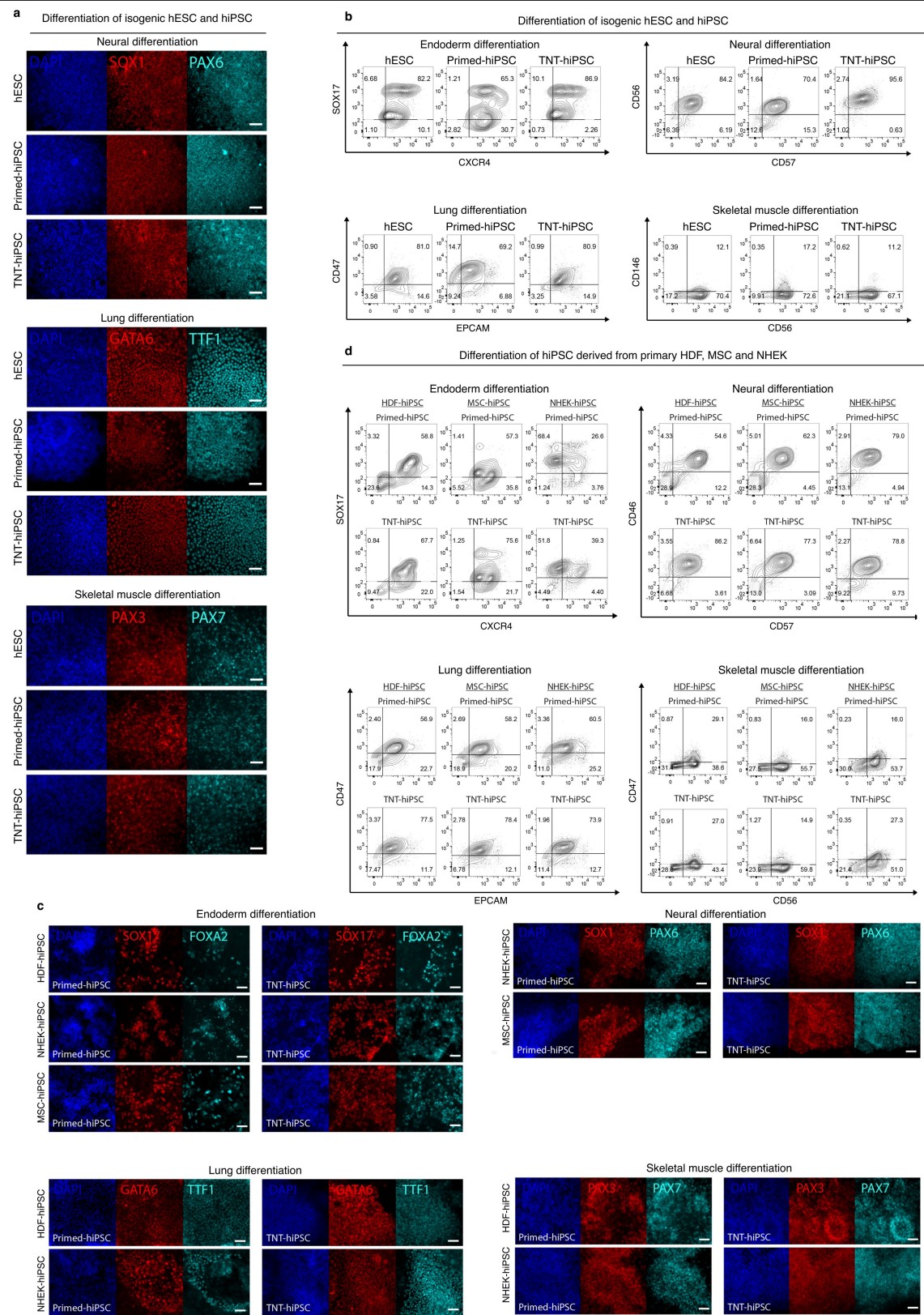

**Extended Data Fig. 10** | See next page for caption.

**Extended Data Fig. 10 | Differentiation of hiPSCs. a)** Representative immunofluorescence analysis images of cell differentiation: SOX1 and PAX6 for cortical neuron differentiation; GATA6 and TTF1 for lung epithelial differentiation; PAX3 and PAX7 for skeletal muscle differentiation. Scale bar: 50 μm. **b)** Representative flow cytometric profile of cell differentiation: CXCR4/SOX17 for endoderm differentiation; CD56/CD57 for cortical neuron differentiation; CD47/EPCAM for lung epithelial differentiation; and CD56/CD146 for skeletal muscle differentiation. **c)** Representative immunofluorescence analysis images of cell differentiation: SOX17 and FOXA2 for endoderm differentiation; SOX1 and PAX6 for cortical neuron differentiation; GATA6 and TTF1 for lung epithelial differentiation; PAX3 and PAX7 for skeletal muscle differentiation. Scale bar: 50 μm. **d)** Representative flow cytometric profile of cell differentiation: CXCR4/SOX17 for endoderm differentiation; CD56/ CD57 for cortical neuron differentiation; CD47/EPCAM for lung epithelial differentiation; and CD56/CD146 for skeletal muscle differentiation. Replicate details of the differentiation experiments can be found in the 'Statistic and reproducibility' section in Methods.

# nature research

| | |
|---|---|

# Reporting Summary

Nature Research wishes to improve the reproducibility of the work that we publish. This form provides structure for consistency and transparency in reporting. For further information on Nature Research policies, see Authors & Referees and the Editorial Policy Checklist.

## Statistics

For all statistical analyses, confirm that the following items are present in the figure legend, table legend, main text, or Methods section.

| n/a | Confirmed | |
|---|---|---|
| ☐ | ☒ | The exact sample size (*n*) for each experimental group/condition, given as a discrete number and unit of measurement |
| ☐ | ☒ | A statement on whether measurements were taken from distinct samples or whether the same sample was measured repeatedly |
| ☐ | ☒ | The statistical test(s) used AND whether they are one- or two-sided<br>*Only common tests should be described solely by name; describe more complex techniques in the Methods section.* |
| ☐ | ☒ | A description of all covariates tested |
| ☐ | ☒ | A description of any assumptions or corrections, such as tests of normality and adjustment for multiple comparisons |
| ☐ | ☒ | A full description of the statistical parameters including central tendency (e.g. means) or other basic estimates (e.g. regression coefficient) AND variation (e.g. standard deviation) or associated estimates of uncertainty (e.g. confidence intervals) |
| ☐ | ☒ | For null hypothesis testing, the test statistic (e.g. *F*, *t*, *r*) with confidence intervals, effect sizes, degrees of freedom and *P* value noted<br>*Give P values as exact values whenever suitable.* |
| ☒ | ☐ | For Bayesian analysis, information on the choice of priors and Markov chain Monte Carlo settings |
| ☒ | ☐ | For hierarchical and complex designs, identification of the appropriate level for tests and full reporting of outcomes |
| ☐ | ☒ | Estimates of effect sizes (e.g. Cohen's *d*, Pearson's *r*), indicating how they were calculated |

*Our web collection on statistics for biologists contains articles on many of the points above.*

## Software and code

Policy information about availability of computer code

| Data collection | Motic Image Plus 2.0, BD FACSDivia software version 8.0.1 for LSRIIa/b, BD FACS TM software, version 1.2.0.142 for influx sorters, DP controller version 2.1.1.183 and DP manager version 2.1.1.163 for fluorescence microscope imaging, Leica application suite X version 3.7.1.21655 for DMi8. |
|---|---|

| Data analysis | Data was analyzed using: GraphPad Prism (v7), ImageJ (v1.8.0_112), FlowJo (v10), Cellranger (v2.1.0, v3.0.2, v3.1.0), CITE-seq-Count (v1.4.3), bcl2fastq (v2.19), R(3.5.1), data.table(v1.12.2), Matrix(v1.2-17), ggplot2(v3.2.1), pheatmap(v1.0.12), shiny(v1.2.0), celda(v1.1.6), Seurat(v3.1.1), uwot(v0.1.4), irlba(v2.3.3), reticulate(v1.12), monocle3(v0.1.3), SingleCellExperiment(v1.4.1), scanpy(v1.4.4.post1), edgeR(v3.24.3), CytoTRACE(v0.1.0), metascape (v3.0), cutadapt (v1.8), bowtie2 (v2.3.2), MACS2 (v2.1.1.20160309), bedtools2 (v2.25.0), mfuzz (v2.38.0), homer (v4.10.3), BSseeker2 (v2.1.8), CGmapTools (v0.1.2), sambamba (v0.8.0), backports_1.2.1, Hmisc_4.5-0, BiocFileCache_1.14.0, plyr_1.8.6, lazyeval_0.2.2, splines_4.0.3, BiocParallel_1.24.1, digest_0.6.27, htmltools_0.5.1.1, ensembldb_2.14.1, foreach_1.5.1, fansi_0.4.2, checkmate_2.0.0, memoise_2.0.0, cluster_2.1.2, recipes_0.1.16, gower_0.2.2, R.utils_2.10.1, askpass_1.1, prettyunits_1.1.1, jpeg_0.1-8.1, colorspace_2.0-1, blob_1.2.1, rappdirs_0.3.3, xfun_0.23, dplyr_1.0.6, crayon_1.4.1, RCurl_1.98-1.3, graph_1.68.0, survival_3.2-11, iterators_1.0.13, glue_1.4.2, gtable_0.3.0, ipred_0.9-11, zlibbioc_1.36.0, DelayedArray_0.16.3, Rhdf5lib_1.12.1, HDF5Array_1.18.1, scales_1.1.1, futile.options_1.0.1, DBI_1.1.1, Rcpp_1.0.6, htmlTable_2.2.1, progress_1.2.2, foreign_0.8-81, bit_4.0.4, Formula_1.2-4, lava_1.6.9, prodlim_2019.11.13, htmlwidgets_1.5.3, httr_1.4.2, RColorBrewer_1.1-2, ellipsis_0.3.2, pkgconfig_2.0.3, XML_3.99-0.6, R.methodsS3_1.8.1, nnet_7.3-16, dbplyr_2.1.1, locfit_1.5-9.4, utf8_1.2.1, tidyselect_1.1.1, rlang_0.4.11, reshape2_1.4.4, munsell_0.5.0, tools_4.0.3, cachem_1.0.5, cli_2.5.0, generics_0.1.0, RSQLite_2.2.7, fastmap_1.1.0, yaml_2.2.1, ModelMetrics_1.2.2.2, bit64_4.0.5, purrr_0.3.4, AnnotationFilter_1.14.0, KEGGREST_1.30.1, packrat_0.6.0, RBGL_1.66.0, nlme_3.1-152, sparseMatrixStats_1.2.1, formatR_1.9, R.oo_1.24.0, xml2_1.3.2, biomaRt_2.46.3, compiler_4.0.3, rstudioapi_0.13, curl_4.3.1, png_0.1-7, tibble_3.1.2, stringi_1.6.2, futile.logger_1.4.3, ProtGenerics_1.22.0, Matrix_1.3-3, multtest_2.46.0, permute_0.9-5, vctrs_0.3.8, pillar_1.6.1, lifecycle_1.0.0, rhdf5filters_1.2.1, bitops_1.0-7, latticeExtra_0.6-29, R6_2.5.0, gridExtra_2.3, codetools_0.2-18, dichromat_2.0-0, lambda.r_1.2.4, MASS_7.3-54, assertthat_0.2.1, rhdf5_2.34.0, openssl_1.4.4, withr_2.4.2, regioneR_1.22.0, GenomicAlignments_1.26.0, GenomeInfoDbData_1.2.4, hms_1.1.0, VennDiagram_1.6.20, rpart_4.1-15, timeDate_3043.102, tidyr_1.1.2, class_7.3-19, DelayedMatrixStats_1.12.3, biovizBase_1.38.0, pROC_1.17.0.1, base64enc_0.1-3, lubridate_1.7.10 |

For manuscripts utilizing custom algorithms or software that are central to the research but not yet described in published literature, software must be made available to editors/reviewers. We strongly encourage code deposition in a community repository (e.g. GitHub). See the Nature Research guidelines for submitting code & software for further information.

# Data

Policy information about availability of data

All manuscripts must include a data availability statement. This statement should provide the following information, where applicable:
- Accession codes, unique identifiers, or web links for publicly available datasets
- A list of figures that have associated raw data
- A description of any restrictions on data availability

Raw and processed high throughput sequencing datasets have been deposited at the NCBI Gene Expression Omnibus (GEO) repository under the SuperSeries accession number GSE159297 that is composed of WGBS, bulk RNA-seq, single-cell RNA-seq, H3K9me3 ChIP-seq, ATAC-seq, and nanopore sequencing data. Bulk RNA-seq data for human naive and primed reprogramming intermediates are available under GSE149694.

# Field-specific reporting

Please select the one below that is the best fit for your research. If you are not sure, read the appropriate sections before making your selection.

☒ Life sciences    ☐ Behavioural & social sciences    ☐ Ecological, evolutionary & environmental sciences

For a reference copy of the document with all sections, see nature.com/documents/nr-reporting-summary-flat.pdf

# Life sciences study design

All studies must disclose on these points even when the disclosure is negative.

| Sample size | We did not involve statistical methods to predetermine the sample size, this was determined based on previous experience and other similar studies. All experiments were performed (if not otherwise stated) with at least two to three independent experiments with similar results. |
| Data exclusions | No data were excluded. |
| Replication | Each experiment was reproduced at least two to three times, with biological and/or technical replicates if not otherwise stated. Please refer to figure legends and methods for details. |
| Randomization | Randomization was done during reprogramming in which random wells were chosen to undergo primed reprogramming or TNT reprogramming, however for downstream characterization, randomization was not applicable in characterizing the molecular and functional difference between primed and TNT-reprogrammed iPS cells. |
| Blinding | The investigators were not blinded during data collection and analysis, as neither human/animal studies or specific grouping were involved in this manuscript. |

# Reporting for specific materials, systems and methods

We require information from authors about some types of materials, experimental systems and methods used in many studies. Here, indicate whether each material, system or method listed is relevant to your study. If you are not sure if a list item applies to your research, read the appropriate section before selecting a response.

## Materials & experimental systems

| n/a | Involved in the study |
|-----|-----------------------|
| ☐ | ☒ Antibodies |
| ☐ | ☒ Eukaryotic cell lines |
| ☒ | ☐ Palaeontology |
| ☒ | ☐ Animals and other organisms |
| ☒ | ☐ Human research participants |
| ☒ | ☐ Clinical data |

## Methods

| n/a | Involved in the study |
|-----|-----------------------|
| ☐ | ☒ ChIP-seq |
| ☐ | ☒ Flow cytometry |
| ☒ | ☐ MRI-based neuroimaging |

# Antibodies

| Antibodies used | Details of all antibodies used in this study were provided in Supplementary Table 9.

For flow cytometry:
PE-Cy7 mouse anti-human CD13 BD Biosciences Cat# 561599, clone WM15, 1:200 dilution
BUV395 mouse anti-human TRA-1-60 BD Biosciences Cat# 563878, clone TRA-1-60, 1:100 dilution
Anti-TRA-1-85 (CD147)-VioBright FITC Miltenyi Biotec Cat#130-107-106, clone REA476, 1:20 dilution
PE-SSEA3 BD Biosciences Cat#560237, clone MC-631, 1:10 dilution
F11R-APC, clone CSIRO CSTEM27APC, O'Brien et al., 2017, 1:200 dilution
PE mouse anti-Rat IgM eBiosciences Cat# 12-4342-82, clone RM-7B4, 1:250 dilution
AF647 goat anti-mouse IgG secondary ThermoFisher Cat#A21235, polyclonal, 1:400 dilution
BV 421 mouse anti-human CD326 (EpCAM) Biolegend Cat# 324220, clone 9C4, 1:100 dilution
Mouse anti-human F11R IgG2a clone CSIRO CSTEM27, O'Brien et al., 2017, 1:100 dilution
APC PSA-NCAM, Miltenyi Biotec, Cat# 130-120-437, clone 2-2B, 1:50 dilution
Anti-Histone H3 (tri methyl K9), abcam, Cat# ab8898, polyclonal, 1:100 dilution
PE-Cy7 CD146, BD Biosciences, Cat# 562135, clone P1H12, 1:100 dilution
BUV395 CD56, BD Biosciences, Cat# 563554, clone NCAM16.2, 1:100 dilution
APC CD57, Biolegend, Cat# 322314, clone HNK-1, 1:100 dilution
BUV395 CD47, BD Biosciences, Cat# 744308, clone B6H12, 1:200 dilution
PE anti-CXCR4, Miltenyi Biotec, Cat# 130-117-690, clone 12G5, 1:100 dilution
anti-SOX17, Abcam, Cat# 224637, EPR20684, 1:300 dilution
Alexa647 FAP, R&D Systems, Cat# FAB3715R, clone 427819, 1:100 dilution
Goat anti-mouse IgG2b AF647, ThermoFisher, Cat# A-21242, polyclonal, 1:1000 dilution
Goat anti-rabbit IgG AF488, ThermoFisher, Cat# A-11008, polyclonal, 1:1000 dilution

For Immunostaining:
Rabbit anti-NANOG polyclonal, Abcam, Cat# ab21624, polyclonal, 1:100 dilution
Mouse anti-TRA-1-60 IgM, BD Biosciences, Cat# 560071, clone TRA-1-60, 1:300 dilution
Goat anti-SOX17, R&D Systems, Cat# AF1924, polyclonal, 1:300 dilution
Rabbit anti-FOXA2, Abcam, Cat# ab256493, clone EPR22919-71, 1:200 dilution
Goat anti-SOX1, R&D Systems, Cat# AF3369, polyclonal, 1:200 dilution
Mouse anti-PAX6, IgG1  DHSB, Cat# PAX6, clone NA, 1:100 dilution
Goat anti-GATA6, R&D Systems, Cat# AF1700, polyclonal, 1:300 dilution
Rabbit anti-TTF1, Abcam, Cat# ab76013, clone EP1584Y, 1:200 dilution
Mouse anti-PAX3 IgG2a, R&D Systems, Cat# MAB2457, clone 274212, 1:200 dilution
Mouse anti-PAX7 IgG1, DHSB, Cat# PAX7, clone NA, 1:100 dilution
Donkey anti-mouse IgG-488 secondary ThermoFisher Cat# A-21202, polyclonal, 1:400 dilution
Donkey anti-goat IgG-555 secondary ThermoFisher Cat# A-21432, polyclonal, 1:400 dilution
Donkey anti-rabbit IgG-647 secondary ThermoFisher Cat# A-31573, polyclonal, 1:400 dilution
Goat anti-mouse IgG1-AF488 secondary ThermoFisher Cat# A-21121, polyclonal, 1:400 dilution
Goat anti-mouse IgG2a-AF647 secondary ThermoFisher Cat# A-21241, polyclonal, 1:400 dilution
Goat anti-mouse IgM AF488 secondary, ThermoFisher, Cat#A-21042, polyclonal, 1:400 dilution
Goat anti-rabbit IgG AF555 secondary, ThermoFisher, Cat#A-21428, polyclonal, 1:400 dilution

For ChIP-seq:
Rabbit polyclonal anti-H3K9me3 Abcam Cat# ab8898, 3 µg |

| Validation | Antibodies obtained from the commercial source were validated by the suppliers, and detailed validation analyses and relevant literatures are provided on the company website for the products used in this study. Some antibodies were validated in a previously published study as indicated in methods or relevant literature was cited.

PE-Cy7 mouse anti-human CD13 (561599) https://www.labome.com/product/BD-Biosciences/561599.html
BUV395 mouse anti-human TRA-1-60 (563878) https://www.bdbiosciences.com/zh-cn/products/reagents/flow-cytometry-reagents/research-reagents/single-color-antibodies-ruo/buv395-mouse-anti-human-tra-1-60-antigen.563878
Anti-TRA-1-85 (CD147)-VioBright FITC (130-107-106) https://www.miltenyibiotec.com/US-en/products/tra-1-85-cd147-antibody-anti-human-reafinity-rea476.html
PE-SSEA3 BD Biosciences (560237) https://www.bdbiosciences.com/en-nz/products/reagents/flow-cytometry-reagents/ |

research-reagents/single-color-antibodies-ruo/pe-rat-anti-ssea-3.560237

F11R-APC CSIRO CSTEM27APC, validated in O'Brien et al., 2017

PE mouse anti-Rat IgM (12-4342-82) https://www.thermofisher.cn/cn/zh/antibody/product/IgM-Antibody-clone-RM-7B4-Monoclonal/12-4342-82

AF647 goat anti-mouse IgG (A21235) https://www.thermofisher.cn/cn/zh/antibody/product/Goat-anti-Mouse-IgG-H-L-Cross-Adsorbed-Secondary-Antibody-Polyclonal/A-21235

BV 421 mouse anti-human CD326 (EpCAM) (324220) https://www.biolegend.com/en-us/search-results/brilliant-violet-421-anti-human-cd326-epcam-antibody-7549

Mouse anti-human F11R IgG2a CSIRO CSTEM27, validated in O'Brien et al., 2017

APC PSA-NCAM (130-120-437) https://www.miltenyibiotec.com/US-en/products/psa-ncam-antibody-anti-human-mouse-rat-2-2b

Anti-Histone H3 (tri methyl K9) (ab8898) https://www.abcam.com/products/primary-antibodies/histone-h3-tri-methyl-k9-antibody-chip-grade-ab8898.html

PE-Cy7 CD146 (562135) https://www.bdbiosciences.com/en-us/products/reagents/flow-cytometry-reagents/research-reagents/single-color-antibodies-ruo/pe-cy-7-mouse-anti-human-cd146.562135

BUV395 CD56 (563554) https://www.bdbiosciences.com/en-us/products/reagents/flow-cytometry-reagents/research-reagents/single-color-antibodies-ruo/buv395-mouse-anti-human-cd56.563554

APC CD57 (322314) https://www.biolegend.com/de-at/products/apc-anti-human-cd57-antibody-9023

BUV395 CD47 (Cat# 744308) https://www.bdbiosciences.com/en-at/products/reagents/flow-cytometry-reagents/research-reagents/single-color-antibodies-ruo/buv395-mouse-anti-human-cd47.744308

PE anti-CXCR4 (130-117-690) https://www.miltenyibiotec.com/IE-en/products/cd184-cxcr4-antibody-anti-human-12g5.html

anti-SOX17 (224637) https://www.abcam.com/products/primary-antibodies/sox17-antibody-epr20684-ab224637.html

Alexa647 FAP (FAB3715R) https://www.rndsystems.com/cn/products/human-fibroblast-activation-protein-alpha-fap-alexa-fluor-647-conjugated-antibody-427819_fab3715r

Goat anti-mouse IgG2b AF647 (A-21242) https://www.thermofisher.cn/cn/zh/antibody/product/Goat-anti-Mouse-IgG2b-Cross-Adsorbed-Secondary-Antibody-Polyclonal/A-21242

Goat anti-rabbit IgG AF488 (A-11008) https://www.thermofisher.cn/cn/zh/antibody/product/Goat-anti-Rabbit-IgG-H-L-Cross-Adsorbed-Secondary-Antibody-Polyclonal/A-11008

Rabbit anti-NANOG (ab21624) https://www.abcam.com/products/primary-antibodies/nanog-antibody-ab21624.html

Mouse anti-TRA-1-60 IgM (560071) https://www.bdbiosciences.com/en-nz/products/reagents/flow-cytometry-reagents/research-reagents/single-color-antibodies-ruo/purified-mouse-anti-human-tra-1-60-antigen.560071

Goat anti-SOX17 (AF1924) https://www.rndsystems.com/products/human-sox17-antibody_af1924

Rabbit anti-FOXA2 (ab256493) https://www.abcam.com/products/primary-antibodies/foxa2-antibody-epr22919-71-chip-grade-ab256493.html

Goat anti-SOX1 (AF3369) https://www.rndsystems.com/cn/products/human-mouse-rat-sox1-antibody_af3369

Mouse anti-PAX6 (PAX6) https://dshb.biology.uiowa.edu/PAX6

Goat anti-GATA6 (AF1700) https://www.rndsystems.com/cn/products/human-gata-6-antibody_af1700

Rabbit anti-TTF1 (ab76013) https://www.abcam.com/products/primary-antibodies/ttf1-antibody-ep1584y-ab76013.html

Mouse anti-PAX3 (MAB2457) https://www.rndsystems.com/cn/products/human-mouse-pax3-pax7-antibody-274212_mab2457

Mouse anti-PAX7 IgG1 (PAX7) https://dshb.biology.uiowa.edu/PAX7

Donkey anti-mouse IgG-488 (A-21202) https://www.thermofisher.cn/cn/zh/antibody/product/Donkey-anti-Mouse-IgG-H-L-Highly-Cross-Adsorbed-Secondary-Antibody-Polyclonal/A-21202

Donkey anti-goat IgG-555 (A-21432) https://www.thermofisher.cn/cn/zh/antibody/product/Donkey-anti-Goat-IgG-H-L-Cross-Adsorbed-Secondary-Antibody-Polyclonal/A-21432

Donkey anti-rabbit IgG-647 (A-31573) https://www.thermofisher.com/antibody/product/Donkey-anti-Rabbit-IgG-H-L-Highly-Cross-Adsorbed-Secondary-Antibody-Polyclonal/A-31573

Goat anti-mouse IgG1-AF488 (A-21121) https://www.thermofisher.cn/cn/zh/antibody/product/Goat-anti-Mouse-IgG1-Cross-Adsorbed-Secondary-Antibody-Polyclonal/A-21121

Goat anti-mouse IgG2a-AF647 (A-21241) https://www.thermofisher.cn/cn/zh/antibody/product/Goat-anti-Mouse-IgG2a-Cross-Adsorbed-Secondary-Antibody-Polyclonal/A-21241

Goat anti-mouse IgM AF488 (A-21042) https://www.thermofisher.cn/cn/zh/antibody/product/Goat-anti-Mouse-IgM-Heavy-chain-Cross-Adsorbed-Secondary-Antibody-Polyclonal/A-21042

Goat anti-rabbit IgG AF555 (A-21428) https://www.thermofisher.cn/cn/zh/antibody/product/Goat-anti-Rabbit-IgG-H-L-Cross-Adsorbed-Secondary-Antibody-Polyclonal/A-21428

# Eukaryotic cell lines

Policy information about cell lines

| | |
|---|---|
| Cell line source(s) | Human fibroblasts were sourced from ThermoFisher (Catalogue number, C-013-5C and lot#1029000 for 38F, lot#1569390 for 32F) for reprogramming experiments. MEL1 and H9 human embryonic stem cells were obtained from the Laslett lab as collaboration. Adipocyte-derived mesenchymal stem cells (MSCs) were obtained from the Heng lab. Normal human epidermal keratinocytes (NHEKs) were sourced from Lonza (donors 34014, lot# 0000665959) |
| Authentication | Human dermal fibroblasts and NHEKs were authenticated by ThermoFisher and Lonza respectively, and human embryonic stem cells were authenticated in the Laslett lab and MSCs authenticated in the Heng lab. Routinely, these cell lines were also authenticated in the lab via morphological assessment, immunofluorescence for identity markers or RNA-seq. |
| Mycoplasma contamination | Fibroblasts lines and NHEKs were tested by ThermoFisher and Lonza respectively, human embryonic stem cells were tested by the Laslett lab and MSCs by the Heng lab. Furthermore, cell lines were regularly tested and were mycoplasma negative. |
| Commonly misidentified lines (See ICLAC register) | No commonly misidentified cell lines were used in this study. |

# ChIP-seq

## Data deposition

☒ Confirm that both raw and final processed data have been deposited in a public database such as GEO.

☒ Confirm that you have deposited or provided access to graph files (e.g. BED files) for the called peaks.

| | |
|---|---|
| Data access links<br>*May remain private before publication.* | https://www.ncbi.nlm.nih.gov/geo/query/acc.cgi?acc=GSE159718<br>This accession provides access to the raw fastq files and bigwig fold-enrichment files. |
| Files in database submission | GSM4838439 D13_plus_10_32F_N2P_H3K9me3_ChIP<br>GSM4838440 P12_plus_13_38F_N2P_H3K9me3_ChIP<br>GSM4838441 P24_32F_primed_in_E8_H3K9me3_ChIP<br>GSM4838442 P16_38F_primed_in_E8_H3K9me3_ChIP<br>GSM4838443 P11_plus_11_38F_N2P_H3K9me3_ChIP<br>GSM4838444 D13_plus_7_32F_N2P_H3K9me3_ChIP<br>GSM4838445 D13_plus_7_38F_N2P_H3K9me3_ChIP<br>GSM4838446 D13_plus_10_32F_N2P_Input<br>GSM4838447 P24_32F_primed_in_E8_Input<br>GSM4838448 P17_MEL1_HDF_to_SR_H3K9me3_ChIP<br>GSM4838449 P13_plus_20_MEL1_to_E8_H3K9me3_ChIP<br>GSM4838450 P18_MEL1_HDF_to_E8_H3K9me3_ChIP<br>GSM4838451 P17_MEL1_D13_TNT_H3K9me3_ChIP<br>GSM4838452 P4_plus_10_TNT_MEL1_H3K9me3_ChIP<br>GSM4838453 P9_plus_6_TNT_MEL1_H3K9me3_ChIP<br>GSM4838454 P33_32F_Naive_SR_clone1_H3K9me3_ChIP<br>GSM4838455 P17_MEL1_HDF_to_SR_Input<br>GSM4838456 P13_plus_20_MEL1_to_E8_Input<br>GSM4838457 P18_MEL1_HDF_to_E8_Input<br>GSM4838458 P17_MEL1_D13_TNT_Input<br>GSM4838459 P4_plus_10_TNT_MEL1_Input<br>GSM4838460 P33_32F_Naive_SR_clone1_Input |
| Genome browser session<br>(e.g. UCSC) | no longer applicable |

## Methodology

| | |
|---|---|
| Replicates | Minimum of 2 biological replicates for 2 adult donors and 1 secondary reprogramming fibroblast line, and pluripotent cells treatment groups |
| Sequencing depth | 60-100 million reads |
| Antibodies | H3K9me3 antibody (Abcam, ab8898) |
| Peak calling parameters | H3K9me3 fibroblast and ESC peaks from ENCODE were used in this study. ENCFF963GBQ (fibroblast), ENCFF001SUW (hESC H3K9me3 peaks) |
| Data quality | As H3K9me3 is a broad histone mark that shows variability in peak with and intensity based on genomic context and region, we visually inspected in the genome browser for fold-enrichment over input libraries to assess quality. |
| Software | Bowtie2, samtools, deeptools. |

# Flow Cytometry

## Plots

Confirm that:

☒ The axis labels state the marker and fluorochrome used (e.g. CD4-FITC).

☒ The axis scales are clearly visible. Include numbers along axes only for bottom left plot of group (a 'group' is an analysis of identical markers).

☒ All plots are contour plots with outliers or pseudocolor plots.

☒ A numerical value for number of cells or percentage (with statistics) is provided.

## Methodology

| | |
|---|---|
| Sample preparation | Cells were dissociated with TrypLE express (ThermoFisher), and DPBS (ThermoFisher) supplemented with 2% FBS (Hyclone) and 10µM Y-27632 (Abcam) was used for antibody labeling steps and final resuspension of the samples. The antibody labeling steps were carried out in a volume of 500 µl per 1 million cells, and incubation time was 10 mins on ice per step; after each antibody |

labeling step, cells were washed with 10 ml cold PBS and pelleted at 400× g for 5 mins. The cells were then resuspended in a final volume of 500 µl, and propidium iodide (PI) (Sigma) was added to a concentration of 2µg/ml. Cell sorting was carried out with a 100 µm nozzle on an Influx instrument (BD Biosciences), and flow cytometry analysis was carried out using an LSRIIb or LSRIIA analyser (BD Biosciences).

| | |
|---|---|
| Instrument | LSRIIb, LSRIIa analyser or BD Influx cell sorters (BD). |
| Software | Collection: FACSDiva software suit (BD) for analysers, FACS TM software suit (BD) for influx sorters. Analysis: FlowJo (FlowJo, LLC) & Cytobank (Cytobank, Inc.). |
| Cell population abundance | Abundance of distinct cell populations of interest was determined using appropriate negative controls and puritiy of sorted populations as determined by post sort reanalysis. |
| Gating strategy | Standard gating settings commonly utilized at the flowcore facility of Monash University were used. Cell debris was excluded using a FSC vs SSC gate; aggregates were excluded via a FSC-H vs FSC-W approach; dead cells were defined as PI high/positve and gated out; furthermore iMEF feeder cells were gated out via the FITC channel (TRA-1-85 negative). Apart from using appropriate isotype, FMO and unstained controls, positive, negative control cell samples were used to set appropriate gates and determine real positive cell populations and confirmed by post sort reanalysis. |

☒ Tick this box to confirm that a figure exemplifying the gating strategy is provided in the Supplementary Information.

