## [Peer Review File · Nature]

Manuscript Title: Epigenetic and functional correction of human iPS cells by transient naive reprogramming

Reviewer Comments & Author Rebuttals

Reviewer Reports on the Initial Version:

Referees' comments:

Referee #1 (Remarks to the Author):

Over the past decade, multiple publications have documented the epigenetic distinctions between iPSCs and ESCs which are largely due to the differing cellular origins and culture conditions under which these pluripotent cell types are derived. Both “epigenetic memory”—ie, residual chromatin configurations that represent vestiges of the cell of origin—and aberrant chromatin configurations due to the imprecision of the reprogramming process contribute to such differences. Efforts to optimize reprogramming strategies to achieve epigenetic states more closely approaching that of native embryo-derived pluripotent stem cells represent an important goal given the importance of iPSCs for disease modeling and cell therapies. Protocols for Yamanaka-type reprogramming typically produce Primed-iPSCs which retain epigenetic memory. Naïve-type reprogramming produces a more globally hypo-methylated state, but the effects on epigenetic memory had not previously been widely documented. Lister, Polo and colleagues present compelling evidence that a transient naïve reprogramming approach produces iPSCs which are highly similar to human embryo-derived ESCs. The authors provide extensive molecular documentation and functional characterization to support their analysis. The paper is a tour de force.

Naïve conditions were found to accelerate global demethylation relative to primed conditions, resulting in earlier chromatin remodeling. From the summary: “We discovered that reprogramming-induced epigenetic aberrations establish midway through primed reprogramming, and that somatic cell memory erasure begins early during naïve reprogramming.”

Another important observation is that culture under primed aberrant CG methylation conditions promotes aberrant CG methylation. Line 236: “aberrant hyper-methylation did not accumulate in naive reprogramming (Fig. 2c), indicating that aberrant hyper-methylation is a feature of primed, and not naive reprogramming”

Naïve reprogramming is known to erase methylation at imprint control regions and leads to aberrant gene expression of imprinted loci. The authors probed the temporal dynamics of this demethylation and found that naïve conditions began imprint erasure between days 7-13 but wasn't complete until late, day 21, suggesting that prolonged culture was the culprit.

The authors then use this temporal analysis of demethylation to hypothesize that a transient exposure to naïve reprogramming conditions might more closely replicate the conditions of global demethylation that occurs in the cells of the early pre-implantation embryo. They tested generation

of hiPSCs via a transient-naïve-treatment (TNT-hiPSCs) and a prolonged naïve reprogramming followed by culture under primed conditions (NtP-hiPSCs). Both conditions triggered a closer approximation of the CG-DMRs to the hESC state than even SCNT “corrected” reprogramming (by rather modest amounts of 5.5 and 11.6%, respectively for TNT and NtP).

The authors then note that of the CG-DMRs “corrected” by TNT were over-represented amongst loci carrying repressive H3K9 methylation in fibroblasts, but under-represented in regions of H3K9 methylation in hESCs. Furthermore the corrected loci were associated with LADs in fibroblasts. They went on to show that these large repressive chromatin domains that were susceptible to maintaining epigenetic memory were efficiently reset by TNT reprogramming.

The authors go on to demonstrate that the loci that tend to harbor epigenetic memory in Primed-hiPSCs tend to be the large repressive heterochromatin domains associated with nuclear lamina, an observation that reinforces the conclusions of an earlier paper from Lister and colleagues (Nature 2011). Their overall conclusion is that TNT (and NtP) reprogramming do a better job of achieving methylation states more akin to hESCs than does prior efforts at primed reprogramming.

Analysis of ICRs suggested that the TNT method maintained more appropriate methylation than did NtP, which was associated with more aberrant loss or disordered gain of methylation. Both methods maintained XCI. Their data support the notion that TNT is more effective at retaining proper imprinting, which is otherwise lost with prolonged naïve culture.

Taken together, their data suggest that TNT is more effective at both erasure of epigenetic memory as well as maintenance of imprinting.

They then show that NSC cultures differentiated from Primed-hiPSCs produced more fibroblast-like cells, which was not observed with NSC cultures differentiated from TNT or NtP-hiPSCs. This observation was confirmed by single cell RNA sequencing.

To control for differing genetic backgrounds that might confound the conclusive differences between Primed-hiPSCs and hESCs, the authors performed a controlled isogenic cross-comparison of TNT and NtP and Primed-hiPSCs all derived from fibroblasts differentiated from hESCs. They go on to show that a window of naïve reprogramming erases the epigenetic memory of the fibroblast/mesodermal state, which they claim effectively minimizes the cell-of-origin bias for differentiation. These data are then reinforced and strengthened by assessing differentiation bias after TNT reprogramming relative to NtP and Primed for neurons, muscle, and lung epithelia.

Further evidence that TNT approximates the hESC state comes from analysis of differential TE expression, which is largely activated under Primed conditions yet maintained in a silenced (hESC-like) state in TNT-hiPSCs. Avoidance of TE reactivation may enable culture of more stable PSCs, given that TEs can account for mutagenesis and differences in differentiation capacity.

This paper puts forth a novel reprogramming protocol that appears to generate hiPSCs that are more akin to hESCs in their epigenetic state. Moreover, they offer a mechanistic dissection of the loci that are responsible for epigenetic memory, tying them to large regions of repressive heterochromatin

associated with LADs.

Minor point of revision: The descriptions of the dynamic changes in methylation status, as written, are confusing. Although the initial topic sentence (line 204) indicates that CG-DMRs are assessed relative to hESCs, the following descriptors of distinct states could be written to be more intuitive. Note that in line 134, the description “with the majority of CG positions in the genome becoming partially methylated” refers to the relative DE-methylation, or global loss of methylation, of loci under naïve conditions. Describing this as “becoming methylated” confuses the reader. A more accurate wording would state “global demethylation under naïve conditions results in a state of partial methylation at the majority of CG loci...” or something similar. Again, in lines 217-219, the wording “This revealed that 69% of hypo-methylated CG-DMRs in Primed-hiPSCs feature methylation levels <20% different to fibroblasts, which is indicative of somatic cell epigenetic memory (Memory DMRs, Fig. 2b).” ...is better described as “This revealed that in Primed-hiPSCs, 69% of CG-DMRs were <20% hypo-methylated relative to fibroblasts, which is indicative of somatic cell epigenetic memory”. And then the next reference is again confusing because the descriptor “hypo-methylated”, while relative to hESCs, is here used to describe a state in Primed-hiPSCs that is actually higher-level relative to fibroblasts: line 220-221 “An additional 21% of hypo-methylated CG-DMRs showed partial epigenetic memory, characterised by higher methylation in Primed-hiPSCs than in the fibroblast state but still remaining >20% lower compared to hESCs” would be less confusing if stated “An additional 21% of CG-DMRs that showed hypo-methylation relative to hESCs nevertheless harbored higher methylation in Primed-hiPSCs relative to fibroblasts, indicative of partial epigenetic memory.” Attention should be paid throughout to revise to make such descriptions as easy to follow as possible, lest the key messages get lost.

Referee #2 (Remarks to the Author):

In this manuscript, the authors performed temporal DNA methylation profiling during human primed and naïve reprogramming, and identified different time points where major DNA methylation change occurs in primed and naïve reprogramming. Based on the knowledge, they have developed the transient naïve treatment reprogramming strategy which resulted in hiPSC lines more similar to hESCs in terms of gene expression, DNA methylation patterns, and the differentiation capacity.

The data in this manuscript is high quality and seem to be solid. This transient naïve treatment could be practically very useful for the labs who struggle to generate high quality hiPSCs with a good differentiation capacity and for whom a large clonal variation affects down-stream analysis. As the transient naïve treatment protocol is so simple, I can see that anybody can incorporate it in their reprogramming pipeline without any harm.

What was not very clear to me was how frequently people have this (not being able to make good iPSCs) problem, and how small the clonal variation becomes by using this transient naïve treatment. For example, Butcher has reported that about 50% of 25 hiPSCs derived mesodermal lineage (mainly dermal fibroblasts) could differentiate well to the endoderm lineage (<https://www.nature.com/articles/ncomms10458>). In this manuscript, the differentiation

incapability (Fig 3lm) and bias (Fig 4jk) of primed iPSCs is very clear. But considering many groups use primed hiPSCs derived from fibroblasts (mesoderm) for neural (ectoderm) or endodermal lineage differentiation, not all the primed hiPSCs should have this problem.

In fact, several groups have reported a strategy to distinguish good and bad hiPSCs (<https://www.nature.com/articles/ncomms10458>, <https://www.nature.com/articles/nmeth.1580>, <https://www.nature.com/articles/nbt.1684>, <https://www.pnas.org/content/109/40/16196>, <https://www.pnas.org/content/110/51/20569>, etc), demonstrating 'good' primed iPSC lines. It means that some primed hiPSC lines are probably not as bad as ones in this manuscript, at least in terms of functionality.

Bock has demonstrated that some but not all hiPSC lines fall within the normal spectrum of ESC variation <https://www.sciencedirect.com/science/article/pii/S0092867410015242?via=ihub#dfig1>. I do believe demonstrating how much this transient naïve treatment can increase the frequency of obtaining 'good iPSC lines' would convince others to follow this protocol, e.g. make 25 TNT iPSC lines and test the Butcher's endoderm differentiation protocol/assessment to see what % of cell lines are good iPSC lines, compared to Butcher's work. (I do not think in vitro differentiated fibroblasts are good materials for this experiment, as they are not the source of cells people use.) In fact, one of the TNT samples (r1) in Extended Data Fig.7c shows much lower CG methylation at ICRs than ESCs. It would anyway be better to have and characterize more TNT iPSC clones.

In addition, so many criteria to select good iPSC clones have been suggested as above papers. Summarising those published criteria, and showing how primed-hiPSC and TNT hiPSC lines fits with these criteria, how much DMRs in this paper overlaps with previous published papers would be informative.

Minor points

If possible, it would be better to have naïve hESC samples in Fig 1bcd.

Line 155-156 "This included regions of overall gain (clusters 1-3), overall loss (cluster 4), and transient loss (cluster 5) of CG methylation." Some of these description fit only with primed reprogramming, so the sentence should be followed by "in primed reprogramming".

Figure 1f, very nice data.

Line 236-237 "Importantly, aberrant CG methylation did not accumulate in naïve reprogramming (Fig. 2c), indicating that aberrant hyper-methylation is a feature of primed, and not naïve reprogramming." I am not sure if this conclusion is correct, as naïve iPSCs were not compared with naïve ESCs.

Extended Data Fig. 8e. It would be interesting to see all the samples together in one hierarchical clustering figure, as it will show us if the genetic background variation has a larger impact than primed-reprogramming, or not. As it is, it is not possible to know if differentially expressed TE are same in the HUES2 and HUES3.

Referee #4 (Remarks to the Author):

General

The manuscript “Transient naïve reprogramming eliminates epigenetic memory and differentiation bias in human iPS cells” by Buckberry et al describes a modified protocol for somatic reprogramming aiming to obtain human induced pluripotent stem cells (hiPSCs) epigenetically and functionally equivalent to human embryonic stem cells (hESCs). Epigenetic aberrations and differentiation defects in hiPSCs have been previously reported in the literature, and despite both represent significant hindrances for hiPSCs clinical application, the problem remained unsolved. Therefore, the main question addressed by the paper is of significant interest and potentially has a high impact on the field.

The authors developed an approach based on transient culturing of intermediates of the somatic reprogramming in conditions for naïve hPSCs followed by transfer to conditions for conventional (primed) hPSCs. They performed an extensive epigenetic profiling, including genome-wide characterisation of DNA methylation, chromatin accessibility and H3K9me3 histone modification, and concluded that the resulted hiPSCs (TNT-hiPSCs and to a lesser extent NtP-hiPSCs) had significantly less epigenetic aberrations as compared to the hiPSCs obtained by previously published methods. Furthermore, the authors claim elimination of “lineage-of-origin differentiation bias” in hiPSCs obtained using transient naïve treatment.

The first important claim of the work is the epigenetic correction of hiPSCs using the novel transient naïve treatment protocol. Overall, the epigenetic profiling analysis has been performed thoroughly and the conclusions about epigenetic correction of TNT-hiPSC are convincing. There are several general questions that remain not addressed and require attention in order to support the claims and increase the impact of the work:

- Do the improved qualities of TNT-hiPSC result from selection of rare subpopulations or from conversion of cells in the naïve culture conditions? This is important for understanding the mechanistic basis and potential pitfalls of the protocol which is necessary for therapeutically relevant protocols.
- The authors employ conventional hESC as the golden standard of epigenetic state, and yet hESC accumulate epigenetic abnormalities during in vitro culturing including aberrant DNA methylation (Nazor et al Cell 2012; Weissbein et al PLOS Genetics 2017). How is the epigenetic profile of TNT-hiPSC aligned to early vs late passage hESC? Is it possible to compare to DNA methylation data from human embryos (such as reported in Zhou et al Nature 2019)? The choice of “the golden standard” is important to support the claim that TNT-hiPSC lack aberrations.
- Is the transient naïve treatment applicable for correction of existing hiPSC lines that carry epigenetic and / or functional aberrations?
- Does transient naïve treatment protocol result in epigenetically and functionally corrected hiPSC when applied for reprogramming cell types other than fibroblasts? The latter two points are critical to address in order to demonstrate the potential of the method. Whilst genome-wide profiling of additional reprogramming experiments may not be feasible, some targeted approaches (eg

amplicon-based) can be realistic to perform.

- Are TNT-hiPSC genetically normal? This is critical for demonstration of clinical applicability of the method.

The second important claim of the manuscript is the correction of the differentiation potential of TNT-hiPSC. This section of work presented only minimal data and requires much deeper assessment to support the conclusion and to demonstrate robustness of the result. It is also critical to provide a link between the epigenetic correction and the improved functional qualities, as claimed in the manuscript. The specific questions on this section are among the listed below.

As a general remark, the results overall seem reproducible and are mostly presented in several replicates. Nevertheless, it is not always clear what variables are controlled by the replicates (eg independent differentiation experiments of the same hiPSC line vs independent differentiation experiments using hiPSC lines from independent reprogramming rounds). Please make sure that critical results are reproduced in fully independent experiments and include this information in the manuscript. The statistics is overall well presented, however occasionally missed (please see specific comments below).

Overall, if the comments are addressed and the differentiation section is strengthened, the manuscript will suit for the publication in Nature journal. The presented method has the potential to open up new avenues in translational research and future applications and will be of significant interest for the field.

Major specific comments

1. Related to Figure 1 and Extended Data Figure 1: has the cell sorting strategy to enrich for reprogramming intermediates been validated? In particular, is there evidence that the cells sorted on different days of the reprogramming process indeed represent a continuum on a single trajectory? How heterogeneous are these cell populations? This is critical for the analysis of the DNA methylation and gene expression dynamics.

2. Related to Figure 1e: Please provide more detailed characterisation of the genes clusters, including details of gene ontology terms enrichment and examples of specific genes.

3. Related to Figure 1 and Extended Data Figure 1, lines 179-181: “the first wave of epigenetic remodelling at regulatory elements is driven by OKSM reprogramming factors, followed by distinct methylation states consolidated by the signalling cues in primed and naive conditions”. What signalling cues consolidate the distinct methylation states? What is the evidence for this statement?

4. Related to Figure 2 and Extended Data Figure 2, lines 207-208 (“Despite these DMRs being reproducible across lines, research groups, and reprogramming methods...”): how are the DMRs identified in the study comparable to the DMRs in other works (eg cited in the manuscript), in terms of their numbers, specific locations, co-localisation with H3K9me3, direction of change?

5. Related to Figure 2 and Extended Data Figure 2: published evidence suggests that continuous passaging of (mouse) iPSC attenuates epigenetic differences between iPSC and ESC (Polo et al., 2010). Are epigenetic aberrations of primed hiPSC corrected over continuous passaging?
6. Related to Figure 2 and Extended Data Figure 2, lines 248-250 (“loss of imprinting in naive media occurs through extended culturing, and is not as rapid as the initial global loss of CG methylation that occurs prior to day 13 of naive reprogramming”): a substantial loss of imprinting (from about 50-60% down to 30-35%) occurs by day 13 of the naïve reprogramming (Figure 2f, Extended Data Figure 2d) and therefore the notion is an overstatement.
7. Figure 3e: what is the reason for over-representation of the corrected CG-DMRs within K9me3 and fibroblast PMD and LAD? Is the correction more efficient in these regions? Or are the CG-DMRs more abundant in these regions in general? This is not very clear from the text, please explain this analysis.
8. Extended Figure 4a: uncorrected CG-DMRs are enriched in transcriptionally functional regions (promoters, exons, enhancers). Is there a functional consequence or significance of the methylation state in these regions? What genes are affected by these regions and is their expression affected?
9. Figure 3f, lines 315-318, “Regions enriched for H3K9me3 in fibroblasts that intersect CG-DMRs showed higher H3K9me3 in Primed-hiPSC compared to TNT-hiPSC (...) and NtP-hiPSC (...), which were both more similar to hESC”): please provide statistics confirming the similarity of TNT-hiPSCs and NtP-hiPSCs to hESCs.
10. Figure 3j and Extended Figure 4f, lines 357-358 (“This is in contrast to NtP-hiPSCs, where we observed more extensive loss of imprinting or disordered gain of methylation”): please provide evidence that NtP-hiPSCs gain methylation in ICR in a disordered fashion. Is it a gain of methylation or maintenance of methylation inherited from fibroblasts? Furthermore, among NtP-hiPSC lines, 32F but not 38F lost methylation in ICRs. Could you please provide wider statistic comparing TNT and NtP, what is the frequency of methylation state maintenance in the ICRs?
11. Figure 3l-m: cultures obtained from Primed-hiPSCs in the conditions for NSC differentiation predominantly contained mesodermal cells, whereas TNT- and NtP-hiPSCs-derived cultures were mostly composed of NSCs – how consistent is this outcome of NSC differentiation? Whilst single-cell RNAseq does not seem feasible to do on multiple samples, qRT-PCR or flow cytometry for markers can be indicative. Please provide statistics for multiple cells lines and independent experiments.
12. Extended Figures 3c and 6g; Figure 3f and Extended Figure 6f: the figures show independent experiments using Primed, TNT and NtP protocols either starting from primary fibroblasts or hESC-derived fibroblasts. Was the epigenetic correction consistent in these two experiments? Have the memory regions and the corrected regions coincided?
13. Supplementary Table 5, lines 467-470 (“Gene ontology analyses revealed that Primed-hiPSCs over-express 113 genes involved in mesoderm development (p=0.01, GO:0007498, Supplementary

Table 5), further demonstrating the cell-of-origin molecular memory in Primed-hiPSCs is corrected by reprogramming through the naive state.”): Supplementary Table 5 reveals that Primed-hiPSC also overexpress genes involved in epithelial cell differentiation (56 genes, $p=3.77E-09$), central nervous system development (77 genes, $p=1.15E-08$), skeletal development, muscle development, brain development, neurogenesis etc, all more significantly enriched than the “mesoderm development” GO term. What is the evidence that the gene expression differences are related to the epigenetic memory reflecting the fibroblast origin of these hiPSC?

14. Extended Figure 7a, Supplementary Table 5, related to RNAseq analysis: what fibroblast-specific genes retain their expression in Primed-hiPSC? Please provide more detailed characterisation of DE genes. This is important to support the point that the observed aberrations are due to epigenetic memory of the origin and not artifacts of reprogramming.

15. Figure 4b, c, Extended Figure 7b: NtP-hiPSCs show significant differences in their ATACseq profile from hESCs, which contrasts with their DNA methylation state that is similar to hESCs. What is the reason for these differences? What regulatory regions are affected, what TF binding motifs are enriched in the differential peaks?

16. Lines 477-479 (“With respect to genomic imprinting, TNT-hiPSCs do not show extensive demethylation at imprint control regions”), Extended Figure 7c: TNT_r1 does show significant loss of DNA methylation in ICR. How frequently DNA methylation is lost in ICRs during TNT reprogramming protocol? Can the protocol be optimised to more reliably retain DNA methylation in ICRs? Please comment in the text that the ICRs can be partially demethylated during TNT reprogramming, the current statement does not fully reflect the result.

17. Figure 4j-k, Extended Figure 9: correction of differentiation bias is crucial for the claim of the paper, this section requires expansion and more careful characterisation.

a. Please provide more detailed characterisation of the differentiated cells: immunostaining, qRT-PCR for markers, flow cytometry. Cortical neurons have much broader spectrum of specific markers than NCAM, so as the cells resulted from other differentiation protocols.

b. Please provide the results of not only independent differentiation experiments, but also of hiPSCs from independent reprogramming experiments.

c. The muscle differentiation protocol seems not to have worked in hESCs. Please show the differentiation protocol that works.

d. Please show the outcomes of alternative differentiation protocols to confirm robustness of the results.

e. Would differentiation bias be corrected if the reprogramming was done with other somatic cell types than fibroblasts (neurons or NSC, blood cells...)?

18. Discussion, lines 614-616 (“the lineage-of-origin differentiation bias in Primed-hiPSCs may be due to heterochromatic memory impacting TF binding dynamics”): it is important to build a mechanistic link between the correction of epigenetic state and the differentiation bias. What genomic elements (promoters or enhancers) associated with differentiation are affected in Primed-hiPSCs? Which of them are corrected in TNT-hiPSCs? Extensive epigenetic profiling performed by the authors allows to

address this question.

Minor specific comments

1. Related to Figure 1 and Extended Data Figure 1: please provide details of the markers used for cell sorting during naïve and primed reprogramming. The antibodies are listed in the Materials and Methods, however this is not mentioned in the paper which particular markers were used for each type of the reprogramming.
2. Figure 4k, middle panel (ectoderm differentiation result): please expand the axis, the current limits give a false impression that neuronal differentiation failed in Primed-hiPSC, although it is 50% efficient.
3. Please expand abbreviations (eg “REs” in Figure 4e).
4. Supplementary Table 5: for GO enrichment analysis, please provide adjusted p-values rather than p-values and list specific genes belonging to each GO term. This information is important for readers.

Point by point response to comments on Nature manuscript 2021-05-08601A

We extend our sincere thanks for the constructive feedback, time and effort in considering our manuscript. We believe we have addressed all comments and requested changes, which has helped improve our manuscript considerably. Below we provide a point-by-point response to all comments and suggestions.

Referee #1

Referee #1 (Remarks to the Author):

Over the past decade, multiple publications have documented the epigenetic distinctions between iPSCs and ESCs which are largely due to the differing cellular origins and culture conditions under which these pluripotent cell types are derived. Both “epigenetic memory”—ie, residual chromatin configurations that represent vestiges of the cell of origin—and aberrant chromatin configurations due to the imprecision of the reprogramming process contribute to such differences. Efforts to optimize reprogramming strategies to achieve epigenetic states more closely approaching that of native embryo-derived pluripotent stem cells represent an important goal given the importance of iPSCs for disease modeling and cell therapies. Protocols for Yamanaka-type reprogramming typically produce Primed-iPSCs which retain epigenetic memory. Naïve-type reprogramming produces a more globally hypo-methylated state, but the effects on epigenetic memory had not previously been widely documented.

Lister, Polo and colleagues present compelling evidence that a transient naïve reprogramming approach produces iPSCs which are highly similar to human embryo-derived ESCs. The authors provide extensive molecular documentation and functional characterization to support their analysis. The paper is a tour de force.

Naïve conditions were found to accelerate global demethylation relative to primed conditions, resulting in earlier chromatin remodeling. From the summary: “We discovered that reprogramming-induced epigenetic aberrations establish midway through primed reprogramming, and that somatic cell memory erasure begins early during naïve reprogramming.”

Another important observation is that culture under primed aberrant CG methylation conditions promotes aberrant CG methylation. Line 236: “aberrant hyper-methylation did not accumulate in naïve reprogramming (Fig. 2c), indicating that aberrant hyper-methylation is a feature of primed, and not naïve reprogramming”

Naïve reprogramming is known to erase methylation at imprint control regions and leads to aberrant gene expression of imprinted loci. The authors probed the temporal dynamics of this demethylation and found that naïve conditions began imprint erasure between days 7-13 but wasn't complete until late, day 21, suggesting that prolonged culture was the culprit.

The authors then use this temporal analysis of demethylation to hypothesize that a transient exposure to naïve reprogramming conditions might more closely replicate the conditions of global demethylation that occurs in the cells of the early pre-implantation embryo. They tested generation of hiPSCs via a transient-naïve-treatment (TNT-hiPSCs) and a prolonged naïve reprogramming followed by culture under primed conditions (NtP-hiPSCs). Both conditions triggered a closer approximation of the CG-DMRs to the hESC state than even SCNT "corrected" reprogramming (by rather modest amounts of 5.5 and 11.6%, respectively for TNT and NtP).

The authors then note that of the CG-DMRs "corrected" by TNT were over-represented amongst loci carrying repressive H3K9 methylation in fibroblasts, but under-represented in regions of H3K9 methylation in hESCs. Furthermore the corrected loci were associated with LADs in fibroblasts. They went on to show that these large repressive chromatin domains that were susceptible to maintaining epigenetic memory were efficiently reset by TNT reprogramming.

The authors go on to demonstrate that the loci that tend to harbor epigenetic memory in Primed-hiPSCs tend to be the large repressive heterochromatin domains associated with nuclear lamina, an observation that reinforces the conclusions of an earlier paper from Lister and colleagues (Nature 2011). Their overall conclusion is that TNT (and NtP) reprogramming do a better job of achieving methylation states more akin to hESCs than does prior efforts at primed reprogramming.

Analysis of ICRs suggested that the TNT method maintained more appropriate methylation than did NtP, which was associated with more aberrant loss or disordered gain of methylation. Both methods maintained XCI. Their data support the notion that TNT is more effective at retaining proper imprinting, which is otherwise lost with prolonged naïve culture.

Taken together, their data suggest that TNT is more effective at both erasure of epigenetic memory as well as maintenance of imprinting.

They then show that NSC cultures differentiated from Primed-hiPSCs produced more fibroblast-like cells, which was not observed with NSC cultures differentiated from TNT or NtP-hiPSCs. This observation was confirmed by single cell RNA sequencing.

To control for differing genetic backgrounds that might confound the conclusive differences between Primed-hiPSCs and hESCs, the authors performed a controlled isogenic cross-comparison of TNT and NtP and Primed-hiPSCs all derived from fibroblasts differentiated from hESCs. They go on to show that a window of naïve reprogramming erases the epigenetic memory of the fibroblast/mesodermal state, which they claim effectively minimizes the

cell-of-origin bias for differentiation. These data are then reinforced and strengthened by assessing differentiation bias after TNT reprogramming relative to NtP and Primed for neurons, muscle, and lung epithelia.

Further evidence that TNT approximates the hESC state comes from analysis of differential TE expression, which is largely activated under Primed conditions yet maintained in a silenced (hESC-like) state in TNT-hiPSCs. Avoidance of TE reactivation may enable culture of more stable PSCs, given that TEs can account for mutagenesis and differences in differentiation capacity.

This paper puts forth a novel reprogramming protocol that appears to generate hiPSCs that are more akin to hESCs in their epigenetic state. Moreover, they offer a mechanistic dissection of the loci that are responsible for epigenetic memory, tying them to large regions of repressive heterochromatin associated with LADs.

Response: We thank the reviewer for the review and for all the encouraging words in regards to our work.

Specific recommendations

Reviewer 1 - point 1: Minor point of revision: The descriptions of the dynamic changes in methylation status, as written, are confusing. Although the initial topic sentence (line 204) indicates that CG-DMRs are assessed relative to hESCs, the following descriptors of distinct states could be written to be more intuitive.

Response: Thanks for pointing this out, we have now revised these descriptions to try to make all of these points clearer.

Reviewer 1 - point 2: Note that in line 134, the description “with the majority of CG positions in the genome becoming partially methylated” refers to the relative DE-methylation, or global loss of methylation, of loci under naïve conditions. Describing this as “becoming methylated” confuses the reader. A more accurate wording would state “global demethylation under naïve conditions results in a state of partial methylation at the majority of CG loci...” or something similar.

Response: Thanks for this suggestion, we have now done exactly that. This sentence now reads “*As expected, global demethylation under naïve conditions results in a state of partial methylation at the majority of CG loci*”.

Reviewer 1 - point 3: Again, in lines 217-219, the wording “This revealed that 69% of hypo-methylated CG-DMRs in Primed-hiPSCs feature methylation levels <20% different to fibroblasts, which is indicative of somatic cell epigenetic memory (Memory DMRs, Fig. 2b).”...is better described as “This revealed that in Primed-hiPSCs, 69% of CG-DMRs were <20% hypo-methylated relative to fibroblasts, which is indicative of somatic cell epigenetic memory”.

Response: As suggested, this sentence now reads *“This revealed that in Primed-hiPSCs, 60.4% of CG-DMRs were <20% hypo-methylated relative to fibroblasts, which is indicative of somatic cell epigenetic memory.”* The percentage value in this sentence has been updated with the results for the new DMR calling with more hESC lines.

Reviewer 1 - point 4: And then the next reference is again confusing because the descriptor “hypo-methylated”, while relative to hESCs, is here used to describe a state in Primed-hiPSCs that is actually higher-level relative to fibroblasts: line 220-221 “An additional 21% of hypo-methylated CG-DMRs showed partial epigenetic memory, characterised by higher methylation in Primed-hiPSCs than in the fibroblast state but still remaining >20% lower compared to hESCs” would be less confusing if stated “An additional 21% of CG-DMRs that showed hypo-methylation relative to hESCs nevertheless harbored higher methylation in Primed-hiPSCs relative to fibroblasts, indicative of partial epigenetic memory.”

Response: As suggested, this sentence now reads *“An additional 20.9% of CG-DMRs that showed hypo-methylation relative to hESCs nevertheless harboured higher methylation in Primed-hiPSCs relative to fibroblasts, indicative of partial epigenetic memory.”*

Reviewer 1 - point 5: Attention should be paid throughout to revise to make such descriptions as easy to follow as possible, lest the key messages get lost.

Response: Thanks again for the suggestions. We have revised such descriptions in this section and throughout the manuscript with these comments in mind to improve clarity.

Referee #2

Referee #2 (Remarks to the Author):

In this manuscript, the authors performed temporal DNA methylation profiling during human primed and naïve reprogramming, and identified different time points where major DNA methylation change occurs in primed and naïve reprogramming. Based on the knowledge, they have developed the transient naïve treatment reprogramming strategy which resulted in hiPSC lines more similar to hESCs in terms of gene expression, DNA methylation patterns, and the differentiation capacity.

The data in this manuscript is high quality and seem to be solid. This transient naïve treatment could be practically very useful for the labs who struggle to generate high quality hiPSCs with a good differentiation capacity and for whom a large clonal variation affects down-stream analysis. As the transient naïve treatment protocol is so simple, I can see that anybody can incorporate it in their reprogramming pipeline without any harm.

Reviewer 2 - point 1: What was not very clear to me was how frequently people have this (not being able to make good iPSCs) problem, and how small the clonal variation becomes by using this transient naïve treatment. For example, Butcher has reported that about 50% of 25 hiPSCs derived mesodermal lineage (mainly dermal fibroblasts) could differentiate well to the endoderm lineage (<https://www.nature.com/articles/ncomms10458>). In this manuscript, the differentiation incapability (Fig 3lm) and bias (Fig 4jk) of primed iPSCs is very clear. But considering many groups use primed hiPSCs derived from fibroblasts (mesoderm) for neural (ectoderm) or endodermal lineage differentiation, not all the primed hiPSCs should have this problem.

Response: We understand Reviewer 2's point. Yes, although it is very likely that all Primed-hiPSCs have some degree of epigenetic aberrations (indeed the Salk institute has a granted patent that uses these DMRs to discriminate iPSCs from ESCs), not all Primed-hiPSCs derived from fibroblasts have such a degree of epigenetic aberrations as to render them unusable, otherwise, as the Reviewer points out, there would not be iPSC lines that can be used. However, what we would like to emphasise, as supported by our results and many previous publications¹⁻³, is that with the standard Primed reprogramming method, iPSCs exhibit a broad range of epigenetic aberrations making them extremely heterogeneous, which is reflected in their variable differentiation potential as Butcher *et al.* has reported, and as we show in our original and new experiments (see below). As suggested by the Reviewer, and addressing the additional comments below, we have now performed additional experiments and analyses to characterise and quantify the spectrum of hiPSC variance with respect to previously published criteria (see responses below). Furthermore, successful differentiation of a Primed-iPSC clone does not preclude the possibility that the resulting cells could still have aberrant expression of a subset of genes or TEs, with potential undesired consequences.

Reviewer 2 - point 2: In fact, several groups have reported a strategy to distinguish good and bad hiPSCs (<https://www.nature.com/articles/ncomms10458>, <https://www.nature.com/articles/nmeth.1580>, <https://www.nature.com/articles/nbt.1684>, <https://www.pnas.org/content/109/40/16196>, <https://www.pnas.org/content/110/51/20569>, etc), demonstrating 'good' primed iPSC lines. It means that some primed hiPSC lines are probably not as bad as ones in this manuscript, at least in terms of functionality.

Bock has demonstrated that some but not all hiPSC lines fall within the normal spectrum of ESC variation

<https://www.sciencedirect.com/science/article/pii/S0092867410015242?via=ihub#dfig1>.

I do believe demonstrating how much this transient naïve treatment can increase the frequency of obtaining 'good iPSC lines' would convince others to follow this protocol, e.g. make 25 TNT iPSC lines and test the Butcher's endoderm differentiation protocol/assessment to see what % of cell lines are good iPSC lines, compared to Butcher's work. (I do not think in vitro differentiated fibroblasts are good materials for this experiment, as they are not the source of cells people use.) In fact, one of the TNT samples (r1) in Extended Data Fig.7c shows much lower CG methylation at ICRs than ESCs. It would anyway be better to have and characterize more TNT iPSC clones.

Response: We thank the Reviewer for this great suggestion. With regard to characterising more hiPSC clones, in our revised manuscript we present extensive new data from new TNT-hiPSC and Primed-hiPSC clones and their differentiation, as outlined in various responses below. We

have now also included a table detailing all the different lines and experiments that we derived and tested (see Extended Data Fig. 13). In summary, we have now generated 12 independent TNT-hiPSC lines and 12 independent Primed-iPSC lines from four different starting cell types (primary fibroblasts, secondary fibroblasts, keratinocytes, mesenchymal stem cells), with three independent iPSC reprogramming experiments performed for each starting cell type (see Response Fig. 1a / Revised Manuscript Fig. 5a). We then performed the Butcher *et al.* endoderm differentiation⁴ (in addition to several other differentiation assays, see Response Fig. 1 / Revised Manuscript Fig. 5a) with all these 24 lines, performing multiple technical replicates for every differentiation assay (see Response Fig. 1a / Revised Manuscript Fig. 5a, and Extended Data Fig. 13 for specific numbers). For the Butcher *et al.* endoderm differentiation process specifically, we differentiated each independent hiPSC line (i.e. the 24 lines total) as follows: for the 3x primary HDF- and 3x secondary fibroblast-derived iPSC lines we performed 4 technical replicates each (i.e. total n=24, comprised of n=12 for each set of Primed-hiPSC and TNT-hiPSC lines); for the 3x MSCs-derived iPSC lines 3 technical replicates were performed for each (total n=9 for each condition); and for the 3x NHEK-derived iPSC lines 2 technical replicates were performed for each (total n=6 for each condition). Overall, this is a total of 39 Primed-hiPSC and 39 TNT-hiPSC differentiation replicates for the Butcher *et al.* endoderm protocol (see Extended Data Fig. 13). These data show that a) TNT-hiPSCs show significantly higher differentiation capacity into the endoderm lineage compared to Primed-hiPSCs, and b) TNT-hiPSCs are much more homogenous in regards to differentiation, while conversely Primed-hiPSCs show greater between-sample variance compared to TNT-hiPSCs (Response Fig. 1b / Revised Manuscript Fig. 5b).

Response Figure 1 / Revised Manuscript Figure 5: Multi-lineage reprogramming and differentiation confirms transient-naive-treatment reprogramming enhances epigenome resetting. **a)** Schematic representation of the experimental design of multi-lineage Primed and TNT reprogramming and differentiation into five different cell types. Upper panels show the four somatic cell lines reprogrammed into Primed-hiPSCs and TNT-hiPSCs with three independent reprogrammings performed per group, and with each subsequently differentiated into five different cell types, with independent replication. Lower panel shows the number of independent differentiation replicates performed for each origin cell type (rows) and differentiated cell type (columns). Coloured circles represent Primed-hiPSC (green), TNT-hiPSC (yellow), and hESC (grey). **b)** Quantification of the endoderm differentiation of 2^o Fibroblast-derived iPSCs, showing the proportion of resulting cells positive for FOXA2 and SOX17 by immunofluorescence (IF) analysis. **c)** Representative images for immunofluorescence analysis of FOXA2 and SOX17 in endoderm differentiation of 2^o Fibroblast-derived iPSCs. **d)** Quantification of multilineage cell differentiation in the different hiPSC lines by FACS and IF analyses: CD56/CD57/PAX6/SOX1 for cortical neuron differentiation; CD146/CD56/PAX3/PAX7 for skeletal muscle differentiation; and CD47/EPCAM/GATA6/TTF1 for lung epithelial differentiation. **e)** Representative IF analysis images of cell differentiation: Cortical neuron differentiation - SOX1 and PAX6; Skeletal muscle differentiation - PAX3 and PAX7; lung epithelial differentiation - GATA6 and TTF1. **f)** Phase contrast images taken 4 days after passaging plated embryoid bodies during differentiation into NSCs. Large stretched-out fibroblast-like cells are evident during differentiation from Primed-hiPSCs, exemplified by the red arrows. **g)** Percentage of NCAM+/FAP- cells after plating of embryoid bodies during differentiation into neural stem cells (NSCs), as measured by FACS.

Reviewer 2 - point 3: In addition, so many criteria to select good iPSC clones have been suggested as in the above papers. Summarising those published criteria, and showing how primed-hiPSC and TNT hiPSC lines fit with these criteria, how much DMRs in this paper overlaps with previous published papers would be informative.

Response: Again, thanks for this suggestion. Indeed, several studies have previously described gene expression and/or DNA methylation signatures for quantifying pluripotency and differentiation potential, and identifying reprogramming-specific epigenetic signatures. Below we outline how we have now summarised these studies in our revised manuscript, added several new analyses to show how our data fit within the criteria outlined in these studies, and assessed overlap of DMRs between this study and previous papers.

Published criteria summary and analysis: we have added a summary of the published criteria, and analysed how primed-hiPSC and TNT-hiPSC lines fit with these criteria, to the revised manuscript (Response Fig. 2 / Extended Data Fig. 11b-e) as follows: *“To further validate the ability of TNT reprogramming to produce hiPSCs that more closely resemble ESCs than conventional reprogramming, we then assessed various published criteria^{9,57,58} for using DNA methylation and gene expression signatures for selecting good hiPSC clones. Using these independent criteria, the results consistently suggest TNT-hiPSCs would produce better hiPSCs for differentiation (Extended Data Fig. 11b-e).”*

Response Figure 2 / Revised Manuscript Extended Data Figure 11: b) Results from PluriTest showing pluripotency and novelty scores for the isogenic fibroblasts, hiPSCs, and hESCs⁸. **c)** Boxplots showing CG methylation levels in LTR7 regions, and **d)** gene expression of genes previously defined for classifying hiPSC differentiation capacity⁶. **e)** Boxplots of CG methylation in gene regions previously described as being able to segregate hESC and hiPSC lines regardless of the somatic cell source or differentiation state⁹.

Bock *et al.* also developed a pluripotency ‘scorecard’ based on Reduced Representation Bisulfite Sequencing (RRBS) quantification of promoter methylation and/or gene expression¹⁰. However, due to the coverage constraints imposed by the RRBS DNA methylation method used, which enriches for specific sequences, we believe it is not possible to meaningfully assess our data with the scorecard method. This is because the DNA methylation scorecard was developed with RRBS data which is largely unable to measure DNA methylation in the regions where we find the greatest differences between hiPSCs and hESCs. Nevertheless, to test the scorecard algorithm, we followed the Bock *et al.* method and processed their RRBS data as described, and then added our Whole Genome Bisulfite Sequencing (WGBS) data along with WGBS data from three other studies^{1,3,11}. The results from this analysis showed that RRBS data estimates drastically

lower DNA methylation levels for the scorecard promoter regions compared to WGBS (Response Fig. 3). Consequently, as the scorecard reference standard was developed with RRBS, all WGBS samples (hESCs and hiPSCs) were shown to have >50% of promoters with outlier levels of increased methylation (Response Fig. 3). This suggests that the scorecard algorithm for DNA methylation in Bock *et al.* is not suitable for WGBS data. We also recognise that Bock *et al.* do specify in the description of their algorithm to use RRBS data. As WGBS data are not compatible with the Bock *et al.* algorithm, we do not wish to include the DNA methylation scorecard assessments of our data in the revised manuscript.

Overlap of DMRs between this study and previous papers: With respect to how the DMRs overlap between the different studies, we obtained the raw data from the Lister *et al.* 2011 (*Nature*)¹ and Ma *et al.* 2014 (*Nature*)³ studies and processed the data in exactly the same way as we did for this study. We believe this enables a fair comparison that is not impacted by the distinct analysis methods of each study (spanning over a decade). What this analysis revealed is that although there are common sites, CG-DMRs vary substantially for each study (Response Fig. 4 a,b / Revised Manuscript Extended Data Fig. 5 a,b). However, the enrichment of CG-DMRs in cell-type specific repressive chromatin domains is largely similar between this study and published studies (Response Fig. 4c,f / Revised Manuscript Extended Data Fig. 5c). As each study used different cell culture media (technical variation) and cell lines with different genetic

backgrounds (biological variation), with different sample sizes and sequencing depth (power and sensitivity), we performed a principal component analysis of CG methylation levels for all DMR sets combined, in an attempt to dissect out the common signal that separates hiPSCs from hESCs. This revealed that principal component 3 (PC3) clearly separates Primed-hiPSCs and hESCs (Response Fig. 4 g,h). Moreover, TNT-hiPSCs are more similar to hESCs for PC3 eigenvalues (Response Fig. 4h). This indicates that when focusing on the CG-DMRs that are most strongly influenced by study, TNT-hiPSCs are distinct from Primed-hiPSCs across all studies assessed. Moreover, when we assess CH-DMRs, we observe significant reproducibility between studies (Response Fig. 4e). Additionally, we see a consistent enrichment of CG-DMRs within the large CH-DMRs, including in the reanalysed data from Lister *et al.* 2011¹ and Ma *et al.* 2014³ (Response Fig. 4d). This appears to reflect an overall lower level of CG methylation in CH-DMRs for hiPSCs that resembles the partially methylated state that the fibroblasts exhibit in large repressive heterochromatin domains. We see this as the potential reason explaining the low direct overlap between CG-DMR coordinates between studies, as these CG-DMRs seem not to be indicative of discrete regulatory elements with relatively strict boundaries. Rather, they are indicative larger regions that show reduced CG DNA methylation. For example, when inspecting the CH-DMR that contains the *TCERG1L* locus, we observe distinct clustering of many Primed-hiPSC CG-DMRs within the CH-DMR for our data, as well as for Primed-hiPSC CG-DMRs from the Lister *et al.* 2011¹ and Ma *et al.* 2014³ studies (Response Fig. 4f). Therefore, when considering the larger CH-DMR features in Primed-hiPSCs, we see a considerable degree of overlap of these regions between different studies. Together, these analyses further support our hypothesis that epigenetic memory in hiPSCs is at least in part due to the retention of large repressive chromatin domains from the somatic cells from which the hiPSCs were reprogrammed.

a CG-DMR overlaps for this study and published studies

e CA methylation in CH-DMRs for this study and published studies

f CG-DMR clustering within CH-DMRs and fibroblast repressive chromatin

c CG-DMR enrichment in cell-type specific repressive chromatin domains

d Enrichment of CG-DMRs and regulatory elements in CH-DMRs

g Principal component analysis of union CG-DMRs for all studies combined

h Combined CG-DMR mCG/CG principal components for all study datasets and hESC/hiPSC group

i Combined CG-DMR mCG/CG principal components and relationship with this study and published studies

Response Figure 4 / Extended Data Fig. 5: Comparison of CG and CH DMRs across studies. **a)** Upset plot shows number of CG-DMRs detected for this study and how they overlap with CG-DMRs detected from previously published data processed using identical methods. **b)** Difference in DNA methylation level between hiPSCs and hESCs at CG-DMRs identified between Primed-hiPSCs and hESCs. Vertical dashed lines indicate the threshold of 20% minimum difference in CG DNA methylation level at CG-DMRs. **c)** Enrichment z-score determined from permutation testing of enrichment of CG-DMRs in repressive chromatin domains and of **d)** CH-DMRs in published studies. **e)** Heatmap of CA methylation levels in CH-DMRs in this study and previously published studies showing Primed-hiPSCs from all studies clustering separately to hESCs. **f)** Genome track of a CH-DMR region that intersects a PMD, fibroblast lamina associated domain (LAD), and clusters of CG-DMRs in each study. **g)** Principal component analysis of CG methylation levels in CG-DMRs for all studies combined. Top left plot shows the proportion of variance explained by each principal component. Scatter plots with coloured points show principal component separation of hESCs, Primed-hiPSCs, and TNT-hiPSCs. Ellipses around points indicate 95% confidence interval for a multivariate t-distribution. These data indicate that principal component 3 (PC3) in the bottom left plot clearly separates Primed-hiPSCs and hESCs for all studies, and shows that TNT-hiPSCs are more similar to hESCs by this measure. **h)** Plots of eigenvalues for each principal component for Primed-hiPSCs, TNT-hiPSCs, and hESCs, and **i)** data split by study/lab. Red bars indicate $P < 0.05$ for ANOVA test, with FDR reported above red bars.

Minor points

Reviewer 2 - point 4: If possible, it would be better to have naïve hESC samples in Fig 1bcd.

Response: As suggested, we have now added data for naïve hESCs to the revised manuscript Fig. 1b-d, as shown below (Response Fig. 5 / Revised Manuscript Fig. 1b-d).

Reviewer 2 - point 5: Line 155-156 “This included regions of overall gain (clusters 1-3), overall loss (cluster 4), and transient loss (cluster 5) of CG methylation.” Some of these descriptions fit

only with primed reprogramming, so the sentence should be followed by “in primed reprogramming”.

Response: As suggested, we have changed this sentence to “This included regions of overall gain (clusters 1-3), overall loss (cluster 4), and transient loss (cluster 5) of CG methylation in primed reprogramming.”

Reviewer 2 - point 6: Figure 1f, very nice data.

Response: Thank you!

Reviewer 2 - point 7: Line 236-237 “Importantly, aberrant CG methylation did not accumulate in naive reprogramming (Fig. 2c), indicating that aberrant hyper-methylation is a feature of primed, and not naive reprogramming.” I am not sure if this conclusion is correct, as naive iPSCs were not compared with naive ESCs.

Response: Thank you for pointing this out. We have now revised this section which now reads:

“Importantly, the sites of aberrant CG hyper-methylation we detected in Primed-hiPSCs were not aberrant in naive reprogramming intermediates or endpoints (Fig. 2c), indicating that these aberrant hyper-methylation loci are a feature of primed, and not naive reprogramming.”

Reviewer 2 - point 8: Extended Data Fig. 8e. It would be interesting to see all the samples together in one hierarchical clustering figure, as it will show us if the genetic background variation has a larger impact than primed-reprogramming, or not. As it is, it is not possible to know if differentially expressed TE are the same in the HUES2 and HUES3.

Response: Reviewer 2 raises a good point here. We have re-plotted these data as suggested. Indeed, this new plot indicates that primed reprogramming has a bigger impact than genetic background (Response Fig. 6). This plot is now included in the revised manuscript as Extended Data Fig. 10e.

Response Figure 6 / Revised Manuscript Extended Data Figure 10e: Differential expression heatmap of relative TE expression in HUES2 and HUES3 hESCs and Primed-hiPSCs derived from secondary fibroblasts in matched isogenic systems. Raw data are from¹² and were re-analysed using the same methods as in this study.

Referee #4

Referee #4 (Remarks to the Author):

General

The manuscript “Transient naive reprogramming eliminates epigenetic memory and differentiation bias in human iPSCs” by Buckberry et al describes a modified protocol for somatic reprogramming aiming to obtain human induced pluripotent stem cells (hiPSCs) epigenetically and functionally equivalent to human embryonic stem cells (hESCs). Epigenetic aberrations and differentiation defects in hiPSCs have been previously reported in the literature, and despite both representing significant hindrances for hiPSCs clinical application, the problem remained unsolved. Therefore, the main question addressed by the paper is of significant interest and potentially has a high impact on the field.

The authors developed an approach based on transient culturing of intermediates of the somatic reprogramming in conditions for naïve hPSCs followed by transfer to conditions for conventional (primed) hPSCs. They performed an extensive epigenetic profiling, including genome-wide

characterisation of DNA methylation, chromatin accessibility and H3K9me3 histone modification, and concluded that the resulted hiPSCs (TNT-hiPSCs and to a lesser extent NtP-hiPSCs) had significantly less epigenetic aberrations as compared to the hiPSCs obtained by previously published methods. Furthermore, the authors claim elimination of “lineage-of-origin differentiation bias” in hiPSCs obtained using transient naïve treatment.

The first important claim of the work is the epigenetic correction of hiPSCs using the novel transient naïve treatment protocol. Overall, the epigenetic profiling analysis has been performed thoroughly and the conclusions about epigenetic correction of TNT-hiPSC are convincing. There are several general questions that remain not addressed and require attention in order to support the claims and increase the impact of the work:

Reviewer 4 - point 1: Do the improved qualities of TNT-hiPSC result from selection of rare subpopulations or from conversion of cells in the naïve culture conditions? This is important for understanding the mechanistic basis and potential pitfalls of the protocol which is necessary for therapeutically relevant protocols.

Response: Excellent question! To address this, we transfected human fibroblasts with a lentivirus to integrate a specific insert sequence randomly into their genomes, creating a population of cells with a large diversity of different lentiviral insertion locations that we could subsequently trace following reprogramming. This single population of cells was split and simultaneously reprogrammed by the conventional Primed or TNT methods, allowing us to assess whether the different reprogramming methods yielded iPSCs differing diversity of lentiviral insertions, which would be expected if one method resulted in a greater “bottleneck” selection of subpopulations of the starting cell population. Following reprogramming, we used guide RNAs to mediate Cas9 cleavage of the inserted DNA sequence approximately 1 kb from the 5’ and 3’ ends of the integrated lentiviral sequence, and then mapped the locations of the integrations using Oxford Nanopore Technology long-read sequencing. This enabled us to quantify the number of unique lentiviral integrations detectable following each reprogramming method, and calculate population diversity (see methods section of revised manuscript). These results indicate that TNT-hiPSCs do not result from a selection of a rare subpopulation of cells, and have no statistically significant differences in population diversity compared to Primed-hiPSCs (Response Fig. 7 / Revised Manuscript Extended Data Fig. 4i). We do note that Primed-hiPSCs show greater variance in our estimations of diversity.

Reviewer 4 - point 2: The authors employ conventional hESC as the golden standard of epigenetic state, and yet hESC accumulate epigenetic abnormalities during *in vitro* culturing including aberrant DNA methylation (Nazor et al Cell 2012; Weissbein et al PLOS Genetics 2017). How is the epigenetic profile of TNT-hiPSC aligned to early vs late passage hESC? Is it possible to compare to DNA methylation data from human embryos (such as reported in Zhou et al Nature 2019)? The choice of “the golden standard” is important to support the claim that TNT-hiPSCs lack aberrations.

Response: With regards to the first point on early and late passage hESCs, we acknowledge that previous work has shown that *in vitro* culturing of hESCs can lead to epigenetic aberrations. But resolving this would require the *de novo* generation of many hESC lines to have full control of the passage. Moreover, this would require us to hold a licence to derive new hESC lines, which we do not have. Nevertheless, it is important to mention that many of the measurements we have done through this manuscript are independent of the passage as we are not always comparing against hESC, and rather we compare Primed-hiPSCs and TNT-hiPSCs of identical passage.

We agree with Reviewer 4’s comment that the choice of ‘the golden standard’ is of critical importance. In revising our manuscript, we have carefully considered the use of *in vitro* hESCs as the standard by which we compare hiPSCs. We think that the use of cells of the human embryo, such as those in Zhou *et al.*¹³, is not an ideal comparator in our view for several reasons. In principle, hiPSCs are meant to behave as hESCs (*in vitro* derived from human embryos) and not the cells of the epiblast. Indeed, hESCs are different from human embryo epiblast cells, as these epiblast cells represent a naive state of pluripotency, ESCs are immortal whereas epiblast cells behave more as progenitors, and epiblast cells have globally lower DNA methylation than hESCs cultured in primed media. Even at day 10 of implantation these cells only reach ~60% global CG methylation (see Fig. 4a from Zhou *et al.*¹³ as compared to the 70-85% CG methylation level we see in human ESC culture systems. These global DNA methylation differences make it very difficult to meaningfully compare *in vivo* vs *in vitro* stem cells.

We emphasise that our primary advances are centred around TNT reprogramming producing cells that more closely resemble *in vitro* hESCs than conventional hiPSC reprogramming does, due to the major importance and applicability of *in vitro* pluripotent cells. *In vitro* hESCs are widely used in the study of pluripotency, differentiation, and the development of stem cell therapies, as opposed to epiblast cells extracted directly from a human embryo that are exceptionally difficult to study and use, for both technical and ethical reasons. Hence, we chose *in vitro* hESCs as our 'gold standard' comparators. To capture as much variation as possible in hESCs we have included five different hESCs lines (H1, H9, HESO7, HESO8, MEL1) with data generated by four different labs, three other than our own (Smith, Mitalipov, Ecker), which were cultured in three different primed media (KSR, E8, mTeSR). We believe our incorporation of all these hESC lines constitutes a robust *in vitro* hESC standard against which we can measure epigenetic differences between hiPSCs and hESCs. Moreover, we performed a series of experiments with genetically matched hESCs and hiPSCs for the clearest possible comparison of these two cell types *in vitro*, which completely supports our claim that TNT-hiPSCs more closely resemble hESCs than conventional Primed-hiPSCs. By taking all of these factors into account, we maintain that *in vitro* hESCs are currently the best standard by which we can evaluate the epigenetic aberrations in hiPSCs, and the most applicable system in the context of the use of human pluripotent cells.

[REDACTED]

[REDACTED]

[REDACTED]

Reviewer 4 - point 3: Is the transient naïve treatment applicable for correction of existing hiPSC lines that carry epigenetic and / or functional aberrations?

Response: Thank you for raising this important question. In short, we find the TNT method with its brief naive media treatment is not applicable for correcting aberrations in existing hiPSC lines. However, more extended naive culturing of existing hiPSC lines does appear to correct epigenetic aberrations, but has various drawbacks as we outline below.

To test this, we took existing hiPSC lines with confirmed epigenetic aberrations and transitioned these to naive media for 5 days as per the TNT protocol. However, despite repeated attempts, Primed-hiPSCs do not adapt well to the naive media that we used in such a short period, and the cells fail to form colonies or resemble naive iPSCs (Response Fig. 9).

Response Figure 9: Images of Primed-hiPSCs (Top) being transitioned to naive media for 5 days (middle). In this short period, Primed-hiPSCs do not adapt well to the naive media and the cells fail to form colonies or resemble naive iPSCs (Bottom).

Therefore, we also cultured these same lines in naive media for an extended period (14 passages) until the cells formed the characteristic naive hiPSC colonies and then transitioned these naive cells to primed media, and profiled the DNA methylome of both populations by whole genome bisulfite sequencing. This experiment showed that the “Primed-to-Naive” hiPSCs (PtN-hiPSCs) feature widespread DNA demethylation as expected and that the “Primed-Naive-Primed” hiPSCs (PNP-hiPSCs) do indeed show partial correction of their epigenome for CG methylation and relatively complete correction for CH methylation (Response Fig. 10 / Revised Manuscript Extended Data Fig. 4f-h). However, like NtP-hiPSCs, we see increased variation in CG methylation at imprint control regions (ICRs) in PNP-hiPSCs, including both higher and lower levels of mCG (Response Fig. 10h). Moreover, it is important to keep in mind that extended culturing of cells in naive conditions can lead to an increase in genetic abnormalities, as we and others have reported^{14–17} and as we show in revised manuscript Extended Data Fig. 3b. These results lead us to conclude that although epigenetic correction is possible with Primed-Naive-Primed reprogramming, the correction is best performed during the reprogramming process to avoid the genome instability and loss of imprinting that occurs in PNP-hiPSCs. Moreover, it is important to note that PNP reprogramming also requires the addition of reprogramming factors in addition to the naive media treatment.

We have included the results of these new experiments in the revised manuscript, which now states: *“As expected from passaging in naive media, PtN-hiPSCs show widespread CG demethylation (Extended Data Fig. 4f). Upon re-priming, the PNP-hiPSCs exhibit remethylation and correction of a subset of CG-DMRs detected between Primed-hiPSCs and hESCs (Extended Data Fig. 4f). Furthermore, PNP-hiPSCs show correction of many of the CH-DMRs present in Primed-hiPSCs (Extended Data Fig. 4g). Therefore, PNP reprogramming can correct aberrant DNA methylation patterns in Primed-hiPSCs. However, similar to NtP-hiPSCs, in PNP-hiPSCs we see increased variation in CG methylation at imprint control regions (ICRs), which show both higher and lower methylation levels (Extended Data Fig. 4h).”*

Reviewer 4 - point 4: Does transient naïve treatment protocol result in epigenetically and functionally corrected hiPSC when applied for reprogramming cell types other than fibroblasts? The latter two points are critical to address in order to demonstrate the potential of the method. Whilst genome-wide profiling of additional reprogramming experiments may not be feasible, some targeted approaches (eg amplicon-based) can be realistic to perform.

Response: This is another interesting point raised by Reviewer 4. To test this, we generated an additional comprehensive set of independent iPSC lines reprogrammed from primary human dermal fibroblasts (primary HDF), keratinocytes (NHEK), mesenchymal stem cells (MSC), and our hESC-derived secondary fibroblast isogenic reprogramming system (2° Fibroblast) to comprehensively test for differences in differentiation capacity between Primed-hiPSCs and TNT-hiPSCs (Response Fig. 11a / Revised Manuscript Fig. 5a). In this series of experiments, we reprogrammed each of the four 'origin' cell types in triplicate for both TNT-hiPSCs and Primed-hiPSCs and then performed replicated differentiations of each hiPSC line into definitive endoderm, cortical neurons, skeletal muscle cells, lung epithelial cells, and neural stem cells (a

total of 24 iPSC lines, see details in response to Reviewer 2 - point 2, and Response Fig. 11a / Revised Manuscript Fig. 5a and Extended Data Fig. 13).

Firstly, regarding the *functional* correction, these differentiation experiments confirmed our previous results (from the original submission) and demonstrate that for hiPSC lines derived from primary human dermal fibroblasts, 2° Fibroblasts, MSCs, and NHEKs the TNT-hiPSCs have a higher differentiation efficiency into multiple different cell types (Response Fig. 11 / Revised Manuscript Fig. 5). Therefore, TNT reprogramming consistently improves differentiation capacity irrespective of cell type of origin.

Secondly, regarding the *epigenetic* correction, we have now performed genome-wide DNA methylation quantification by whole genome bisulfite sequencing and tested for CG-DMRs and CH-DMRs between the genetically identical Primed-hiPSCs and TNT-hiPSCs for the MSC and NHEK-derived hiPSC lines. With the new experiments included in the revised manuscript, in total the epigenetic characterization has been performed on independent iPSC lines as follows: Primary HDF: 6x Primed-hiPSC and 6x TNT-hiPSC lines; 2° Fibroblasts: 2x Primed-hiPSC and 2x TNT-hiPSC lines; MSC: 3x Primed-hiPSC and 3x TNT-hiPSC lines; NHEK: 3x Primed-hiPSC and 3x TNT-hiPSC lines. This analysis shows that in regions where Primed and TNT cells differ, the DNA methylation states of the TNT cells more closely resemble hESCs compared to Primed hiPSCs (Response Fig. 12 / Revised Manuscript Extended Data Fig. 12).

We acknowledge that the magnitude of differences in DNA methylation between NHEK- and MSC-derived Primed-hiPSCs and TNT-hiPSCs was less than some fibroblast derived iPSC lines we've profiled. As per our discussion, this could be for several reasons. For example, the MSC iPSC lines we profiled may have less epigenetic memory than dermal fibroblast-derived iPSCs, as MSCs have the capacity to differentiate into several cell types (adipocytes, muscle, chondrocytes, among others), it is possible that MSCs may not have the same persistent repressive chromatin domains of the more terminally differentiated fibroblasts. Consequently, although when compared to the conventional Primed method the TNT method still produces iPSCs with lower levels of DMRs, the magnitude of the change compared with a conventional method is not as large for MSCs. Furthermore, the timing of the TNT treatment was highly optimised based on our time course profiling of fibroblast reprogramming (Revised Manuscript Fig. 1,2). Since different cell types have different reprogramming kinetics¹⁸, it's possible that slightly different timings of the TNT treatment may be required to further optimise the method for different starting cell types, given the scale of the necessary experiments, we believe this lies outside the scope of this manuscript.

Nonetheless, our new data demonstrate that the TNT protocol does work to yield functionally and epigenetically improved hiPSCs when applied to reprogramming cell types other than fibroblasts.

Response Figure 11 / Revised Manuscript Figure 5: Multi-lineage reprogramming and differentiation confirms transient-naive-treatment reprogramming enhances epigenome resetting. **a)** Schematic representation of the experimental design of multi-lineage Primed and TNT reprogramming and differentiation into five different cell types. Upper panels show the four somatic cell lines reprogrammed into Primed-hiPSCs and TNT-hiPSCs with three independent reprogrammings performed per group, and with each subsequently differentiated into five different cell types, with independent replication. Lower panel shows the number of independent differentiation replicates performed for each origin cell type (rows) and differentiated cell type (columns). Coloured circles represent Primed-hiPSC (green), TNT-hiPSC (yellow), and hESC (grey). **b)** Quantification of the endoderm differentiation of 2^o Fibroblast-derived iPSCs, showing the proportion of resulting cells positive for FOXA2 and SOX17 by immunofluorescence (IF) analysis. **c)** Representative images for immunofluorescence analysis of FOXA2 and SOX17 in endoderm differentiation of 2^o Fibroblast-derived iPSCs. **d)** Quantification of multilineage cell differentiation in the different hiPSC lines by FACS and IF analyses: CD56/CD57/PAX6/SOX1 for cortical neuron differentiation; CD146/CD56/PAX3/PAX7 for skeletal muscle differentiation; and CD47/EPCAM/GATA6/TTF1 for lung epithelial differentiation. **e)** Representative IF analysis images of cell differentiation: Cortical neuron differentiation - SOX1 and PAX6; Skeletal muscle differentiation - PAX3 and PAX7; lung epithelial differentiation - GATA6 and TTF1. **f)** Phase contrast images taken 4 days after passaging plated embryoid bodies during differentiation into NSCs. Large stretched-out fibroblast-like cells are evident during differentiation from Primed-hiPSCs, exemplified by the red arrows. **g)** Percentage of NCAM+/FAP- cells after plating of embryoid bodies during differentiation into neural stem cells (NSCs), as measured by FACS.

Response Figure 12 / Revised Manuscript Extended Data Figure 12: TNT and Primed reprogramming of adult dermal fibroblasts, mesenchymal stem cells, and keratinocytes with DNA methylation profiling by WGBS. a) Heatmap of CG methylation levels in CG-DMRs detected for each origin cell type (HDF: primary human dermal fibroblasts; MSC: mesenchymal stem cells; NHEK: keratinocytes), with hierarchical clustering. **b)** Profile plots showing CA methylation levels in CH-DMRs where there was a significant difference detected between hiPSCs and hESCs. Upper row shows line plots for each reprogramming replicate, lower row shows replicate mean. **c)** CG-DNA methylation levels in ICRs. Grid with red squares on the right indicates if differential methylation between Primed and TNT-hiPSCs was detected using the two-sample t-test with $p < 0.05$. ICRs defined in¹⁵.

Reviewer 4 - point 5: Are TNT-hiPSC genetically normal? This is critical for demonstration of clinical applicability of the method.

Response: Yes, TNT-hiPSCs appear to be genetically normal. To address this question, we tested for copy number variations by whole-genome SNP genotyping of Primed, TNT, and NtP hiPSCs. Our results indicate that there are no insertions or deletions for the TNT-hiPSC lines analysed, however extended culturing in naive media for the NtP-hiPSCs did indeed result in megabase-scale deletions in both NtP lines analysed. In this analysis, we also detected a smaller 607 kb deletion in one Primed-hiPSC line (Response Fig. 13 / Revised Manuscript Extended Data Fig. 3b). These results are consistent with previous reports from others and our lab¹⁴⁻¹⁶ that observed that extended culturing in naive media results in genetic instability. As mentioned above, it is for this reason that we believe the TNT method is superior to the NtP or PNP methods.

Response Figure 13 / Revised Manuscript Extended Data Figure 3b: Summary plot of copy number variation (CNV) analysis performed using Illumina 650k arrays. Left grid plot indicates the samples and chromosomes where CNVs were detected. Right plots show B allele frequency (BAF) and log R ratio (LRR) for samples where a CNV was detected, with each datapoint representing variant sites.

Reviewer 4 - point 6: The second important claim of the manuscript is the correction of the differentiation potential of TNT-hiPSC. This section of work presented only minimal data and requires much deeper assessment to support the conclusion and to demonstrate robustness of the result. It is also critical to provide a link between the epigenetic correction and the improved functional qualities, as claimed in the manuscript. The specific questions on this section are among the listed below.

Response: Regarding differentiation potential, as suggested (and detailed in our response to Reviewer 4 - point 4, above) we have now included an extensive assessment of the differentiation potential into different cell types and lineages, using an extended set of differentiation assays and multiple independent methods to assess differentiation efficiency. This series of experiments confirms and demonstrates the robustness of our results, which shows that TNT reprogramming consistently improves differentiation capacity irrespective of cell type of origin (Revised Manuscript Fig. 5 and Extended Data Figs. 12, 13, 14, 15). Regarding a link between the epigenetic correction and the improved functional qualities, we also now further investigate the presence of aberrant DNA methylation states in regulatory regions and their potential effects on expression (see Reviewer 4 - point 16, below).

Reviewer 4 - point 7: As a general remark, the results overall seem reproducible and are mostly presented in several replicates. Nevertheless, it is not always clear what variables are controlled by the replicates (eg independent differentiation experiments of the same hiPSC line vs independent differentiation experiments using hiPSC lines from independent reprogramming rounds). Please make sure that critical results are reproduced in fully independent experiments and include this information in the manuscript. The statistics is overall well presented, however occasionally missed (please see specific comments below).

Response: Thank you for this comment, we have revised the manuscript to address this point. In the revised manuscript we have ensured that this information is clearly described in the methods and figure legends. Importantly, as detailed in the revised manuscript, this shows that the critical results that we report have been reproduced in fully independent experiments (Response Fig. 1a / Revised Manuscript Fig. 5a and Extended Data Fig. 13).

To ensure this aspect of the manuscript is clear we have generated tables and figures (Response Fig. 1a / Revised Manuscript Fig. 5a, Extended Data Fig. 13, and Supplementary Table 1) summarising all the hiPSC lines and independent reprogramming replicates that have been generated and used in the differentiation experiments.

Reviewer 4 - point 8: Overall, if the comments are addressed and the differentiation section is strengthened, the manuscript will suit for the publication in Nature journal. The presented method has the potential to open up new avenues in translational research and future applications and will be of significant interest for the field.

Response: Thank you for such encouraging and supportive words. We hope that Reviewer 4 finds that we have endeavoured to thoroughly address all the questions and suggestions in the revised manuscript.

Major specific comments

Reviewer 4 - point 9: 1. Related to Figure 1 and Extended Data Figure 1: has the cell sorting strategy to enrich reprogramming intermediates been validated? In particular, is there evidence that the cells sorted on different days of the reprogramming process indeed represent a continuum on a single trajectory? How heterogeneous are these cell populations? This is critical for the analysis of the DNA methylation and gene expression dynamics.

Response: Yes, this strategy has been extensively validated in our previous study¹⁹ (see Supplementary Figure 1 from Liu et al., Nature 2020¹⁹, and Response Fig. 14 below). In brief, using single cell data we derived a FACS sorting strategy to isolate intermediate populations, using the cell surface markers detailed in Response Table 1 and Supplementary Table 1 of this response/manuscript. These intermediate populations were then extensively validated for their capacity to give rise to the subsequent intermediate populations as well as iPSCs (see Response Fig. 14). Therefore, yes this demonstrates that the isolated cell populations from different days of reprogramming do indeed represent true reprogramming cell intermediates. In this manuscript we used the same strategy to purify cell populations for DNA methylation and gene expression analyses described in this study. Furthermore, as demonstrated in our previous study¹⁹ and shown below, these are highly pure populations.

Response Figure 14: Isolation of reprogramming intermediates using a panel of cell surface markers (from Liu *et al.*, Nature 2020, Supplementary Figure 1). Flow cytometry analysis of TRA-1-60 vs SSEA3, EPCAM vs SSEA3 over the reprogramming time-course into primed and naive induced pluripotency. b, Validation of the panel of cell surface markers for isolation of intermediates that carry the reprogramming potential across the reprogramming time-course. e.g., CD13+F11R- and CD13+F11R+ subpopulations were isolated on day 3 and reseeded for 5 days (for flow cytometry reanalysis) and for hiPSCs colony formation (AP staining), see Methods for details. Of note, SSEA3+EPCAM+ population on day 13 of NR showed negative for AP staining due to the substantial differentiation of these cells after reseeding. FM: Fibroblasts Medium (black); PR: Primed Reprogramming (orange); NR: Naive Reprogramming (blue).

Response Table 1: Panel of cell surface markers used to purify reprogramming intermediate populations.

Samples	Marker profile
Day 0 HDFa	CD13+
Day 3 intermediates	CD13+F11R+
Day 7 intermediates	CD13-F11R+TRA-1-60+
D13 Primed intermediates	CD13-F11R+TRA-1-60+ SSEA3+ EpCAM-
D21 Primed intermediates	CD13-F11R+TRA-1-60+ SSEA3+ EpCAM+
P3 Primed intermediates	CD13-F11R+TRA-1-60+ SSEA3+ EpCAM+
P10 Primed iPSCs	CD13-F11R+TRA-1-60+ SSEA3+ EpCAM+
D13 t2iLGoY naïve intermediates	CD13-F11R+TRA-1-60+ SSEA3- EpCAM+
D21 t2iLGoY naïve intermediates	CD13-F11R+TRA-1-60+ SSEA3- EpCAM+
P3 t2iLGoY naïve intermediates	CD13-F11R+TRA-1-60+ SSEA3- EpCAM+
P10 t2iLGoY naïve iPSCs	CD13-F11R+TRA-1-60+ SSEA3- EpCAM+

Reviewer 4 - point 10: 2. Related to Figure 1e: Please provide more detailed characterisation of the genes clusters, including details of gene ontology terms enrichment and examples of specific genes.

Response: We have now performed ontology analyses for the clusters presented in Fig. 1e and have included the full results in Extended Data Table 2 of the revised manuscript. As suggested, we have now also included a new figure panel exemplifying genes for each cluster, which is now included as Extended Data Fig. 1g in the revised manuscript, as shown in Response Fig. 15.

Response Figure 15 / Revised Manuscript Extended Data Figure 1g: Example genes that are assigned to DNA methylation clusters in Fig. 1e.

Reviewer 4 - point 11: 3. Related to Figure 1 and Extended Data Figure 1, lines 179-181: “the first wave of epigenetic remodelling at regulatory elements is driven by OKSM reprogramming factors, followed by distinct methylation states consolidated by the signalling cues in primed and naive conditions”. What signalling cues consolidate the distinct methylation states? What is the evidence for this statement?

Response: Indeed, upon considering this comment, we agree with Reviewer 4 that this statement is not well supported by the data we present. Therefore, we have rephrased this section in the revised manuscript. The revised sentence now reads:

“Taken together, these genome-scale DNA methylation roadmaps of human primed and naive reprogramming reveal that the first wave of epigenetic remodelling at regulatory elements appears to be driven by OKSM reprogramming factors, followed by distinct methylation states coincident with transitioning to primed and naive culture conditions.”

Reviewer 4 - point 12: 4. Related to Figure 2 and Extended Data Figure 2, lines 207-208 (“Despite these DMRs being reproducible across lines, research groups, and reprogramming methods...”): how are the DMRs identified in the study comparable to the DMRs in other works (eg cited in the manuscript), in terms of their numbers, specific locations, co-localisation with H3K9me3, direction of change?

Response: This is an important point, albeit not straightforward to unambiguously address. To address this comment, in the revised manuscript we have included a new comprehensive

re-analysis of previously published data on the differences between hiPSCs and hESCs. We performed this more comprehensive analysis to address several complications in comparing DMRs from different studies as published, which include:

1. Different DMR detection methods: DMRs are statistical heuristics and are not discrete biological units like transcripts. This is why DMRs identified using different software can vary widely, and are influenced by specific parameters used in each study. The number of DMRs detected in each study can be influenced by the DMR detection method, number of replicates, sequencing depth, statistical threshold, and chosen parameters (e.g. searching for enhancer-sized DMRs with a cutoff of 10% methylation difference, versus other length/magnitude thresholds). As many of these parameters can have a larger impact on the results than the biological treatment, it is difficult to draw meaningful conclusions from comparing DMR counts and overlaps.
2. Different cell lines likely have differing levels of epigenetic aberration and memory, and it is common for different hESC lines to have been used as comparison samples. These could be more, fewer, or different loci, which can be influenced by different genetic backgrounds and different culture conditions.
3. Different reference genomes. For example, Lister *et al.* 2011 (*Nature*)¹ used hg18 while Ma *et al.* 2014 (*Nature*)³ used hg19, and genome liftover methods are not ideal for DNA methylation data.

To compare previous studies in a fair but robust way, we obtained the raw data from Lister *et al.* 2011¹ and Ma *et al.* 2014³, who both compared hiPSCs and hESCs by WGBS. We processed these data using the exact analysis pipeline used in this manuscript. What this analysis revealed is that even when processed with identical methods, the specific locations of CG-DMRs vary substantially for each study (Response Fig. 16a / Revised Manuscript Extended Data Fig. 5a - see response to Reviewer 2 - point 3). However, the enrichment of CG-DMRs in fibroblast repressive chromatin domains is largely similar between this study and published studies (Response Fig. 16c / Revised Manuscript Extended Data Fig. 5c). As each study used different cell culture media (technical variation) and cell lines with different genetic backgrounds (biological variation), with different sample sizes and sequencing depth (power and sensitivity), we performed a principal component analysis of CG methylation levels for all DMR sets combined, in an attempt to dissect out the common signal that separates hiPSCs from hESCs. This revealed that principal component 3 (PC3) clearly separates Primed-hiPSCs and hESCs (Response Fig. 16 g-i / Revised Manuscript Extended Data Fig. 5g-i). Moreover, TNT-hiPSCs are more similar to hESCs for PC3 eigenvalues. This indicates that when focusing on the CG-DMRs that are most strongly influenced by study, TNT-hiPSCs are distinct from Primed-hiPSCs across all studies assessed. Additionally, we see a consistent enrichment of CG-DMRs within the large CH-DMRs, including in the reanalysed data from Lister *et al.* 2011¹ and Ma *et al.* 2014³ (Response Fig. 16d,f / Revised Manuscript Extended Data Fig. 5d,f). This appears to reflect an overall lower level of CG methylation in CH-DMRs for hiPSCs that resembles the partially methylated state the fibroblasts exhibit in large repressive

heterochromatin domains. We see this as the primary reason that there is very little direct overlap between CG-DMR coordinates between studies, as these CG-DMRs often seem not to be indicative of discrete regulatory elements with relatively strict boundaries. Rather, they are commonly indicative of larger regions that have reduced CG DNA methylation. For example, when inspecting the CH-DMR that contains the TCERG1L locus, we observe distinct clustering of many Primed-hiPSC CG-DMRs within the CH-DMR for our data, as well as for Primed-hiPSC CG-DMRs from the Lister *et al.* (2011)¹ and Ma *et al.* (2014)³ studies (Response Fig. 16f / Revised Manuscript Extended Data Fig. 5f). Moreover, when considering the larger CH-DMR features in Primed-hiPSCs, we see a considerable degree of overlap of these regions between different studies (Response Fig. 16e,f / Revised Manuscript Extended Data Fig. 5e,f). Together, these observations further support our hypothesis that epigenetic memory in hiPSCs is at least in part due to the retention of large repressive chromatin domains from the somatic cells from which the hiPSCs were reprogrammed.

a CG-DMR overlaps for this study and published studies

e CA methylation in CH-DMRs for this study and published studies

f CG-DMR clustering within CH-DMRs and fibroblast repressive chromatin

c CG-DMR enrichment in cell-type specific repressive chromatin domains

d Enrichment of CG-DMRs and regulatory elements in CH-DMRs

g Principal component analysis of union CG-DMRs for all studies combined

h Combined CG-DMR mCG/CG principal components for all study datasets and hESC/hiPSC group

i Combined CG-DMR mCG/CG principal components and relationship with this study and published studies

Response Figure 16 / Revised Manuscript Extended Data Figure 5: Comparison of CG and CH DMRs across studies. **a)** Upset plot shows number of CG-DMRs detected for this study and how they overlap with CG-DMRs detected from previously published data processed using identical methods. **b)** Difference in DNA methylation level between hiPSCs and hESCs at CG-DMRs identified between Primed-hiPSCs and hESCs. Vertical dashed lines indicate the threshold of 20% minimum difference in CG DNA methylation level at CG-DMRs. **c)** Enrichment z-score determined from permutation testing of enrichment of CG-DMRs in repressive chromatin domains and of **d)** CH-DMRs in published studies. **e)** Heatmap of CA methylation levels in CH-DMRs in this study and previously published studies showing Primed-hiPSCs from all studies clustering separately to hESCs. **f)** Genome track of a CH-DMR region that intersects a PMD, fibroblast lamina associated domain (LAD), and clusters of CG-DMRs in each study. **g)** Principal component analysis of CG methylation levels in CG-DMRs for all studies combined. Top left plot shows the proportion of variance explained by each principal component. Scatter plots with coloured points show principal component separation of hESCs, Primed-hiPSCs, and TNT-hiPSCs. Ellipses around points indicate 95% confidence interval for a multivariate t-distribution. These data indicate that principal component 3 (PC3) in the bottom left plot clearly separates Primed-hiPSCs and hESCs for all studies, and shows that TNT-hiPSCs are more similar to hESCs by this measure. **h)** Plots of eigenvalues for each principal component for Primed-hiPSCs, TNT-hiPSCs, and hESCs, and **i)** data split by study/lab. Red bars indicate $P < 0.05$ for ANOVA test, with FDR reported above red bars.

Reviewer 4 - point 13: 5. Related to Figure 2 and Extended Data Figure 2: published evidence suggests that continuous passaging of (mouse) iPSC attenuates epigenetic differences between iPSC and ESC (Polo *et al.*, 2010). Are epigenetic aberrations of primed hiPSC corrected over continuous passaging?

Response: Indeed, Polo *et al.* (2010)²⁰ showed that continuous passaging can reduce epigenetic differences between mouse iPSCs and ESCs. In contrast, Lister *et al.* (2011)¹ showed that human iPSCs at passage 32, 33, 34, and 65 still featured epigenetic aberrations. Here, we show that Primed-hiPSCs up to passage 26 for primary reprogramming, and passage 33 for secondary reprogramming, still feature epigenetic aberrations. Based on these data, we find no support for attenuation of epigenetic aberrations in human iPSCs by extended passaging to anything near the level we observe for TNT-hiPSCs.

It is important to note here that the mouse iPSCs in Polo *et al.* (2010)²⁰ were cultured in serum+LIF which results in cells that are in a much more naive state, and likely underpins the attenuation of epigenetic differences with extended passaging. Serum+LIF is a substantially different media to what is used to culture human iPSCs in the primed state (i.e. E8 or KSR), where epigenetic aberrations are observed.

Reviewer 4 - point 14: 6. Related to Figure 2 and Extended Data Figure 2, lines 248-250 (“loss of imprinting in naive media occurs through extended culturing, and is not as rapid as the initial global loss of CG methylation that occurs prior to day 13 of naive reprogramming”): a substantial loss of imprinting (from about 50-60% down to 30-35%) occurs by day 13 of the naïve reprogramming (Figure 2f, Extended Data Figure 2d) and therefore the notion is an overstatement.

Response: We agree with this, and have revised this summary sentence accordingly. This statement now reads:

“This indicates that the loss of DNA methylation at imprinted loci becomes more extensive the longer cells are cultured in naive conditions, and suggests some imprints may still be maintained at day 13 of naive reprogramming (Fig. 1b).”

Reviewer 4 - point 15: 7. Figure 3e: what is the reason for over-representation of the corrected CG-DMRs within K9me3 and fibroblast PMD and LAD? Is the correction more efficient in these regions? Or are the CG-DMRs more abundant in these regions in general? This is not very clear from the text, please explain this analysis.

Response: We apologise if we were unclear in explaining these analyses and characteristics of what we observe, especially as these conclusions directly relate to our working model of how epigenetic memory is maintained and corrected. Throughout our manuscript, we show several independent analyses that show that epigenetic memory occurs frequently in large domains of fibroblast-associated repressive chromatin marked by H3K9me3 and depletion of CH methylation (Revised Manuscript Fig. 3e,f,g,h,i,o). With respect to *“is correction more efficient in these regions?”*, yes, our data do indicate that correction is more efficient in fibroblast-associated repressive chromatin domains, and this is independent of the numbers of corrected and uncorrected DMRs as we accounted for these differences in our enrichment tests presented in Fig 3e. Furthermore, yes, the CG-DMRs do have a higher relative abundance in these regions (see Revised Manuscript Fig. 3e and Extended Data Fig. 4a-c).

In the analysis presented in Fig. 3e of the revised manuscript, we tested if CG-DMRs show a statistical under or over-representation in defined genomic regions. We used permutation tests to calculate how many overlaps CG-DMRs have with, for example, fibroblast-specific H3K9me3 regions compared to randomly selected regions (permuted 200 times). To do this, we used the *regionR* bioconductor package that uses a randomisation approach that implicitly takes into account the complexity of the genome without the need of assuming an underlying statistical model²¹. This approach addresses the problem of simply comparing the percentage of overlapping DMRs, as one does not know how many of those occur by chance alone, and makes it challenging to evaluate regions like H3K9me3 domains that can cover a large proportion of the genome. The Z-scores from the permutation testing are a measure of the strength of the association, and is defined as the distance between the expected value and the observed one, measured in standard deviations. For example, a Z-score of +25 would indicate that the number of overlaps for CG-DMRs is 25 standard deviations higher than one would expect by chance.

Regarding ‘the reason’ for over-representation of corrected CG-DMRs in fibroblast-associated repressive chromatin (H3K9me3, PMD, LADs), we believe the totality of our data suggests this is because the TNT method enables large-scale reorganisation of repressive chromatin, and this reorganisation is reflected in DNA methylation changes in these regions, which are represented by the corrected CG-DMRs in the revised manuscript Fig. 3e and Extended Data Fig. 4a-c. From this standpoint, our data do indicate that correction is indeed more efficient in repressive chromatin regions. We emphasise, however, that we draw this reasoning from

several other of our analyses (Fig. 3f-i) that stem from the one presented in Fig. 3e, as these analyses in Fig. 3e were the catalyst for us to pursue this observation further.

We have now revised the text surrounding Fig. 3e as follows:

In our original submission we stated:

“We then performed permutation testing to identify regions of the genome that have over- or under-representation of CG-DMRs, revealing that corrected CG-DMRs are highly enriched (Z-score=38.9; FDR<0.01) in regions featuring the repressive histone modification H3K9me3 in fibroblast cells, but significantly depleted in regions of hESC-specific H3K9me3 (Z-score=-4.5; FDR<0.01; Fig. 3e, Extended Data Fig. 4a). Consistently, corrected CG-DMRs were over-represented (Fig. 3e) in fibroblast partially methylated domains (PMDs; Z-score=25.8; FDR<0.01) and nuclear lamina associated domains (LADs; Z-score=10.6; FDR<0.01), which are known to co-occur with H3K9me3 in large domains of heterochromatin that are gene poor, repressive, late replicating, and relate to chromatin topology and genome architecture^{22,23}.”

In our revised submission, we have attempted to explain this analysis more clearly, and state:

“We then used permutation testing to identify which genomic features show a statistical over- or under-representation of CG-DMRs, measured using Z-scores. Here, the Z-scores are a measure of the strength of the association, and are defined as the distance between the expected and observed values, measured in standard deviations. This revealed that corrected CG-DMRs are highly enriched (Z-score=38.9; FDR<0.01) in regions featuring the repressive histone modification H3K9me3 in fibroblast cells, but significantly depleted in regions of hESC-specific H3K9me3 (Z-score=-4.5; FDR<0.01; Fig. 3e, Extended Data Fig. 4a). Consistently, corrected CG-DMRs were over-represented (Fig. 3e) in fibroblast partially methylated domains (PMDs; Z-score=25.8; FDR<0.01) and nuclear lamina associated domains (LADs; Z-score=10.6; FDR<0.01), which are known to co-occur with H3K9me3 in large domains of heterochromatin that are gene poor, repressive, late replicating, and relate to chromatin topology and genome architecture”

In the methods section of our original submission, we stated:

“To perform association analysis of genomic regions we performed permutation tests calculate enrichment of genomic elements with elements obtained from the GeneHancer database²⁹; ultra-conserved elements (UCE) as defined in⁸²; repeat elements as defined by UCSC repeat masker for hg19; fibroblast partially methylated domains calculated for day_0 fibroblasts with MethylSeeker⁸³; Promoters defined as 2 kb upstream and 500 bases downstream of TSS as defined in UCSC genes; Exons and introns as defined in UCSC genes; Lamina associated domains (LADs) for fibroblasts (4DNFIUIDLJJI) and H1 ESCs (4DNFIP6N54B3) as defined by 4D nucleome project for hg38 and lifted over to hg19 coordinates⁸⁴. H3K9me3 peaks were retrieved from the ENCODE database for fibroblasts (ENCF963GBQ) and hESCs (ENCF001SUW)⁸⁵.

Constitutive regions for LADs or H3K9me3 were defined as those regions where peaks intersected for both fibroblasts and hESCs.”

In our revised manuscript, we have appended the following text to this section of the methods:

“In these enrichment analyses, the permutation tests calculate how many overlaps the features of interest (i.e. CG-DMRs) have, for example, with fibroblast-specific H3K9me3 regions compared to randomly selected regions, and permuted 200 times. This approach addresses the problem of simply comparing the percentage of overlaps, as one does not know how many of those occur by chance. The Z-scores from the permutation testing are a measure of the strength of the association, and is defined as the distance between the expected value and the observed one, measured in standard deviations. For example, a Z-score of +25 would indicate that the number of overlaps is 25 standard deviations higher than one would expect by chance.”

Reviewer 4 - point 16: 8. Extended Figure 4a: uncorrected CG-DMRs are enriched in transcriptionally functional regions (promoters, exons, enhancers). Is there a functional consequence or significance of the methylation state in these regions? What genes are affected by these regions and is their expression affected?

Response: To address this question regarding the functional consequence of CG-DMRs in functional regions, we used our genetically matched hESC/hiPSC secondary reprogramming system to analyse CG-DMRs intersecting promoters (as we can unambiguously assign these regions one-to-one to genes) and then inspected gene expression fold change for the associated genes. We view this as the most controlled way to test for functional consequence as we can simultaneously test for hESC/hiPSC methylation and gene expression differences.

These results show that uncorrected promoter CG-DMR methylation is associated with gene expression differences in a subset of the cases (Response Fig. 17 / Revised Manuscript Extended Data Fig. 8b), and this is most pronounced with Primed-hiPSCs. For Primed-hiPSCs, 172/547 (31.4%) of the CG-DMRs intersecting promoters show differential expression compared to the matched hESCs, compared to 49/215 (22.7%) of promoter CG-DMRs in TNT-hiPSCs had corresponding gene expression differences compared to the matched hESCs (Response Fig. 17 / Extended Data Fig. 8b). Therefore, TNT-hiPSCs have fewer promoter DMRs and also feature less differential expression linked to these promoters. In the revised manuscript we have now included Supplementary Table 7 that list the genes that are affected by these regions and whether their expression is affected.

Response Figure 17 / Revised Manuscript Extended Data Figure 8b: Barplots show the number of CG-DMRs that intersect promoters, for CG-DMRs detected in hiPSCs compared to hESCs. Colours indicate the proportion of genes linked to promoters that show significant differential expression (FDR < 0.05, log₂FC > 1). Scatter plots show the relationship between promoter DNA methylation differences between hiPSCs and hESCs (x-axis) and gene expression differences (y-axis). Individual points indicate DMR-gene pairs, with point colours indicating if the gene was differentially expressed.

Reviewer 4 - point 17: 9. Figure 3f, lines 315-318, “Regions enriched for H3K9me3 in fibroblasts that intersect CG-DMRs showed higher H3K9me3 in Primed-hiPSC compared to TNT-hiPSC (...) and NtP-hiPSC (...), which were both more similar to hESC”): please provide statistics confirming the similarity of TNT-hiPSCs and NtP-hiPSCs to hESCs.

Response: We have now restructured the statistical tests of H3K9me3 enrichment in CG-DMRs to include comparisons of hiPSC groups to hESCs. We have added the results to Fig. 3f in the revised manuscript (Response Fig. 18 / Revised Manuscript Fig. 3f). These results confirm that TNT-hiPSCs have significantly less H3K9me3 in corrected CG-DMRs compared to Primed-hiPSCs, with TNT-hiPSCs not showing a statistical difference to hESCs (p=0.087).

Reviewer 4 - point 18: 10. Figure 3j and Extended Figure 4f, lines 357-358 (“This is in contrast to NtP-hiPSCs, where we observed more extensive loss of imprinting or disordered gain of methylation”): please provide evidence that NtP-hiPSCs gain methylation in ICR in a disordered fashion. Is it a gain of methylation or maintenance of methylation inherited from fibroblasts? Furthermore, among NtP-hiPSC lines, 32F but not 38F lost methylation in ICRs. Could you please provide wider statistic comparing TNT and NtP, what is the frequency of methylation state maintenance in the ICRs?

Response: Regarding the first part of this comment from Reviewer 4, we acknowledge that we do not show evidence of ‘disordered gain’ of methylation in ICRs for NtP-hiPSCs. We thank Reviewer 4 for pointing out our inaccurate description. We have now revised this section which now reads: “*This is in contrast to NtP-hiPSCs, where we observed increased variance in the methylation change at imprinted loci (Fig. 3j, Extended Data Fig. 6a).*”

Regarding the second point about providing a statistical comparison between TNT and NtP, we have now included statistical tests for each imprint control region, and a frequency count of how many ICRs are significantly different to the fibroblasts (Response Fig. 19 / Revised Manuscript Extended Data Fig. 6a). Overall, the data indicate that TNT-hiPSCs do not show an increase in loss of imprinting over Primed-hiPSCs, in contrast to NtP-hiPSCs.

Response Figure 19 / Revised Manuscript Extended Data Figure 6a: CG methylation in imprint control regions (ICRs) for fibroblasts and hiPSCs reprogrammed from these fibroblasts. Right grid shows which hiPSC groups had significantly different (t-test FDR < 0.05) CG methylation levels compared to fibroblasts. The data indicate that TNT-hiPSCs do not show an increase in loss of imprinting over Primed-hiPSCs, in contrast to NtP-hiPSCs.

Reviewer 4 - point 19: 11. Figure 3l-m: cultures obtained from Primed-hiPSCs in the conditions for NSC differentiation predominantly contained mesodermal cells, whereas TNT- and NtP-hiPSCs-derived cultures were mostly composed of NSCs – how consistent is this outcome of NSC differentiation? Whilst single-cell RNAseq does not seem feasible to do on multiple samples, qRT-PCR or flow cytometry for markers can be indicative. Please provide statistics for multiple cells lines and independent experiments.

Response: In the revised manuscript, we provide new results from extensive and highly replicated additional reprogramming and differentiation experiments. We have now repeated the neuronal stem cell differentiation experiments with Primed-hiPSCs and TNT-hiPSCs reprogrammed from primary fibroblasts, secondary fibroblasts, mesenchymal stem cells, and keratinocytes. These experiments were performed with three independent reprogramming-to-hiPSC experiments per reprogramming group (TNT/Primed). As suggested, to quantify the differentiation efficiencies, we performed flow cytometry, and measured the percentage of NCAM+ / FAP- cells for three differentiations per reprogramming experiment (i.e. per independently reprogrammed iPSC line), resulting in nine experimental replicates per group. NCAM was used as a marker for NSCs while FAP was enriched in fibroblast-like cells in the scRNA-seq data. We also differentiated two hESC lines into NSCs to get an expected reference range, albeit from different genetic backgrounds. This series of experiments showed that NSCs differentiated TNT-hiPSCs consistently yielded a higher percentage of NSCs than Primed-hiPSCs (Response Fig. 20 / Revised Manuscript Fig. 5f,g). We believe the results from these biologically and technically replicated experiments further substantiate our claims and attest to the robustness of our findings.

Response Figure 20 / Revised Manuscript Figure 5: **f)** Phase contrast images taken 4 days after passaging plated embryoid bodies during differentiation into NSCs. Large stretched-out fibroblast-like cells are evident during differentiation from Primed-hiPSCs, exemplified by the red arrows. **g)** Percentage of NCAM+/FAP- cells after plating of embryoid bodies during differentiation into neural stem cells (NSCs), as measured by FACS.

Reviewer 4 - point 20: 12. Extended Figures 3c and 6g; Figure 3f and Extended Figure 6f: the figures show independent experiments using Primed, TNT and NtP protocols either starting from primary fibroblasts or hESC-derived fibroblasts. Was the epigenetic correction consistent in these two experiments? Have the memory regions and the corrected regions coincided?

Response: From the standpoint of what regions have epigenetic correction, we observe strong qualitative consistency in correction. The types of regions largely coincide between the two experiments. Specifically, the CG-DMRs are enriched in fibroblast-associated repressive chromatin (Fig. 3e), and from this standpoint the results for the CG-DMRs from these two

different systems are largely consistent. When counting CG-DMR overlaps between the experiment sets (primary fibroblast CG-DMRs vs secondary fibroblast CG-DMRs), we observe that 42% of secondary fibroblast DMRs are also found in at least one primary fibroblast CG-DMR set (Extended Data Fig. 7h). However, we would like to emphasise (as we expand upon in response to Reviewer 2 - point 3, above, Extended Data Fig. 5) that the exact coordinates of CG-DMR loci can differ as expected, and that some of these differences can be attributed to batch and genetic background (Extended Data Fig. 5). Furthermore, we see that the large regions of repressive chromatin that repeatedly show differential methylation in the CH context (CH-DMRs, e.g. Fig. 3g and Fig. 4e) are observed in independent Primed-iPSC lines derived from primary or secondary fibroblasts, and this aberrant CH methylation state is corrected in the corresponding independent TNT-iPSC lines when profiling identical regions (also discussed in detail in response to Reviewer 2 - point 3, above).

Reviewer 4 - point 21: 13. Supplementary Table 5, lines 467-470 (“Gene ontology analyses revealed that Primed-hiPSCs over-express 113 genes involved in mesoderm development ($p=0.01$, GO:0007498, Supplementary Table 5), further demonstrating the cell-of-origin molecular memory in Primed-hiPSCs is corrected by reprogramming through the naive state.”): Supplementary Table 5 reveals that Primed-hiPSC also overexpress genes involved in epithelial cell differentiation (56 genes, $p=3.77E-09$), central nervous system development (77 genes, $p=1.15E-08$), skeletal development, muscle development, brain development, neurogenesis etc, all more significantly enriched than the “mesoderm development” GO term. What is the evidence that the gene expression differences are related to the epigenetic memory reflecting the fibroblast origin of these hiPSC?

Response: Firstly, we would like to acknowledge that this comment from Reviewer 4 has highlighted a minor error in our initial submission. Where we state “113 genes” in the above quoted sentence, this actually refers to the number of genes assigned to that ontology term, not the number of differentially expressed genes. The correct numbers of differentially expressed genes for each ontology term were correctly reported in the referenced Supplementary Table 5. We also acknowledge that many other gene ontology terms beyond the expected mesoderm development networks were also enriched when testing for genes more highly expressed in Primed-hiPSCs, as is often the case with ontology enrichment tests. We view the enrichment of mesoderm development associated genes as one piece of supportive evidence that compliments our other independent analyses (described below) that indicate Primed-hiPSCs retain expression reminiscent of their fibroblast origin.

In order to answer “What is the evidence that the gene expression differences are related to the epigenetic memory reflecting the fibroblast origin of these hiPSC?”, we took all differentially expressed genes with fibroblast-associated GO terms and then plotted their expression for hESC, hiPSCs, and fibroblasts (Response Fig. 21c / Revised Manuscript Extended Data Fig. 8c). This analysis showed that, based on expression of these genes, Primed-hiPSCs cluster most closely with fibroblasts, and that although for many genes Primed-hiPSCs show lower expression than fibroblasts, their expression was more directionally similar to fibroblasts than to TNT-hiPSCs for mesoderm-related genes. Although some of these mesoderm-associated genes showed relatively low expression in the fibroblasts, early mesoderm genes such as *fibroblast growth factor 8 (FGF8)* were elevated in Primed-hiPSCs

compared to hESCs and TNT-hiPSCs. We then plotted early mesoderm differentiation markers for Wnt signalling and the intermediate mesoderm progenitor markers *BMP4*, *MESP1*²⁴ and *FOXC1*²⁵, which shows that Primed-hiPSCs have increased expression of these markers compared to hESCs, and that this is largely corrected in TNT-hiPSCs (Response Fig. 21d / Revised Manuscript Extended Data Fig. 8d). Together, these results provide further evidence supporting the claim that the Primed-hiPSCs retain a mesoderm gene expression signature while in a pluripotent state, which is corrected in TNT-hiPSCs.

Response Figure 21 / Revised Manuscript Extended Data Figure 8c-d: **c)** Heatmap showing clustered standardised gene expression values for differentially expressed genes with fibroblast-associated gene ontology terms. **d)** Gene expression levels for early mesoderm lineage genes. Grey points represent individual samples and black points represent sample means with whiskers showing range. **e)** Gene expression heatmap of fibroblast-specific genes with retained expression in Primed-hiPSCs.

Reviewer 4 - point 22: 14. Extended Figure 7a, Supplementary Table 5, related to RNAseq analysis: what fibroblast-specific genes retain their expression in Primed-hiPSC? Please provide more detailed characterisation of DE genes. This is important to support the point that the observed aberrations are due to epigenetic memory of the origin and not artifacts of reprogramming.

Response: In addition to our response to question 13 above (Reviewer 4 - point 21), to specifically address the question “*what fibroblast-specific genes retain their expression in Primed-hiPSCs*”, we analysed the genes that were differentially expressed and higher in

fibroblasts vs hESCs (fibroblast-specific) while also being not differentially expressed between Primed-hiPSCs and fibroblasts (expression retained in Primed hiPSCs). Using this gene set, we clustered the samples based on their expression levels of these genes, and plotted a gene expression heatmap to visualise these results (Response Fig. 22e / Revised Manuscript Extended Data Fig. 8e). This analysis shows that Primed-hiPSCs feature a gene expression signature that resembles the fibroblast state, in contrast to the TNT- and NtP-hiPSC lines analysed. This supports our conclusion that some gene expression differences in Primed-hiPSCs compared to hESCs are due to epigenetic memory. The analysis in response to question 13 above (Reviewer 4 - point 21) also highlights that some gene expression differences in Primed-hiPSCs are also at an intermediate level between hESCs and fibroblasts, and therefore also indicative of epigenetic memory.

Nevertheless, we would like to clarify that we do not claim that all the observed aberrations are only due to epigenetic memory of the cell of origin rather than artefacts of reprogramming. Indeed as reported in our manuscript (Fig. 2) and past studies (e.g. Lister *et al.* 2011¹), we observe both memory DMRs that have a state similar to the cell of origin, as well as a much smaller set of iPSC-specific DMRs that are induced by the reprogramming process. Importantly, both of these classes of aberrant epigenetic states can be corrected by the TNT method, as shown in Extended Data Fig. 3d,f in the revised manuscript.

Reviewer 4 - point 23: 15. Figure 4b, c, Extended Figure 7b: NtP-hiPSCs show significant differences in their ATACseq profile from hESCs, which contrasts with their DNA methylation

state that is similar to hESCs. What is the reason for these differences? What regulatory regions are affected, what TF binding motifs are enriched in the differential peaks?

Response: We believe the possible reason that the ATAC-seq profiles for two of the three NtP-hiPSC samples shows some differences to the hESCs, in contrast to what we see for DNA methylation and gene expression with respect to similarity to the hESCs, is due to these two NtP-hiPSC ATAC-seq replicates having been made from a frozen stock of cells. Unfortunately, technical failure in the first attempt to generate the ATAC-seq libraries for these two NtP-hiPSC samples meant that we had to go back to cryopreserved stocks of these two NtP-hiPSC lines, and thaw and passage them to obtain enough cells for the repeat of the ATAC-seq library generation. No other samples were affected as we adjusted our methodology before processing them. For transparency, we kept all the data in our manuscript. As these NtP-hiPSC samples are concordant with the TNT-hiPSCs and hESCs for DNA methylation, gene expression, and H3K9me3, we believe these differences are due to the additional freeze-thaw and passaging for the ATAC-seq. We have now made a comment in the methods of the revised manuscript that states:

“Although we observed differences in ATAC-seq peak counts for NtP-hiPSCs that were not consistent with DNA methylation or gene expression for two outlier samples (Fig. 4b-d), we believe this is due to an additional freeze-thaw cycle for the ATAC-seq samples, and the extended recovery of these two replicates which required 2 additional passages.”

Regarding the second point about affected regulatory regions and enriched TF binding motifs in the differential peaks, we have now added a more detailed analysis of the ATAC-seq data, which includes analyses of motif enrichment as suggested (Response Fig. 23 / Revised Manuscript Extended Data Fig. 9b,c). Together, these new analyses show that Primed-hiPSCs feature accessible chromatin (compared to hESCs and TNT-iPSCs) in regions that have motifs for TFs associated with differentiation (like the forkhead FOXP TFs), and lack accessibility in regions featuring OKSM motifs.

Response Figure 23 / Revised Manuscript Extended Data Figure 9: a) MA plots and b) heatmap representation of differential ATAC-seq peaks between hESCs and hiPSCs, for each class of hiPSC (Primed, TNT, NIP). c) Transcription factors (TFs) with significantly enriched motifs in differential ATAC-seq peaks.

Reviewer 4 - point 24: 16. Lines 477-479 (“With respect to genomic imprinting, TNT-hiPSCs do not show extensive demethylation at imprint control regions”), Extended Figure 7c: TNT_r1 does show significant loss of DNA methylation in ICR. How frequently DNA methylation is lost in ICRs during TNT reprogramming protocol? Can the protocol be optimised to more reliably retain DNA methylation in ICRs? Please comment in the text that the ICRs can be partially demethylated during TNT reprogramming, the current statement does not fully reflect the result.

Response: In the revised manuscript we have included a more comprehensive analysis of ICRs, where the frequency of ICR loss of methylation is reported through statistical comparisons for each ICR for all experiment sets [see response to question 10 (Reviewer 4 - point 18 above)]. Here, both Primed- and TNT-hiPSC lines had 3/67 ICRs with significantly different CG methylation, and these were the same significant ICRs for Primed and TNT (see Response Fig. 19 / Revised Manuscript Extended Data Fig. 6a). We hypothesise that it may indeed be possible to improve this even more through further optimisation of the protocol, however this would likely require extensive testing of different reprogramming parameters, which we think is better suited for future studies focused on further protocol optimisation.

As requested, we have now also revised the text to clarify that the ICRs can be partially demethylated during TNT reprogramming, as follows: “*With respect to genomic imprinting, the TNT-hiPSCs did not show extensive demethylation at imprint control regions, in contrast to NtP-hiPSCs, which more closely resembled naive hiPSCs (Extended Data Fig. 9d)*”.

Reviewer 4 - point 25: 17. Figure 4j-k, Extended Figure 9: correction of differentiation bias is crucial for the claim of the paper, this section requires expansion and more careful characterisation.

Response: As detailed in response to Reviewer 4 - point 4, and in the revised manuscript, we have now added extensive new experimental data to demonstrate the correction of differentiation bias, including production of many new hiPSC lines from multiple different starting lineages, and differentiated all these lines (Fibroblast-, NHEK-, and MSC-derived Primed- and TNT-hiPSCs) into neural stem cells (NSCs), early progenitor endoderm cells, cortical neurons, skeletal muscle, and lung epithelial cells (Revised Manuscript Fig. 5 and Extended Data Figs. 13-15). These new and highly replicated reprogramming and differentiation experiments demonstrate that the correction of the differentiation bias using the TNT method is repeatedly observed across cell of origin and differentiation assays.

Reviewer 4 - point 26: 17a. Please provide more detailed characterisation of the differentiated cells: immunostaining, qRT-PCR for markers, flow cytometry. Cortical neurons have a much broader spectrum of specific markers than NCAM, so as the cells resulted from other differentiation protocols.

Response: As suggested by the reviewer, in the revised manuscript we have included more detailed characterization and quantification for all the differentiations, including analysis by FACS, qPCR and immunostaining (IF) using multiple markers (see Revised Manuscript Fig. 5 and Extended Data Figs. 13-18) amongst other methods, as follows:

Cortical neurons:

Immunostaining (IF): PAX6+, SOX1+

FACS: CD56+/CD57+

Skeletal muscle cells:

Immunostaining (IF): PAX7+, PAX3+

FACS: CD146+/CD56+

Lung epithelial cells:

Immunostaining (IF): TTF1+, GATA6+

FACS: CD47+/EPCAM+

Endoderm:

Immunostaining (IF): SOX17+, FOXA2

Neural Stem Cells:

FACS: NCAM+/FAP-

Single cell RNA-seq

Furthermore, cells were inspected for expected morphology in all cases.

Reviewer 4 - point 27: 17b. Please provide the results of not only independent differentiation experiments, but also of hiPSCs from independent reprogramming experiments.

Response: We apologise if this was not clear in the original manuscript. Please note that in the original manuscript we did provide the results of independent differentiation experiments, and from hiPSCs from independent reprogramming experiments. Furthermore, as described above, in the revised manuscript we have now expanded all the experiments to many more independent reprogramming experiments and new iPSC lines, different cell types of origin, and several differentiation assays performed independently. To more clearly outline the experimental designs and replication, all this information is now summarised in Fig. 5 and Supplementary Table 1 in the revised manuscript.

Reviewer 4 - point 28: 17c. The muscle differentiation protocol seems not to have worked in hESCs. Please show the differentiation protocol that works.

Response: As described above, in the revised manuscript, we have optimised the skeletal muscle differentiation assays to perform better in our hands, and as can be seen in Revised Manuscript Fig. 5, the hESCs now successfully differentiate into skeletal muscle.

Reviewer 4 - point 29: 17d. Please show the outcomes of alternative differentiation protocols to confirm robustness of the results.

Response: As described above, and as suggested by the reviewer, we have now added several alternative differentiation protocols, and in the revised manuscript we performed and quantified differentiation of ESCs and Primed- and TNT-hiPSC lines derived from primary fibroblasts, secondary fibroblasts, mesenchymal stem cells, and keratinocytes into endoderm, cortical neurons, skeletal muscle, lung epithelial cells, and neural stem cells. These new and highly replicated reprogramming and differentiation experiments clearly demonstrate the robustness of our results.

Reviewer 4 - point 30: 17e. Would differentiation bias be corrected if the reprogramming was done with other somatic cell types than fibroblasts (neurons or NSC, blood cells...)?

Response: This question has two interpretations: (1) Would bias be introduced in iPSCs if the reprogramming was done from other somatic cell types? The answer is yes, and this has previously been shown by us and others (e.g. ²⁶; ²⁰; ²⁷⁻²⁹), and we show it here again in our new experiments summarised in Fig. 5. (2) The other interpretation is, will the TNT method correct the bias of iPSCs created from other somatic cell types? The answer is also yes, as we explain above in response to several points. Please see our detailed answer to this question above (Reviewer 4 - point 4, point 25, and point 26) and shown in Revised Manuscript Fig. 5.

Reviewer 4 - point 31: 18. Discussion, lines 614-616 (“the lineage-of-origin differentiation bias in Primed-hiPSCs may be due to heterochromatic memory impacting TF binding dynamics”): it is important to build a mechanistic link between the correction of epigenetic state and the differentiation bias. What genomic elements (promoters or enhancers) associated with differentiation are affected in Primed-hiPSCs? Which of them are corrected in TNT-hiPSCs? Extensive epigenetic profiling performed by the authors allows to address this question.

Response: In our manuscript, we have originally outlined a mechanistic link between the retention of lineage-of-origin repressive chromatin (H3K9me3, LADs, CH methylation) in

Primed-hiPSCs that is associated with altered Primed-hiPSC differentiation capacity compared to hESCs and TNT-hiPSCs. To quote the above referenced discussion section in full:

“If a cell’s response to differentiation cues depends upon how chromatin is spatially organised to make loci available for TF binding⁶⁴, the differentiation bias in Primed-hiPSCs associated with the different cell type of origin may be due to heterochromatic memory impacting TF binding dynamics. Therefore, the spatial organisation of the chromatin in Primed-hiPSCs may be favoring the binding of differentiation TFs at loci that regulate lineage-specific signalling.”

As suggested by the reviewer, to provide further support for this proposed mechanistic link in the discussion, we have now used the differences in ATAC-seq peak signals between Primed-hiPSCs and TNT-hiPSCs as an indicator of likely TF occupancy differences in putative regulatory genomic elements (promoters and enhancers), and examined the regions where these differences occur (see Response Fig. 23 / Revised Manuscript Extended Data Fig. 9a-c).

When examining the regions with accessible chromatin in Primed-hiPSCs, but inaccessible in TNT-hiPSCs and hESCs, the most significant enriched transcription factor motif is NFY which has been reported to promote chromatin accessibility for cell-type-specific master transcription factors³¹. Moreover, Primed-hiPSCs (but not TNT-hiPSCs) show accessible chromatin in regions enriched for TF’s associated with differentiation, such as the Homeobox (HOX) transcription factors (see Response Fig. 23c / Revised Manuscript Extended Data Fig. 9c). This suggests that Primed-hiPSCs (but not TNT-hiPSCs and hESCs) have accessible chromatin at loci that could bind differentiation TFs.

Importantly, this analysis also revealed that the ATAC-seq peaks not present in Primed-hiPSCs, but present in TNT-hiPSCs and hESCs, are enriched for OKSM motifs (see Response Fig. 23c / Revised Manuscript Extended Data Fig. 9c), which strongly suggests that Primed-hiPSCs feature fewer OKSM binding loci compared to hESCs and TNT-hiPSCs. In reference to the second part of the question, “Which of them are corrected in TNT-hiPSC”, our data indicate that they are effectively all corrected (99.3%) (Revised Manuscript Fig. 4c).

Further investigations into how these loci are occupied by TF’s upon differentiation would require careful epigenomic time course analyses of differentiation, which are beyond the scope of this study.

Minor specific comments

Reviewer 4 - point 32: 1. Related to Figure 1 and Extended Data Figure 1: please provide details of the markers used for cell sorting during naïve and primed reprogramming. The antibodies are listed in the Materials and Methods, however this is not mentioned in the paper which particular markers were used for each type of the reprogramming.

Response: As explained in Reviewer 4 - point 9 response, the sorting strategy has been extensively validated in our previous study (Response Table 1 and Liu *et al.*, Nature 2020, Supplementary Figure 1). Details of the markers used are included in Supplementary Table 1.

Reviewer 4 - point 33: 2. Figure 4k, middle panel (ectoderm differentiation result): please

expand the axis, the current limits give a false impression that neuronal differentiation failed in Primed-hiPSC, although it is 50% efficient.

Response: The secondary fibroblast differentiation assays have now been further replicated, expanded, and characterised in more detail and are now presented in Revised Manuscript Fig. 5d, which no longer gives this false impression regarding the Primed-hiPSC.

Reviewer 4 - point 34: 3. Please expand abbreviations (eg “REs” in Figure 4e).

Response: We have now expanded this abbreviation in the figure to “regulatory elements”.

Reviewer 4 - point 35: 4. Supplementary Table 5: for GO enrichment analysis, please provide adjusted p-values rather than p-values and list specific genes belonging to each GO term. This information is important for readers.

Response: The p-values included in this table were already adjusted for multiple comparisons by the software used (GProfiler). We have now changed the column names in revised manuscript Supplementary Table 6 indicating these values are adjusted. We have also now included the lists of genes for each GO term as suggested.

References:

1. Lister, R. *et al.* Hotspots of aberrant epigenomic reprogramming in human induced pluripotent stem cells. *Nature* **471**, 68–73 (2011).
2. Ohi, Y. *et al.* Incomplete DNA methylation underlies a transcriptional memory of somatic cells in human iPS cells. *Nat. Cell Biol.* **13**, 541–549 (2011).
3. Ma, H. *et al.* Abnormalities in human pluripotent cells due to reprogramming mechanisms. *Nature* **511**, 177–183 (2014).
4. Butcher, L. M. *et al.* Non-CG DNA methylation is a biomarker for assessing endodermal differentiation capacity in pluripotent stem cells. *Nat. Commun.* **7**, 10458 (2016).
5. Müller, F.-J., Brändl, B. & Loring, J. F. Assessment of human pluripotent stem cells with PluriTest. in *StemBook* (Harvard Stem Cell Institute, 2012).
6. Koyanagi-Aoi, M. *et al.* Differentiation-defective phenotypes revealed by large-scale analyses of human pluripotent stem cells. *Proc. Natl. Acad. Sci. U. S. A.* **110**, 20569–20574 (2013).
7. Ruiz, S. *et al.* Identification of a specific reprogramming-associated epigenetic signature in human induced pluripotent stem cells. in *Proceedings of the National Academy of Sciences*

- vol. 109 16196–16201 (2012).
8. Müller, F.-J. *et al.* A bioinformatic assay for pluripotency in human cells. *Nat. Methods* **8**, 315–317 (2011).
 9. Ruiz, S. *et al.* Identification of a specific reprogramming-associated epigenetic signature in human induced pluripotent stem cells. *Proc. Natl. Acad. Sci. U. S. A.* **109**, 16196–16201 (2012).
 10. Bock, C. *et al.* Reference Maps of human ES and iPS cell variation enable high-throughput characterization of pluripotent cell lines. *Cell* **144**, 439–452 (2011).
 11. Takashima, Y. *et al.* Resetting Transcription Factor Control Circuitry toward Ground-State Pluripotency in Human. *Cell* **162**, 452–453 (2015).
 12. Choi, J. *et al.* A comparison of genetically matched cell lines reveals the equivalence of human iPSCs and ESCs. *Nat. Biotechnol.* **33**, 1173–1181 (2015).
 13. Zhou, F. *et al.* Reconstituting the transcriptome and DNA methylome landscapes of human implantation. *Nature* **572**, 660–664 (2019).
 14. Theunissen, T. W. *et al.* Systematic Identification of Culture Conditions for Induction and Maintenance of Naive Human Pluripotency. *Cell Stem Cell* **15**, 524–526 (2014).
 15. Pastor, W. A. *et al.* Naive Human Pluripotent Cells Feature a Methylation Landscape Devoid of Blastocyst or Germline Memory. *Cell Stem Cell* **18**, 323–329 (2016).
 16. Liu, X. *et al.* Comprehensive characterization of distinct states of human naive pluripotency generated by reprogramming. *Nat. Methods* **14**, 1055–1062 (2017).
 17. Di Stefano, B. *et al.* Reduced MEK inhibition preserves genomic stability in naive human embryonic stem cells. *Nat. Methods* **15**, 732–740 (2018).
 18. Nefzger, C. M. *et al.* Cell Type of Origin Dictates the Route to Pluripotency. *Cell Rep.* **21**, 2649–2660 (2017).
 19. Liu, X. *et al.* Reprogramming roadmap reveals route to human induced trophoblast stem cells. *Nature* **586**, 101–107 (2020).

20. Polo, J. M. *et al.* Cell type of origin influences the molecular and functional properties of mouse induced pluripotent stem cells. *Nat. Biotechnol.* **28**, 848–855 (2010).
21. Gel, B. *et al.* regioneR: an R/Bioconductor package for the association analysis of genomic regions based on permutation tests. *Bioinformatics* **32**, 289–291 (2016).
22. Salhab, A. *et al.* A comprehensive analysis of 195 DNA methylomes reveals shared and cell-specific features of partially methylated domains. *Genome Biol.* **19**, 150 (2018).
23. van Steensel, B. & Belmont, A. S. Lamina-Associated Domains: Links with Chromosome Architecture, Heterochromatin, and Gene Repression. *Cell* **169**, 780–791 (2017).
24. den Hartogh, S. C., Wolstencroft, K., Mummery, C. L. & Passier, R. A comprehensive gene expression analysis at sequential stages of in vitro cardiac differentiation from isolated MESP1-expressing-mesoderm progenitors. *Sci. Rep.* **6**, 19386 (2016).
25. Wilm, B., James, R. G., Schultheiss, T. M. & Hogan, B. L. M. The forkhead genes, *Foxc1* and *Foxc2*, regulate paraxial versus intermediate mesoderm cell fate. *Dev. Biol.* **271**, 176–189 (2004).
26. Kim, K. *et al.* Epigenetic memory in induced pluripotent stem cells. *Nature* **467**, 285–290 (2010).
27. Phetfong, J. *et al.* Cell type of origin influences iPSC generation and differentiation to cells of the hematoendothelial lineage. *Cell Tissue Res.* **365**, 101–112 (2016).
28. Chlebanowska, P. *et al.* Origin of the Induced Pluripotent Stem Cells Affects Their Differentiation into Dopaminergic Neurons. *Int. J. Mol. Sci.* **21**, (2020).
29. Hu, S. *et al.* Effects of cellular origin on differentiation of human induced pluripotent stem cell-derived endothelial cells. *JCI Insight* **1**, (2016).
30. Poleshko, A. *et al.* Genome-Nuclear Lamina Interactions Regulate Cardiac Stem Cell Lineage Restriction. *Cell* **171**, 573–587.e14 (2017).
31. Oldfield, A. J. *et al.* Histone-fold domain protein NF-Y promotes chromatin accessibility for cell type-specific master transcription factors. *Mol. Cell* **55**, 708–722 (2014).

Reviewer Reports on the First Revision:

Referees' comments:

Referee #2 (Remarks to the Author):

The authors supplemented a large amount of data, and it made the manuscript very strong. Revised Manuscript Figure 5 is very useful to know the variation of the differentiation capacity for all researchers in the field, and I very much appreciate the authors' effort.

While I strongly recommend editors publish this work in Nature, I have a few suggestions to modify the presentation of the data in Revised Manuscript Figure 5 and Revised Manuscript Extended Data Figure 11.

Figure 5a bottom panel, 5b and 5d: It is not clear what the replicates mean from the manuscript, while the authors wrote it in the response. It could be informative to show that information in Figure 5a bottom panel, and perhaps indicate the data from independent lines with different tones of green/yellow in Figures 5b and 5d? I found it interesting to know when the variation is big (e.g. Lung differentiation/HDF) if data from 3 independent experiments with one line cluster together or are scattered in a wide range.

Figure 5b, 5d, and 5g: Having hESC in the box of 2o fibroblasts is a bit confusing. I feel it is better if hESCs have an independent box, but with an indication that it has the same genetic background as 2o fibroblast samples.

I assume the stars on the top of boxes in Figure 5b and 5d are statistical significance in the difference between Primed-hiPSC and TNT-hiPSCs. But isn't it better to have a comparison between hESC vs Primed-hiPSC, and hESC vs TNT-hiPSCs, as the authors use hESCs as a gold standard throughout the manuscript?

Extended Data Figure 11c and 11e: Can you include hESC? The authors have the data, don't they?

I would be happy if the editors discuss with the authors whether the above change needs to be implemented.

Referee #4 (Remarks to the Author):

Buckberry et al presented a revised version of the manuscript on the modified method of somatic reprogramming. I would like to acknowledge their work, which significantly strengthened the manuscript. Specifically, there was a big effort to support the section about the improved differentiation potential of TNT-hiPSCs (Revised Manuscript Figure 5, Extended Figures 13-18). Most of my points were adequately addressed, nevertheless there are minor unresolved questions and remaining overstatements that require attention. The manuscript is close to the shape appropriate for acceptance provided the title and the text of the manuscript are adjusted to address those

points.

Major comments

1. The title of the manuscript “Transient naive reprogramming eliminates epigenetic memory and differentiation bias in human iPSCs” is overstated for both aspects, epigenetic and differentiation. For the epigenetic memory: whilst the authors showed that in fibroblast-derived TNT-hiPSCs a large fraction of CG-DMRs resembles their state in hESCs, this fraction is much more modest in case of MSC- and keratinocyte-derived TNT-hiPSCs (Extended Data Figure 12a, new results). Therefore “elimination of epigenetic memory” is far from complete.

For the differentiation: whilst differentiation efficiency of TNT-hiPSCs is in many cases higher than of Primed-hiPSCs of the same origin (Figure 5), this is still not always the case (Figure 5d: cortical neurons from keratinocyte-derived hiPSCs; skeletal muscle from HDF-derived hiPSCs), moreover their response is variable in replicates and doesn't always reach the efficiency of hESC (Figure 5b). Most importantly – the interesting observation is that the relative differentiation efficiency of hiPSCs derived by the same reprogramming method from different cell types shows very similar trends when comparing Primed-hiPSCs and TNT-hiPSCs. This is clear from cortical neuron differentiation and skeletal muscle differentiation (compare the relative efficiencies of HDF-, NHEK-, MSC-, and 2@fibroblast-derived Primed-hiPSCs between each other, and the same for TNT-hiPSCs). This strongly suggests that the origin still does have an impact on differentiation regardless of the reprogramming method. Therefore, differentiation bias is not eliminated, even when differentiation is improved.

I would suggest changing the title to avoid overstatements; “attenuates”, “ameliorates”, “mitigates” or another similar term might be more appropriate instead of “eliminates”.

I would also suggest to reconsider the term “differentiation bias”, rather emphasizing improved differentiation abilities of TNT-hiPSCs.

2. Line 641-643 “These reprogramming and differentiation experiments provide strong evidence that the epigenetic memory in Primed-hiPSCs results in widespread reduced differentiation capacity that can be nullified by TNT reprogramming” – there is no evidence that epigenetic memory is the cause of the reduced differentiation capacity, this is a correlation. The reduced differentiation capacity is attenuated but not nullified by TNT reprogramming (see comment above).

3. Line 327-329 (“Furthermore, it is important to emphasise that extended culturing of cells in naive conditions is known to cause an increase in the frequency of genetic abnormalities, as we and others have reported^{19,25,39}”): the aforementioned manuscripts reveal genetic instability of naïve hPSCs in 5iLAF system, whereas cells in other culture conditions such as t2iLGY were stable (see Liu ... Polo, 2019, ref 19). The original manuscripts on various naïve culture systems show the same trend – genetic instability was mentioned in the original paper on 5iLAF (Theunissen et al, 2014); but t2iLGY / PXGL conditions support genetically normal lines (Guo et al 2016, Guo et al 2017). Therefore, this statement can be applicable only to specific conditions, not to the pluripotency state, and is incorrect.

Minor comments

1. Response to the response to point 2: by no means I asked the authors to generate their own low-passage hESC, this is a strange interpretation of my comment. The manuscripts that I mentioned contain DNA methylation data from hESC on different passages, for example Weissbein et al 2017 analysed a large collection of published sequencing results from hESCs extending from passage 3 to passage 105. Therefore, this comparison could have been done. I don't insist on this analysis, although I do think that if this work focuses on culture-induced epigenetic aberrations in hiPSCs, it should also consider culture-induced epigenetic aberrations in the reference hESCs. Nevertheless, I agree that the comparison of Primed-hiPSC and TNT-hiPSC is the most informative (provided the passage is matching). I encourage to include specific passage number for all samples to Supplementary Table 1.

2. Extended Figure 8c (new results), lines 476-481 ("This showed that Primed-hiPSCs cluster most closely with fibroblasts, and that although for many genes Primed-hiPSCs show lower expression than fibroblasts, their expression was closer to that of fibroblasts than of TNT-hiPSCs for mesoderm-related genes (Extended Data Fig. 8c)": firstly, the figure is based on a very small selection of genes; secondly, it is not obvious that Primed-hiPSCs cluster most closely with fibroblasts. Primed-hiPSCs cluster with other hPSCs in this figure, not with fibroblasts; and because the branches of similarity trees can be rotated, Primed-hiPSCs can be likewise placed to the far most position from fibroblasts.

3. Related to Extended Figure 8e (new results), lines 487-492 ("To further inspect fibroblast specific genes that retain their expression in Primed-hiPSCs, we selected genes that were differentially expressed and higher in fibroblasts compared to hESCs (fibroblast-specific) and not differentially expressed between Primed-hiPSCs and fibroblasts (expression retained in Primed hiPSCs). This showed that Primed-hiPSCs feature a gene expression signature with elements of the fibroblast state (Extended Data Fig. 8e)": this approach is supervised, it is designed to find a group of genes that is common between Primed-hiPSCs and fibroblasts, but not hESCs, isn't it? For any three groups of cells, it is very likely to find a proportion of genes shared between the two, but not the third, randomly by chance. So, what is the conclusion from this analysis?

4. Related to Extended Figure 16-18 (new result): is it possible to show qPCR of differentiated cells obtained from Primed- and TNT-hiPSC side-by-side for comparison? This comparison is the key for the paper conclusion.

5. Lines 614-615 ("when assessing primitive endoderm differentiation"): do the authors mean "definitive endoderm"?

Author Rebuttals to First Revision:

Point by point response to comments on Nature manuscript 2021-05-08601B

We extend our sincere thanks for the constructive feedback, time and effort in considering our manuscript. We believe we have addressed all comments and requested changes, which has again helped improve our study. Below we provide a point-by-point response to all comments and suggestions.

Referee #2 (Remarks to the Author):

The authors supplemented a large amount of data, and it made the manuscript very strong. Revised Manuscript Figure 5 is very useful to know the variation of the differentiation capacity for all researchers in the field, and I very much appreciate the authors' effort.

While I strongly recommend editors publish this work in Nature, I have a few suggestions to modify the presentation of the data in Revised Manuscript Figure 5 and Revised Manuscript Extended Data Figure 11.

Reviewer 2 - point 1: Figure 5a bottom panel, 5b and 5d: It is not clear what the replicates mean from the manuscript, while the authors wrote it in the response. It could be informative to show that information in Figure 5a bottom panel, and perhaps indicate the data from independent lines with different tones of green/yellow in Figures 5b and 5d? I found it interesting to know when the variation is big (e.g. Lung differentiation/HDF) if data from 3 independent experiments with one line cluster together or are scattered in a wide range.

Response: In order to make this more clear we have now added the specific line and replicate information to Supplementary Table 10. In regards to adding colours and shapes to the figures, we did attempt this, however with the large number of lines and replicates used for that figure, adding colours and shapes to the plots makes them confusing and hard to interpret. We have inspected the data in regards to the Reviewer's point, and see that the lines for each reprogramming strategy group are largely overlapping, and the results are not dominated by one particular line, as shown in the example below (Response Fig. 1) that is the specific example used by Reviewer 2 above.

Response Fig. 1. Quantification of lung epithelial differentiation of NHEK-hiPSCs, showing the proportion of cells positive for CD47 and EPCAM by FACS. Points represent individual reprogramming replicates, colours represent independent hiPSC reprogramming lines.

Reviewer 2 - point 2: Figure 5b, 5d, and 5g: Having hESC in the box of 2o fibroblasts is a bit confusing. I feel it is better if hESCs have an independent box, but with an indication that it has the same genetic background as 2o fibroblast samples.

Response: Respectfully, we feel that presenting this data with the hESCs in the same box as the other lines of the isogenic reprogramming system is consistent with the structure of the figure, where the data is grouped by genetic background, and that separating the hESCs may reduce the clarity of this connection. In its current format, it serves to more clearly show how the difference between hESCs and Primed-hiPSC lines are corrected in the TNT-hiPSC lines.

Reviewer 2 - point 3: I assume the stars on the top of boxes in Figure 5b and 5d are statistical significance in the difference between Primed-hiPSC and TNT-hiPSCs. But isn't it better to have a comparison between hESC vs Primed-hiPSC, and hESC vs TNT-hiPSCs, as the authors use hESCs as a gold standard throughout the manuscript?

Extended Data Figure 11c and 11e: Can you include hESC? The authors have the data, don't they?

Response: In regard to the indicators of statistical significance between hESC vs Primed-hiPSC, and hESC vs TNT-hiPSCs, we structured Figure 5 so that all the comparisons we have done are grouped within the same genetic background. While we do maintain that hESCs are the gold standard that we aim to achieve with hiPSC reprogramming, comparing the differentiation of non-genetically matched hiPSCs and hESCs would be confounded with any differentiation bias due to the different genetic backgrounds. This was the primary reason that we generated the hESC-derived isogenic

reprogramming system and resulting entire data set, so that we could compare genetically matched hiPSCs and hESCs. In the revised Supplementary Table 9, we have added the statistics for direct comparisons between hESCs and their isogenic hiPSC counterpart lines. When performing these statistical tests, there are (as one could intuit from Fig. 5d) comparisons where TNT-hiPSCs differentiate marginally more efficiently than hESCs. Given the size of the panels in Fig. 5b,d,g, we feel that the addition of additional comparison statistics to the figure would make the data more confusing to interpret. Therefore, we have chosen to report these hESC vs Primed-hiPSC and hESC vs TNT-hiPSC in Supplementary Table 9. We hope that this delivers a suitable balance to maintain figure interpretability while providing the comparisons suggested by the Reviewer.

In regard to the request to include hESCs in the Extended Data Figures, we have now included the hESCs in the revised plots, which are now presented in the revised manuscript in Extended Data Figures 9c and 9e.

Referee #4 (Remarks to the Author):

Buckberry et al presented a revised version of the manuscript on the modified method of somatic reprogramming. I would like to acknowledge their work, which significantly strengthened the manuscript. Specifically, there was a big effort to support the section about the improved differentiation potential of TNT-hiPSCs (Revised Manuscript Figure 5, Extended Figures 13-18). Most of my points were adequately addressed, nevertheless there are minor unresolved questions and remaining overstatements that require attention. The manuscript is close to the shape appropriate for acceptance provided the title and the text of the manuscript are adjusted to address those points.

Major comments

Reviewer 4 - point 1: 1. The title of the manuscript “Transient naive reprogramming eliminates epigenetic memory and differentiation bias in human iPS cells” is overstated for both aspects, epigenetic and differentiation.

For the epigenetic memory: whilst the authors showed that in fibroblast-derived TNT-hiPSCs a large fraction of CG-DMRs resembles their state in hESCs, this fraction is much more modest in case of MSC- and keratinocyte-derived TNT-hiPSCs (Extended Data Figure 12a, new results). Therefore “elimination of epigenetic memory” is far from complete.

For the differentiation: whilst differentiation efficiency of TNT-hiPSCs is in many cases higher than of Primed-hiPSCs of the same origin (Figure 5), this is still not always the case (Figure 5d: cortical neurons from keratinocyte-derived hiPSCs; skeletal muscle from HDF-derived hiPSCs), moreover their response

is variable in replicates and doesn't always reach the efficiency of hESC (Figure 5b). Most importantly – the interesting observation is that the relative differentiation efficiency of hiPSCs derived by the same reprogramming method from different cell types shows very similar trends when comparing Primed-hiPSCs and TNT-hiPSCs. This is clear from cortical neuron differentiation and skeletal muscle differentiation (compare the relative efficiencies of HDF-, NHEK-, MSC-, and 2@fibroblast-derived Primed-hiPSCs between each other, and the same for TNT-hiPSCs). This strongly suggests that the origin still does have an impact on differentiation regardless of the reprogramming method. Therefore, differentiation bias is not eliminated, even when differentiation is improved.

I would suggest changing the title to avoid overstatements; “attenuates”, “ameliorates”, “mitigates” or another similar term might be more appropriate instead of “eliminates”.

I would also suggest to reconsider the term “differentiation bias”, rather emphasizing improved differentiation abilities of TNT-hiPSCs.

Response: We agree with the Reviewer's comments in this regard and have revised the manuscript title accordingly, and also reduced its length significantly. The revised title is:

Epigenetic and functional correction of human iPS cells by transient naive reprogramming

Reviewer 4 - point 2: 2. Line 641-643 “These reprogramming and differentiation experiments provide strong evidence that the epigenetic memory in Primed-hiPSCs results in widespread reduced differentiation capacity that can be nullified by TNT reprogramming” – there is no evidence that epigenetic memory is the cause of the reduced differentiation capacity, this is a correlation. The reduced differentiation capacity is attenuated but not nullified by TNT reprogramming (see comment above).

Response: We understand the Reviewer's point, and therefore we have modified the sentence accordingly, as follows:

“These reprogramming and differentiation experiments provide strong evidence that the epigenetic memory in Primed-hiPSCs is associated with widespread reduced differentiation capacity that can be attenuated by TNT reprogramming.”

Reviewer 4 - point 3: 3. Line 327-329 (“Furthermore, it is important to emphasise that extended culturing of cells in naive conditions is known to cause an increase in the frequency of genetic abnormalities, as we and others have reported^{19,25,39}): the aforementioned manuscripts reveal genetic instability of naïve hPSCs in 5iLAF system, whereas cells in other culture conditions such as t2iLGY were stable (see Liu ... Polo, 2019, ref 19). The original manuscripts on various naïve culture systems show the same trend – genetic instability was mentioned in the original paper on 5iLAF (Theunissen et al, 2014); but t2iLGY / PXGL conditions support genetically normal lines (Guo et al 2016, Guo et al 2017). Therefore, this statement can be applicable only to specific conditions, not to the pluripotency state, and is incorrect.

Response: We understand the Reviewer's concern and thus we have now modified the sentence accordingly, as follows:

“Furthermore, it is important to emphasise that extended culturing of cells in naive conditions may cause an increase in the frequency of genetic abnormalities, as we and others have reported^{19,25,39}”

Minor comments

Reviewer 4 - point 4: 1. Response to the response to point 2: by no means I asked the authors to generate their own low-passage hESC, this is a strange interpretation of my comment. The manuscripts that I mentioned contain DNA methylation data from hESC on different passages, for example Weissbein et al 2017 analysed a large collection of published sequencing results from hESCs extending from passage 3 to passage 105. Therefore, this comparison could have been done. I don't insist on this analysis, although I do think that if this work focuses on culture-induced epigenetic aberrations in hiPSCs, it should also consider culture-induced epigenetic aberrations in the reference hESCs. Nevertheless, I agree that the comparison of Primed-hiPSC and TNT-hiPSC is the most informative (provided the passage is matching). I encourage to include specific passage number for all samples to Supplementary Table 1.

Response: Regarding the cited study (Weissbein et al 2017), they used a different experimental approach (Infinium 450K Methylation beadChips) to profile DNA methylation compared to our use of WGBS throughout our study, which precludes comparison with our data. Regarding passage numbers, wherever this is possible we have added the passage number for the samples in Supplementary Table 1.

Reviewer 4 - point 5: 2. Extended Figure 8c (new results), lines 476-481 (“This showed that Primed-hiPSCs cluster most closely with fibroblasts, and that although for many genes Primed-hiPSCs show lower expression than fibroblasts, their expression was closer to that of fibroblasts than of TNT-hiPSCs for mesoderm-related genes (Extended Data Fig. 8c)”): firstly, the figure is based on a very small selection of genes; secondly, it is not obvious that Primed-hiPSCs cluster most closely with fibroblasts. Primed-hiPSCs cluster with other hPSCs in this figure, not with fibroblasts; and because the branches of similarity trees can be rotated, Primed-hiPSCs can be likewise placed to the far most position from fibroblasts.

Response: We agree with the Reviewer that branches can be rotated and therefore the relative position is not indicative of how “near” Primed-hiPSCs are to fibroblasts in this case.. For this reason, we have changed the wording of these results to more accurately reflect what we observe in this plot, and is not subject to the misinterpretation through branch rotation, as follows:

“This showed that TNT-hiPSCs cluster more closely with hESCs than Primed hiPSCs (Extended Data Fig. 7c)”.

Reviewer 4 - point 6: 3. Related to Extended Figure 8e (new results), lines 487-492 (“To further inspect fibroblast specific genes that retain their expression in Primed-hiPSCs, we selected genes that were differentially expressed and higher in fibroblasts compared to hESCs (fibroblast-specific) and not differentially expressed between Primed-hiPSCs and fibroblasts (expression retained in Primed hiPSCs). This showed that Primed-hiPSCs feature a gene expression signature with elements of the fibroblast state (Extended Data Fig. 8e)”: this approach is supervised, it is designed to find a group of genes that is common between Primed-hiPSCs and fibroblasts, but not hESCs, isn’t it? For any three groups of cells, it is very likely to find a proportion of genes shared between the two, but not the third, randomly by chance. So, what is the conclusion from this analysis?

Response: The analysis presented in Extended Data Fig. 8e (now Extended Data Fig. 7e) was included in response to Reviewer 4’s previous question “*what fibroblast-specific genes retain their expression in Primed-hiPSC?*”. In order to undertake this, it was necessary to perform a supervised subselection to identify such genes for inspection, whereby we first identified the genes that were differentially expressed and higher in fibroblasts vs hESCs (i.e. fibroblast-specific) while also being not differentially expressed between Primed-hiPSCs and fibroblasts (i.e. expression retained in Primed hiPSCs). This showed that there are a subset of fibroblast specific genes that show expression in Primed-hiPSCs but not in TNT-hiPSCs. It is important to state here that the selection of genes was completely independent of what the expression states were in TNT-hiPSCs and NtP-hiPSCs.

In response to the question asked by Reviewer 4 here “*it is designed to find a group of genes that is common between Primed-hiPSCs and fibroblasts, but not hESCs, isn’t it?*”, yes it is designed this way, as this was the most direct way to answer Reviewer 4’s initial question.

In response to the second part of the question: “*For any three groups of cells, it is very likely to find a proportion of genes shared between the two, but not the third, randomly by chance. So, what is the conclusion from this analysis?*”. When we select fibroblast-specific genes with retained expression in Primed-hiPSCs (as requested), this selection was completely agnostic to what the expression level was in TNT-hiPSCs and NtP-hiPSCs. When we inspect these data (see Response Fig. 2 / Extended Data Fig. 7e, below), TNT-hiPSCs and NtP-hiPSCs cluster with hESCs. We view this analysis as an additional line of evidence that TNT-hiPSCs and NtP-hiPSCs feature less cell-of-origin molecular memory than Primed-hiPSCs. When inspecting Extended Data Fig. 7e (below), this does not appear to be a random sub-selection of genes where TNT-hiPSCs and NtP-hiPSCs are more dissimilar to Primed-hiPSC, as most genes for TNT-hiPSCs and NtP-hiPSC are similar to hESCs. Due to the agnostic aspect of the gene selection described above, it is highly unlikely that the differences in expression between the TNT-hiPSCs/NtP-hiPSCs compared to the Primed-hiPSCs would arise by chance. Moreover, we have also performed unsupervised differential expression testing for these samples (Fig. 4b-d, Extended Data Fig. 7a, Supplementary Table 6).

We have slightly modified our results section on this point to state:

“To further inspect fibroblast specific genes that retain their expression in Primed-hiPSCs, we selected genes that were differentially expressed and higher in fibroblasts compared to hESCs

(fibroblast-specific) and not differentially expressed between Primed-hiPSCs and fibroblasts (expression retained in Primed hiPSCs). This showed that Primed-hiPSCs feature a gene expression signature with elements of the fibroblast state not observed in TNT-hiPSCs or NtP hiPSCs (Extended Data Fig. 7e)."

Reviewer 4 - point 7: 4. Related to Extended Figure 16-18 (new result): is it possible to show qPCR of differentiated cells obtained from Primed- and TNT-hiPSC side-by-side for comparison? This comparison is the key for the paper conclusion.

Response: We respectfully disagree with the Reviewer on this point as we do not think that this is a valid way to interpret our data, and the experiments were not designed to undertake this comparison. As requested by the Reviewer previously, these qPCR experiments were designed to show that the cells successfully differentiated into the desired cell type, but not to compare the differentiation efficiency, as differences in qPCR cannot show this. To quantify differences in efficiencies, we performed the FACS and IF quantification experiments (Fig. 5). Indeed, we think that it could be misleading to present the qPCR data as suggested by the Reviewer, which is why we only used it as a qualitative measurement, while we used FACS and IF as our quantitative measures. The value of a qPCR measurement in a heterogeneous cell population will depend on both the expression of the

genes and the number of cells expressing the gene. However, “more differentiation” may not always correspond to higher expression of a given gene. For example, fewer cells may be expressing more of the gene of interest, giving a false impression, and the opposite case is also possible.

Reviewer 4 - point 8: 5. Lines 614-615 (“when assessing primitive endoderm differentiation”): do the authors mean “definitive endoderm”?

Response: Thanks for detecting this typo. We have corrected this.